# Reducing Variance of Stochastic Optimization for Approximating Nash Equilibria in Normal-Form Games

**Linjian Meng** [1]  **Wubing Chen** [1]  **Wenbin Li** [1]  **Tianpei Yang** [1]  **Youzhi Zhang** [2]  **Yang Gao** [1]

## Abstract

Nash equilibrium (NE) plays an important role in game theory. How to efficiently compute an NE in NFGs is challenging due to its complexity and non-convex optimization property. Machine Learning (ML), the cornerstone of modern artificial intelligence, has demonstrated remarkable empirical performance across various applications including non-convex optimization. To leverage non-convex stochastic optimization techniques from ML for approximating an NE, various loss functions have been proposed. Among these, only one loss function is unbiased, allowing for unbiased estimation under the sampled play. Unfortunately, this loss function suffers from high variance, which degrades the convergence rate. To improve the convergence rate by mitigating the high variance associated with the existing unbiased loss function, we propose a novel surrogate loss function named Nash Advantage Loss (NAL). NAL is theoretically proved unbiased and exhibits significantly lower variance than the existing unbiased loss function. Experimental results demonstrate that the algorithm minimizing NAL achieves a significantly faster empirical convergence rates compared to other algorithms, while also reducing the variance of estimated loss value by several orders of magnitude.

## 1. Introduction

Game theory is a powerful tool for modeling multi-agent interactions. A common goal in addressing games is the Nash equilibrium (NE), where no player gains by unilaterally deviating from it. However, computing an NE often involves a complex, non-convex optimization problem. Theoretically, computing an NE is known to be PPAD-complete and thus computationally intractable (Daskalakis et al., 2009).

Machine Learning (ML) (Mitchell & Mitchell, 1997) is a predominant technology in contemporary artificial intelligence, demonstrating remarkable performance in diverse real-world applications including non-convex optimization problems such as image and speech recognition (Deng et al., 2014), natural language processing (Achiam et al., 2023), autonomous vehicles (Bojarski et al., 2016), and financial modeling (Heaton et al., 2017). Since NE computation is known to be a non-convex optimization problem (Gemp et al., 2024), leveraging ML for NE computation presents a promising research direction. However, the application of ML to NE computation remains largely unexplored.

A significant challenge in applying ML to compute an NE is designing an appropriate loss function. Specifically, for an $n$-player, $m$-action, general-sum normal-form game (NFG), storing the payoff matrix requires $nm^n$ entries. As $m$ and $n$ increase, the storage complexity $O(nm^n)$ grows exponentially, making it computationally prohibitive to load the entire payoff matrix into memory for large-scale NFGs. Thus, sampling a portion of the payoff matrix becomes necessary, *i.e.*, minimizing the expectation of a random variable using non-convex stochastic optimization techniques in ML. However, most existing loss functions (Nikaidô & Isoda, 1955; Shoham & Leyton-Brown, 2008; Raghunathan et al., 2019; Gemp et al., 2022; Duan et al., 2023) introduce bias under the sampled play, which can be seen as the stochastic optimization setting in ML, making it infeasible to compute an NE under this setting. To address this bias, Gemp et al. (2024) propose a loss function that can be unbiasedly estimated under the stochastic optimization setting. However, it suffers from high variance, as its value is estimated via the inner product of two independent and identically distributed random variables, whose variance is the square of that of estimating an individual random variable, thus significantly degrading the convergence rate.

To improve the convergence rate by mitigating the high variance associated with the existing unbiased loss function, we

[1]National Key Laboratory for Novel Software Technology, Nanjing University, Nanjing, China [2]Centre for Artificial Intelligence and Robotics, Hong Kong Institute of Science & Innovation, Chinese Academy of Sciences. Correspondence to: Tianpei Yang <tianpei.yang@nju.edu.cn>, Youzhi Zhang <youzhi.zhang@cair-cas.org.hk>.

*Proceedings of the 42nd International Conference on Machine Learning*, Vancouver, Canada. PMLR 267, 2025. Copyright 2025 by the author(s).

propose a novel surrogate loss function called Nash Advantage Loss (NAL). Our key insight is: finding a way to obtain an unbiased estimate of the first-order gradient to eliminate the need for calculating the inner product that introduces high variance. Specifically, previous works have overlooked the fact that commonly used non-convex stochastic optimization techniques in ML (Robbins & Monro, 1951; Bottou, 2010; Kingma & Ba, 2014) require only unbiased estimates of the first-order gradient, not the loss function itself. Consequently, NAL ensures that obtaining an unbiased estimate of its first-order gradient does not require computing the inner product of the two random variables, avoiding the high variance associated with such computations. In addition, we demonstrate that the first-order gradient of NAL can approximate that of a variant of the loss function proposed by Gemp et al. (2024) under certain conditions. This implies that when applying commonly used non-convex stochastic optimization techniques in ML, minimizing NAL approximates the process of minimizing the loss function proposed by Gemp et al. (2024), while mitigating the high variance caused by the inner product of two estimated variables.

We conduct an empirical evaluation of the convergence rates and the variances of the estimated values of the loss functions on eight NFGs from OpenSpiel (Lanctot et al., 2019) and GAMUT (Nudelman et al., 2004). Our results reveal that the algorithm minimizing NAL substantially outperforms algorithms that minimize existing unbiased or biased loss functions in terms of the convergence rate. Additionally, our algorithm exhibits significantly lower variance in the estimated values of its loss function than the algorithm minimizing the existing unbiased loss function. Particularly, compared to the existing unbiased loss function, the variance in estimating the value of NAL is typically reduced by two orders of magnitude. In some games, this variance reduction can even reach six orders of magnitude. Furthermore, we analyze the discrepancies between estimated and true values for different loss functions. Our findings indicate that the difference between the estimated and true values for our loss function is usually two orders of magnitude smaller compared to that of other tested loss functions.

## 2. Related Work

Our research aligns with studies that conceptualize the problem of computing an NE in NFGs as a non-convex optimization problem and address it through non-convex stochastic optimization techniques in ML due to their remarkable empirical performance in solving such problems (Chen et al., 2019; Zou et al., 2019). Specifically, we focus on studies that reduce NE computation to minimize a loss function via non-convex stochastic optimization techniques in ML.

Sampling is critical for solving large-scale NFGs since the action size increases linearly while the shape of the payoff matrix grows exponentially. Although many works (Goktas et al., 2022; Marris et al., 2022; Liu et al., 2024) have investigated computing an NE via optimization techniques of ML, they did not consider whether loss functions would incur bias under the sampled play, which can be interpreted as the stochastic optimization setting in ML. Actually, most existing loss functions (Nikaidô & Isoda, 1955; Shoham & Leyton-Brown, 2008; Raghunathan et al., 2019; Gemp et al., 2022; Duan et al., 2023) will incur bias under the sampled play. Specifically, these functions are biased under the sampled play due to either (i) the presence of a random variable as the argument of a complex, nonlinear function, or (ii) the unclear sampling methods (Gemp et al., 2024). For instance, duality gap-based loss functions (Nikaidô & Isoda, 1955; Shoham & Leyton-Brown, 2008; Duan et al., 2023; Gemp et al., 2022) incur bias through a max operator. Additionally, Gradient-based Nash Iteration (NI) (Raghunathan et al., 2019) is biased due to a projection operator that projects a random variable onto the simplex, which involves a max operator (Chen & Ye, 2011). Moreover, unconstrained optimization methods (Shoham & Leyton-Brown, 2008) that penalize deviation from the simplex lose the ability to sample from strategies when any iterate is no longer within the simplex. To mitigate the bias under the stochastic optimization setting, Gemp et al. (2024) propose a loss function that allows unbiased estimation under this setting. However, it suffers from high variance.

Our approach is different from algorithms that replicate tabular methods, i.e., those that use deep neural networks (DNNs) (LeCun et al., 2015; Goodfellow, 2016) to approximate variables in tabular algorithms without modifying the update rules, such as NFSP (Heinrich & Silver, 2016), PSRO (Lanctot et al., 2017), and Deep CFR (Brown et al., 2019). These algorithms employ non-convex stochastic optimization techniques to train a DNN for approximating variables in tabular algorithms, rather than directly computing an NE. More details can be found in Appendix B.

In addition, while our algorithm bears resemblance to existing simultaneous gradient descent algorithms (Sokota et al., 2023) under a certain specific setting, this setting is not supported by our algorithm (Appendix C). Specifically, even when an NE is learned, existing simultaneous gradient descent algorithms may continue to operate and deviate from this NE rather than halt their progress, while our algorithm cease once the NE is reached. This distinction is clearly demonstrated in our experimental results, which show that under the same set of conditions, our algorithm successfully converges to the NE, while the traditional simultaneous gradient descent algorithms fail to do so. Moreover, we establish a connection between NAL and the loss function in Gemp et al. (2024), which does not hold for any existing simultaneous gradient descent algorithm. Additional details can be found in Appendix C.

## 3. Preliminaries

**Normal-form games** (NFGs) are fundamental games in game theory (Osborne et al., 2004), which consists of players $\mathcal{N} = \{1, 2, \ldots, n\}$, an action set $\mathcal{A}_i$ for each player $i$, and a utility function $u_i$ for each player $i$. Each player $i \in \mathcal{N}$ simultaneously chooses an action $a_i \in \mathcal{A}_i$ and receives a utility $u_i(a_i, a_{-i}) \in [0, 1]$, where $-i$ denotes all players except player $i$. The strategy of player $i$ is represented by $\boldsymbol{x}_i \in \mathcal{X}_i$, and the strategy profile is denoted as $\boldsymbol{x} = \{\boldsymbol{x}_i \in \mathcal{X}_i \mid i \in \mathcal{N}\}$, where $\mathcal{X}_i$ is a $(|\mathcal{A}_i| - 1)$-dimensional simplex. The strategy space of all players is represented by $\mathcal{X} = \times_{i \in \mathcal{N}} \mathcal{X}_i$. Moreover, the interior of $\mathcal{X}$ is denoted as $\mathcal{X}^\circ$. Precisely, for each $\boldsymbol{x} \in \mathcal{X}^\circ$, $\boldsymbol{x}_i(a_i) > 0, \forall i \in \mathcal{N}$ and $a_i \in \mathcal{A}_i$. The utility of player $i$, given that all players follow the strategy profile $\boldsymbol{x} \in \mathcal{X}$, is $u_i(\boldsymbol{x}_i, \boldsymbol{x}_{-i}) = \sum_{\boldsymbol{a} \in \times_{i \in \mathcal{N}} \mathcal{A}_i} u_i(\boldsymbol{a}) \prod_{j \in \mathcal{N}} \boldsymbol{x}_j(a_j)$, where $a_j \in \mathcal{A}_j$ denotes player $j$'s component of the joint action $\boldsymbol{a}$.

**Nash equilibrium** (NE) describes a rational behavior where no player can benefit by unilaterally deviating from the equilibrium. As analyzed in Facchinei (2003), if the utility function of each player $i$ is concave over $\mathcal{X}_i$, an NE $\boldsymbol{x}$ is such that $\langle \nabla_{\boldsymbol{x}_i} u_i(\boldsymbol{x}), \boldsymbol{x}_i - \boldsymbol{x}_i \rangle \leq 0, \forall i \in \mathcal{N}$ and $\boldsymbol{x} \in \mathcal{X}$. This concavity condition is satisfied in NFGs since the utility function of each player $i$ is linear over $\mathcal{X}_i$. We denote the set of NE by $\mathcal{X}^*$. If the utility function of each player $i$ is concave over $\mathcal{X}_i$, a well-known metric to measure the distance from the strategy profile $\boldsymbol{x}$ to NE is the duality gap: $\mathrm{dg}(\boldsymbol{x}) = \sum_{i \in \mathcal{N}} \max_{\boldsymbol{x}_i' \in \mathcal{X}_i} \langle \nabla_{\boldsymbol{x}_i} u_i(\boldsymbol{x}), \boldsymbol{x}_i' - \boldsymbol{x}_i \rangle$. If and only if $\mathrm{dg}(\boldsymbol{x}) = 0$, $\boldsymbol{x} \in \mathcal{X}^*$. If $\mathrm{dg}(\boldsymbol{x}) = \delta$, then $\boldsymbol{x}$ is a $\delta$-NE. $\mathcal{X}^{*,\circ}$ denotes interior NE that $\forall \boldsymbol{x}^* \in \mathcal{X}^{*,\circ}, \boldsymbol{x}_i^*(a_i) > 0, \forall i \in \mathcal{N}, a_i \in \mathcal{A}_i$. The duality gap is the upper bound of exploitability: $\exp(\boldsymbol{x}) = \sum_{i \in \mathcal{N}} (\max_{\boldsymbol{x}_i'} u_i(\boldsymbol{x}_i', \boldsymbol{x}_{-i}) - u_i(\boldsymbol{x}_i, \boldsymbol{x}_{-i})) / |\mathcal{N}| \leq \mathrm{dg}(\boldsymbol{x}) / |\mathcal{N}|$, as $u_i(\cdot)$ is linear over $\mathcal{X}_i$.

**Existing unbiased loss function.** To our knowledge, the only known unbiased loss function for approximating an NE is proposed by Gemp et al. (2024). The key insight of this loss function is that gradients of all actions, w.r.t. an interior strategy profile $\boldsymbol{x} \in \mathcal{X}^\circ$, are equal if and only if $\boldsymbol{x} \in \mathcal{X}^*$ when the utility function of each player $i$ is concave over $\mathcal{X}_i$. Formally, for any $\boldsymbol{x} \in \mathcal{X}^\circ$ and $i \in \mathcal{N}$, $\forall a_i, a_i' \in \mathcal{A}_i, \nabla_{\boldsymbol{x}_i} u_i(\boldsymbol{x})(a_i) = \nabla_{\boldsymbol{x}_i} u_i(\boldsymbol{x})(a_i')$ if and only if $\boldsymbol{x} \in \mathcal{X}^{\circ,*}$ when the utility function $u_i(\boldsymbol{x})$ of each player $i$ is concave over $\mathcal{X}_i$ (Gemp et al., 2024). To ensure that the interior NE always exists, they add an entropy $-\tau \boldsymbol{x}_i^{\mathrm{T}} \log \boldsymbol{x}_i$ to each player's utility function, where $\tau > 0$ is a constant. From their analysis, the addition of entropy guarantees that all equilibria of the regularization game with utility function $u_i^\tau(\boldsymbol{x}) = u_i(\boldsymbol{x}) - \tau \boldsymbol{x}_i^{\mathrm{T}} \log \boldsymbol{x}_i$ are interior. Formally, given a strategy profile $\boldsymbol{x} \in \mathcal{X}$, their loss function is defined as follows:

$$\mathcal{L}_G^\tau(\boldsymbol{x}) = \sum_{i \in \mathcal{N}} \|\boldsymbol{F}_i^{\tau,\boldsymbol{x}} - \overline{\boldsymbol{F}_i^{\tau,\boldsymbol{x}}}\|_2^2, \tag{1}$$

where $\boldsymbol{F}_i^{\tau,\boldsymbol{x}} = -\nabla_{\boldsymbol{x}_i} u_i^\tau(\boldsymbol{x}) = -\nabla_{\boldsymbol{x}_i} u_i(\boldsymbol{x}) + \tau \log \boldsymbol{x}_i$ and $\overline{\boldsymbol{F}_i^{\tau,\boldsymbol{x}}} = \sum_{a_i \in \mathcal{A}_i} \boldsymbol{F}_i^{\tau,\boldsymbol{x}}(a_i) / |\mathcal{A}_i| \mathbf{1}$. As the utility function $u_i^\tau(\cdot)$ of each player $i$ is concave over $\mathcal{X}_i$, $\forall a_i, a_i' \in \mathcal{A}_i, \nabla_{\boldsymbol{x}_i} u_i^\tau(\boldsymbol{x})(a_i) = \nabla_{\boldsymbol{x}_i} u_i^\tau(\boldsymbol{x})(a_i')$ if and only if $\boldsymbol{x}$ is an NE of the regularization game. In other words, $\mathcal{L}_G^\tau(\boldsymbol{x}) = 0$ if and only if $\boldsymbol{x}$ is an NE of the regularization game. By gradually decreasing $\tau$, the sequence of NEs of the regularization games converges to an NE of the original game. The advantage of this function is that this function can be unbiasedly estimated given two independent unbiased estimations of $\boldsymbol{F}_i^{\tau,\boldsymbol{x}}$.

To improve the readability, we include a table of notations and definitions, as shown in Appendix A.

## 4. Our Method

To the best of our knowledge, Gemp et al. (2024) propose the only unbiased loss function that enables unbiased estimation when computing an NE by using non-convex stochastic optimization techniques of ML. However, this loss function often exhibits high variance, resulting in significant instability that degrades the convergence rate. To address the high variance, we propose a novel surrogate loss function, termed the Nash Advantage Loss (NAL).

### 4.1. Overview of NAL

Our key insight is: finding a way to obtain an unbiased estimate of the first-order gradient to eliminate the need for calculating the inner product, which introduces high variance. In particular, the insight comes from a fact overlooked in previous works that commonly used non-convex stochastic optimization techniques of ML (Robbins & Monro, 1951; Bottou, 2010; Kingma & Ba, 2014) require only unbiased estimates of the first-order gradient.

**Lemma 4.1.** *For any vector $\boldsymbol{b} \in \mathbb{R}^d$ and any $\boldsymbol{y}$ in a $(d-1)$-dimensional simplex, the equation $\boldsymbol{b} - \langle \boldsymbol{b}, \boldsymbol{y} \rangle \mathbf{1} = \mathbf{0}$ holds if and only if all coordinates of $\boldsymbol{b}$ are all equal to each other.*

Specifically, NAL aims to ensure that (i) its first-order gradient can be estimated without bias by using a single random variable to reduce the variance, and (ii) its first-order gradient with respect to $\boldsymbol{x} \in \mathcal{X}^\circ$ equals $\mathbf{0}$ if and only if $\boldsymbol{x} \in \mathcal{X}^{*,\circ}$ to ensure that the algorithm stops once an NE is learned. To achieve these, we build on the key insight from the loss function in Gemp et al. (2024)—where for any $\boldsymbol{x} \in \mathcal{X}^\circ$, $i \in \mathcal{N}$, and $a_i, a_i' \in \mathcal{A}_i, \nabla_{\boldsymbol{x}_i} u_i(\boldsymbol{x})(a_i) = \nabla_{\boldsymbol{x}_i} u_i(\boldsymbol{x})(a_i')$ if and only if $\boldsymbol{x} \in \mathcal{X}^{*,\circ}$—and recognize that $\nabla_{\boldsymbol{x}_i} u_i(\boldsymbol{x})$ can be estimated without bias using a single random variable (e.g., via importance sampling). Then, inspired by Lemma 4.1, we define NAL's first-order gradient as the difference between the gradient of the utility function of the game and the inner product of the utility function's gradient with any arbitrary given strategy $\hat{\boldsymbol{x}}$. This difference is the advantage

of each action's gradient for making the gradients of actions more uniform. Formally, the first-order gradient can be

$$[-\nabla_{\boldsymbol{x}_i} u_i(\boldsymbol{x}) + \langle \nabla_{\boldsymbol{x}_i} u_i(\boldsymbol{x}), \hat{\boldsymbol{x}}_i \rangle \mathbf{1} \mid i \in \mathcal{N}].$$

As $\forall \boldsymbol{x} \in \mathcal{X}^{\circ}$, $i \in \mathcal{N}$ and $a_i, a_i' \in \mathcal{A}_i$, $\nabla_{\boldsymbol{x}_i} u_i(\boldsymbol{x})(a_i) = \nabla_{\boldsymbol{x}_i} u_i(\boldsymbol{x})(a_i')$ if and only if $\boldsymbol{x} \in \mathcal{X}^{*,\circ}$, from Lemma 4.1, we have that $\forall \boldsymbol{x} \in \mathcal{X}^{\circ}$, $[-\nabla_{\boldsymbol{x}_i} u_i(\boldsymbol{x}) + \langle \nabla_{\boldsymbol{x}_i} u_i(\boldsymbol{x}), \hat{\boldsymbol{x}}_i \rangle \mathbf{1} \mid i \in \mathcal{N}] = \mathbf{0}$ if and only if $\boldsymbol{x} \in \mathcal{X}^{*,\circ}$. In addition, to ensure that the interior NE always exists, as done in Gemp et al. (2024), we add an entropy $-\tau \boldsymbol{x}_i^{\mathrm{T}} \log \boldsymbol{x}_i$ to each player's utility, where $\tau > 0$ is a constant. As we mentioned above, Gemp et al. (2024) show that the additional entropy guarantees that all NE of the regularization game, with utility function $u_i^{\tau}(\boldsymbol{x}) = u_i(\boldsymbol{x}) - \tau \boldsymbol{x}_i^{\mathrm{T}} \log \boldsymbol{x}_i$, are interior.

Now, we provide the formal definition of NAL. Given a strategy profile $\boldsymbol{x} \in \mathcal{X}$, NAL is defined as

$$\mathcal{L}_{NAL}^{\tau}(\boldsymbol{x}) = \sum_{i \in \mathcal{N}} \langle sg[\boldsymbol{F}_i^{\tau,\boldsymbol{x}} - \langle \boldsymbol{F}_i^{\tau,\boldsymbol{x}}, \hat{\boldsymbol{x}}_i \rangle \mathbf{1}], \boldsymbol{x}_i \rangle, \quad (2)$$

where $\hat{\boldsymbol{x}} = [\hat{\boldsymbol{x}}_0, \hat{\boldsymbol{x}}_1, \cdots, \hat{\boldsymbol{x}}_{|\mathcal{N}|-1}]$ can be any strategy profile in $\mathcal{X}$ (notably, $\forall i \in \mathcal{N}$, $\hat{\boldsymbol{x}}_i \neq \mathbf{0}$ and must in $\mathcal{X}_i$), $\boldsymbol{F}_i^{\tau,\boldsymbol{x}} = -\nabla_{\boldsymbol{x}_i} u_i^{\tau}(\boldsymbol{x}) = -\nabla_{\boldsymbol{x}_i} u_i(\boldsymbol{x}) + \tau \log \boldsymbol{x}_i$ is defined in Eq. (1), and $sg[\cdot]$ is the stop-gradient operator that implies the term in this operator is not involved in gradient backpropagation, i.e., for any variable $\boldsymbol{b} \in \mathbb{R}^d$, $sg[\boldsymbol{b}] = \boldsymbol{b}$ while $\nabla_{\boldsymbol{b}} sg[\boldsymbol{b}] = \mathbf{0}$ (see details in Appendix D). Therefore, in Eq. (2), $\boldsymbol{x}_i$ participates in gradient backpropagation, whereas $sg[\boldsymbol{F}_i^{\tau,\boldsymbol{x}} - \langle \boldsymbol{F}_i^{\tau,\boldsymbol{x}}, \hat{\boldsymbol{x}}_i \rangle \mathbf{1}]$ do not. Consequentially, we obtain $\nabla_{\boldsymbol{x}_i} \mathcal{L}_{NAL}^{\tau}(\boldsymbol{x}) = sg[\boldsymbol{F}_i^{\tau,\boldsymbol{x}} - \langle \boldsymbol{F}_i^{\tau,\boldsymbol{x}}, \hat{\boldsymbol{x}}_i \rangle \mathbf{1}]$. As $\hat{\boldsymbol{x}}_i$ in NAL can be any strategy, not just $\boldsymbol{x}_i$, we are free to employ any sampling strategy to estimate $\nabla_{\boldsymbol{x}_i} \mathcal{L}_{NAL}^{\tau}(\boldsymbol{x})$ to further reduce the variance.

While other loss functions do not include the stop-gradient operator in their definitions, in practice, these loss functions must employ the stop-gradient operator when solving real-world games. This is because $\nabla_{\boldsymbol{x}_i} u_i(\boldsymbol{x})$ in $\boldsymbol{F}_i^{\tau,\boldsymbol{x}}$ cannot feasibly participate in gradient backpropagation. Enabling $\boldsymbol{F}_i^{\tau,\boldsymbol{x}}$ to participate in backpropagation would require iterating over all action pairs for every two players, as done in Gemp et al. (2022) and Gemp et al. (2024), which is practically infeasible in real-world games. More details about the implementation of other loss functions are in Appendix G.

**Unbiased estimation of NAL.** Assume we can obtain an unbiased estimate of $\boldsymbol{F}_i^{\tau,\boldsymbol{x}}$, which can be achieved through importance sampling, as described in Section 4.3. Since $\boldsymbol{F}_i^{\tau,\boldsymbol{x}}$ is estimated without bias and $\hat{\boldsymbol{x}}_i$ is given, we have that $\langle \boldsymbol{F}_i^{\tau,\boldsymbol{x}}, \hat{\boldsymbol{x}}_i \rangle$ remains unbiased. By using the unbiased estimates of $\boldsymbol{F}_i^{\tau,\boldsymbol{x}}$ and $\langle \boldsymbol{F}_i^{\tau,\boldsymbol{x}}, \hat{\boldsymbol{x}}_i \rangle$, we can obtain an unbiased estimate of $\nabla_{\boldsymbol{x}_i} \mathcal{L}_{NAL}^{\tau}(\boldsymbol{x})$. Then, with the unbiased estimate of $\nabla_{\boldsymbol{x}_i} \mathcal{L}_{NAL}^{\tau}(\boldsymbol{x})$ and knowledge of $\boldsymbol{x}_i$, an unbiased estimate of $\mathcal{L}_{NAL}^{\tau}(\boldsymbol{x})$ is obtained. This unbiased estimate

of $\mathcal{L}_{NAL}^{\tau}(\boldsymbol{x})$ is used for non-convex stochastic optimization techniques in ML to update the strategy profile. Further details on the unbiased estimation process are in Section 4.3.

**Relationship between duality gap in the regularization game and NAL.** As analyzed in Gemp et al. (2024), $\forall a_i, a_i' \in \mathcal{A}_i$, $\nabla_{\boldsymbol{x}_i} u_i^{\tau}(\boldsymbol{x})(a_i) = \nabla_{\boldsymbol{x}_i} u_i^{\tau}(\boldsymbol{x})(a_i')$ holds if and only if $\boldsymbol{x}$ is an NE of the regularization game with the utility function $u_i^{\tau}(\boldsymbol{x})$. Then, from Lemma 4.1, we have that $\nabla_{\boldsymbol{x}} \mathcal{L}_{NAL}^{\tau}(\boldsymbol{x}) = \mathbf{0}$ if and only if $\boldsymbol{x}$ is an NE of the regularization game with the utility function $u_i^{\tau}(\boldsymbol{x})$. A formal relationship between the duality gap of a strategy profile $\boldsymbol{x}$ in the regularization game and the gradient of NAL is in Theorem 4.2. The proof of Theorem 4.2 depends on the properties of the tangent residual (Cai et al., 2022).

**Theorem 4.2** (Proof is in Appendix E.2). *The duality gap of a strategy profile $\boldsymbol{x}$ in the regularization game with the utility function $u_i^{\tau}(\boldsymbol{x}) = u_i(\boldsymbol{x}) - \tau \boldsymbol{x}_i^{\mathrm{T}} \log \boldsymbol{x}_i$ is bounded as:*

$$dg^{\tau}(\boldsymbol{x}) = \sum_{i \in \mathcal{N}} \max_{\boldsymbol{x}_i' \in \mathcal{X}_i} \langle \nabla_{\boldsymbol{x}_i} u_i^{\tau}(\boldsymbol{x}), \boldsymbol{x}_i' - \boldsymbol{x}_i \rangle$$
$$\leq C_0 \| \nabla_{\boldsymbol{x}} \mathcal{L}_{NAL}^{\tau}(\boldsymbol{x}) \|_2,$$

*where $C_0$ is a game-dependent constant.*

For the exploitability in the regularization game, we have $\exp^{\tau}(\boldsymbol{x}) = \sum_{i \in \mathcal{N}} (\max_{\boldsymbol{x}_i'} u_i^{\tau}(\boldsymbol{x}_i', \boldsymbol{x}_{-i}) - u_i^{\tau}(\boldsymbol{x}_i, \boldsymbol{x}_{-i})) / |\mathcal{N}| \leq dg^{\tau}(\boldsymbol{x}) / |\mathcal{N}|$ since the function $u_i(\boldsymbol{x}_i)$ is linear and $-\tau \boldsymbol{x}_i^{\mathrm{T}} \log \boldsymbol{x}_i$ is concave over $\mathcal{X}_i$, respectively, as well as for any concave function $f(\cdot)$ with any $\boldsymbol{u}, \boldsymbol{v}$ in its domain, the inequality $f(\boldsymbol{u}) - f(\boldsymbol{v}) \leq \langle \nabla f(\boldsymbol{v}), \boldsymbol{u} - \boldsymbol{v} \rangle$ holds.

**Relationship between duality gap in the original game and NAL.** NAL ensures that a zero point of the first-order gradient of NAL corresponds to an NE of the regularization game rather than the original game. To find an NE of the original game, we establish a precise relationship between the duality gap in the original game and NAL, as shown in Theorem 4.3. This relationship allows us to approximate an NE of the original game by minimizing NAL. Specifically, by progressively decreasing the value of $\tau$, we guarantee that the sequence of NEs of the regularization games, characterized by the utility function $u_i^{\tau}(\boldsymbol{x}) = u_i(\boldsymbol{x}) - \tau \boldsymbol{x}_i^{\mathrm{T}} \log \boldsymbol{x}_i$, converges to the set of the NE of the original game.

**Theorem 4.3** (Proof is in Appendix E.4). *The duality gap of a strategy profile $\boldsymbol{x}$ in the original game is bounded as:*

$$dg(\boldsymbol{x}) \leq \tau C_1 + C_2 \| \nabla_{\boldsymbol{x}} \mathcal{L}_{NAL}^{\tau}(\boldsymbol{x}) \|_2,$$

*where $C_1$ and $C_2$ are game-dependent constants.*

From the analysis above, we observe that NAL learns the global minimum of the loss function proposed by Gemp

et al. (2024), without requiring the inner product between the two estimated variables that introduces high variance (Section 4.2). Therefore, NAL can be viewed as a surrogate for the loss function in Gemp et al. (2024), which is why we refer to NAL as a surrogate loss function. Furthermore, as demonstrated in Appendix C and H, under specific conditions, the first-order gradient of NAL approximates that of a variant of the loss function proposed by Gemp et al. (2024), while mitigating high variance. It implies that minimizing NAL approximates the process of minimizing the loss function proposed by Gemp et al. (2024)

### 4.2. Analysis of Variances of NAL and Existing Unbiased Loss Function

We now analyze the variance in the estimated values of NAL and the unbiased loss function defined in Eq. (7). We demonstrate that when the variance in estimating the value of NAL is $O(\sigma)$, that of the unbiased loss function defined in Eq. (7) may be $O(\sigma^2)$, where $\sigma > 0$ is a constant.

Firstly, assume that the components of the vector $\boldsymbol{F}_i^{\tau,\boldsymbol{x}} - \langle \boldsymbol{F}_i^{\tau,\boldsymbol{x}}, \hat{\boldsymbol{x}}_i \rangle \mathbf{1}$ at each $\boldsymbol{a}_i \in \mathcal{A}_i$ are estimated independently, with the variance for each estimation being less than $\sigma$. Specifically, let the estimation of $\boldsymbol{F}_i^{\tau,\boldsymbol{x}} - \langle \boldsymbol{F}_i^{\tau,\boldsymbol{x}}, \hat{\boldsymbol{x}}_i \rangle \mathbf{1}$ at action $\boldsymbol{a}_i \in \mathcal{A}_i$ be denoted as $\hat{\boldsymbol{g}}_i^{\tau,\boldsymbol{x}}(a_i)$. Under this assumption, we have $\hat{\boldsymbol{g}}_i^{\tau,\boldsymbol{x}}(a_i) \perp \hat{\boldsymbol{g}}_i^{\tau,\boldsymbol{x}}(a_i')$, where $\perp$ denotes that the two random variables are independent, and $\text{Var}[\hat{\boldsymbol{g}}_i^{\tau,\boldsymbol{x}}(a_i)] = \sigma$ for all $a_i, a_i' \in \mathcal{A}_i$. By the definition of variance, the variance of $\mathcal{L}_{NAL}^\tau(\boldsymbol{x})$ is

$$
\begin{aligned}
\text{Var}[\mathcal{L}_{NAL}^\tau(\boldsymbol{x})] &= \sum_{i \in \mathcal{N}} \sum_{a_i \in \mathcal{A}_i} \text{Var}[\hat{\boldsymbol{g}}_i^{\tau,\boldsymbol{x}}(a_i)\boldsymbol{x}_i(a_i)] \\
&= \sum_{i \in \mathcal{N}} \sum_{a_i \in \mathcal{A}_i} (\boldsymbol{x}_i(a_i))^2 \text{Var}[\hat{\boldsymbol{g}}_i^{\tau,\boldsymbol{x}}(a_i)] \\
&\leq |\mathcal{N}|\sigma,
\end{aligned}
$$

where the second equality is from that for a random variable $Y$ with a constant $c$, $\text{Var}[cY] = c^2\text{Var}[Y]$, and the inequality follows from the fact that $\sum_{a_i \in \mathcal{A}_i} (\boldsymbol{x}_i(a_i))^2 \leq 1$.

For the unbiased loss function defined in Eq. (7), we make similar assumptions. Specifically, let the two estimates of $\boldsymbol{F}_i^{\tau,\boldsymbol{x}}(a_i) - \overline{\boldsymbol{F}_i^{\tau,\boldsymbol{x}}}(a_i)$ be $\bar{\boldsymbol{g}}_i^{\tau,\boldsymbol{x},1}(a_i)$ and $\bar{\boldsymbol{g}}_i^{\tau,\boldsymbol{x},2}(a_i)$, we assume that each $\bar{\boldsymbol{g}}_i^{\tau,\boldsymbol{x},j}(a_i)$ is sampled independently $\forall i \in \mathcal{N}$, $a_i \in \mathcal{A}_i$, $j \in \{1,2\}$, and the variances for each estimation are less than $\sigma$. Formally, $\forall i \in \mathcal{N}$, $a_i, a_i' \in \mathcal{A}_i$, $j, j' \in \{1,2\}$, $\bar{\boldsymbol{g}}_i^{\tau,\boldsymbol{x},j}(a_i) \perp \bar{\boldsymbol{g}}_i^{\tau,\boldsymbol{x},j'}(a_i')$ and $\text{Var}[\bar{\boldsymbol{g}}_i^{\tau,\boldsymbol{x},j}(a_i)] = \sigma$. Then, the variances of the estimation for this loss function are

$$
\begin{aligned}
\text{Var}[\mathcal{L}_G^\tau(\boldsymbol{x})] &= \sum_{i \in \mathcal{N}} \sum_{a_i \in \mathcal{A}_i} \text{Var}[\bar{\boldsymbol{g}}_i^{\tau,\boldsymbol{x},1}(a_i)\bar{\boldsymbol{g}}_i^{\tau,\boldsymbol{x},2}(a_i)] \\
&\geq |\mathcal{N}|\sigma^2 \min_{i \in \mathcal{N}} |\mathcal{A}_i| + \\
&\quad 2|\mathcal{N}|\sigma \min_{i \in \mathcal{N}, a_i \in \mathcal{A}_i} \|\boldsymbol{F}_i^{\tau,\boldsymbol{x}}(a_i) - \overline{\boldsymbol{F}_i^{\tau,\boldsymbol{x}}}(a_i)\|_2^2 \min_{i \in \mathcal{N}} |\mathcal{A}_i|,
\end{aligned}
$$

---

**Algorithm 1** Learning an NE via Minimizing NAL

1: **Input:** An optimizer $\mathcal{OPT}$, the exploration ratio $\epsilon$, the uniform strategy profile $\boldsymbol{x}^u = [\boldsymbol{x}_i^u | i \in \mathcal{N}]$, the initial parameter $\boldsymbol{\theta}$, the learning rate $\eta$, the regularization scalar $\tau$, the number of total iterations $T$, the number of instances $S$ sampled at per iteration, the frequency $T_u$ of updating $\eta$ and $\tau$, the weight $\alpha$ on updating $\eta$, the weight $\beta$ on updating $\tau$, simulator $\mathcal{G}$ that returns player $i$'s payoff given a joint action.
2: **for** each $t \in [1, 2, \cdots, T]$ **do**
3:  Initialize buffer $\mathcal{M}_i \leftarrow \{\}$, $\forall i \in \mathcal{N}$
4:  $v_i \leftarrow 0$, $\forall i \in \mathcal{N}$
5:  **for** each $s \in [1, 2, \cdots, S]$ **do**
6:   $a_i \sim \boldsymbol{x}_i^{\boldsymbol{\theta}}$, $\forall i \in \mathcal{N}$
7:   $\boldsymbol{a} \leftarrow [a_i : i \in \mathcal{N}]$
8:   $a_i' \sim (1 - \epsilon)\boldsymbol{x}_i^{\boldsymbol{\theta}} + \epsilon\boldsymbol{x}_i^u$, $\forall i \in \mathcal{N}$
9:   $p_i \leftarrow (1 - \epsilon)\boldsymbol{x}_i^{\boldsymbol{\theta}}(a_i') + \epsilon\boldsymbol{x}_i^u(a_i')$, $\forall i \in \mathcal{N}$
10:   $r_i \leftarrow -\mathcal{G}(i, a_i', a_{-i}) + \tau \log \boldsymbol{x}_i^{\boldsymbol{\theta}}(a_i')$, $\forall i \in \mathcal{N}$ **// To estimate $F_i^{\tau,\boldsymbol{x}^{\boldsymbol{\theta}}}(a_i')$**
11:   $\mathcal{M}_i.\text{append}([i, a_i', r_i, p_i])$, $\forall i \in \mathcal{N}$
12:   $v_i \leftarrow v_i + r_i$
13:  **end for**
14:  $\tilde{\mathcal{L}}_{NAL}^\tau(\boldsymbol{\theta}) \leftarrow 0$
15:  $v_i \leftarrow \frac{v_i}{S}$, $\forall i \in \mathcal{N}$ **// To estimate $\langle F_i^{\tau,\boldsymbol{x}^{\boldsymbol{\theta}}}, \hat{\boldsymbol{x}}_i \rangle$**
16:  **for** each $i \in \mathcal{N}$ **do**
17:   **for** each $[i, a_i^s, r_i^s, p_i^s] \in \mathcal{M}_i$ **do**
18:    $\boldsymbol{g}_i^s \leftarrow \frac{r_i^s - v_i}{p_i^s}\boldsymbol{e}_{a_i^s}$ **// To estimate $F_i^{\tau,\boldsymbol{x}^{\boldsymbol{\theta}}} - \langle F_i^{\tau,\boldsymbol{x}^{\boldsymbol{\theta}}}, \hat{\boldsymbol{x}}_i \rangle \mathbf{1}$**
19:    $\tilde{\mathcal{L}}_{NAL}^\tau(\boldsymbol{\theta}) \leftarrow \tilde{\mathcal{L}}_{NAL}^\tau(\boldsymbol{\theta}) + \langle sg[\boldsymbol{g}_i^s], \boldsymbol{x}_i^{\boldsymbol{\theta}} \rangle$
20:   **end for**
21:  **end for**
22:  $\boldsymbol{\theta} \leftarrow \mathcal{OPT}.\text{update}(\tilde{\mathcal{L}}_{NAL}^\tau(\boldsymbol{\theta}))$
23:  **if** $t \% T_u = 0$ **then**
24:   $\eta \leftarrow \alpha\eta$, $\tau \leftarrow \beta\tau$
25:  **end if**
26: **end for**
27: **Return** $\boldsymbol{\theta}$

---

where the last inequality follows from Appendix F. Thus, the variance in estimating $\mathcal{L}_G^\tau(\boldsymbol{x})$ is $\sigma \min_{i \in \mathcal{N}} |\mathcal{A}_i|$ times larger than for NAL. Then, the variance in estimating $\mathcal{L}_G^\tau(\boldsymbol{x})$ is expected to be substantially higher than that of NAL.

### 4.3. Minimizing NAL under the sampled play

We now detail our algorithm that learns an NE by minimizing NAL. The pseudocode is in Algorithm 1. Specifically, consider an approximation function $\Pi(\cdot)$ parameterized by $\boldsymbol{\theta}$, where the resulting strategy profile is denoted as $\boldsymbol{x}^{\boldsymbol{\theta}} = \Pi(\boldsymbol{\theta})$. Our goal is to minimize the following loss function $\mathcal{L}_{NAL}^\tau(\boldsymbol{\theta})$ through a two-step process: **sampling** and **updating**.

$$
\mathcal{L}_{NAL}^\tau(\boldsymbol{\theta}) = \sum_{i \in \mathcal{N}} \langle sg[\boldsymbol{F}_i^{\tau,\boldsymbol{x}^{\boldsymbol{\theta}}} - \langle \boldsymbol{F}_i^{\tau,\boldsymbol{x}^{\boldsymbol{\theta}}}, \hat{\boldsymbol{x}}_i \rangle \mathbf{1}], \boldsymbol{x}_i^{\boldsymbol{\theta}} \rangle.
$$

**Sampling.** The sampling process is outlined from lines 3 to 13 in Algorithm 1. At each iteration $t$, we begin by initializing the buffer $\mathcal{M}_i = \{\}$ and the random variable $v_i$ for

each player $i$ (lines 3 and 4 of Algorithm 1). The random variable $v_i$ is used to estimate the value of $-\langle \boldsymbol{F}_i^{\tau,\boldsymbol{x}^{\boldsymbol{\theta}}}, \hat{\boldsymbol{x}}_i \rangle$. Next, for each player $i$, $S$ instances are sampled. In each instance, an action $a_i$ is selected for each player $i$ according to the strategy profile $\boldsymbol{x}^{\boldsymbol{\theta}}$ (line 6 of Algorithm 1), resulting in the action profile $\boldsymbol{a} = [a_i : i \in \mathcal{N}]$ (line 7 of Algorithm 1). Each $\boldsymbol{a}_{-i} = [a_j : j \in \mathcal{N}, j \neq i]$ serves as the environmental dynamic for player $i$, enabling the estimation of $\mathcal{L}_{NAL}^{\tau}(\boldsymbol{\theta})$. Subsequently, based on the exploration parameter $\epsilon$, the uniform strategy profile $\boldsymbol{x}^u$ ($\boldsymbol{x}_i^u = \boldsymbol{1}/|\mathcal{A}_i|$), and the current strategy profile $\boldsymbol{x}^{\boldsymbol{\theta}}$, an alternative action $a_i'$ is sampled for each player $i$ according to the strategy $\hat{\boldsymbol{x}}_i = (1-\epsilon)\boldsymbol{x}_i^{\boldsymbol{\theta}} + \epsilon \boldsymbol{x}_i^u$ (line 8 of Algorithm 1). The exploration parameter $\epsilon$ and the uniform strategy $\boldsymbol{x}^u$ ensure that the probability of selecting any action $a$ within the strategy $\hat{\boldsymbol{x}}_i$ is not too small, which guarantees the variance of estimating via importance sampling is not too large. The probability of selecting action $a_i'$ through $\hat{\boldsymbol{x}}_i$ is denoted by $p_i$ (line 9 of Algorithm 1). The unbiased estimation of $\boldsymbol{F}_i^{\tau,\boldsymbol{x}^{\boldsymbol{\theta}}}(a_i')$ is then computed as $r_i \leftarrow -\mathcal{G}(i, a_i', \boldsymbol{a}_{-i}) + \tau \log \boldsymbol{x}_i^{\boldsymbol{\theta}}(a_i')$, $\forall i \in \mathcal{N}$ (line 10 of Algorithm 1), where $\mathcal{G}$ represents the simulator returning player $i$'s payoff for the joint action $[a_i', \boldsymbol{a}_{-i}]$. Specifically,

$$
\begin{aligned}
\mathbb{E}[r_i] =& \mathbb{E}[-\mathcal{G}(i, a_i', \boldsymbol{a}_{-i}) + \tau \log \boldsymbol{x}_i^{\boldsymbol{\theta}}(a_i')] \\
=& \mathbb{E}[-\mathcal{G}(i, a_i', \boldsymbol{a}_{-i})] + \tau \log \boldsymbol{x}_i^{\boldsymbol{\theta}}(a_i') \\
=& \boldsymbol{F}_i^{\boldsymbol{x}^{\boldsymbol{\theta}}}(a_i') + \tau \log \boldsymbol{x}_i^{\boldsymbol{\theta}}(a_i') = \boldsymbol{F}_i^{\tau,\boldsymbol{x}^{\boldsymbol{\theta}}}(a_i'),
\end{aligned} \tag{3}
$$

where the third line follows from the fact that $\boldsymbol{a}_{-i}$ is sampled according to $\boldsymbol{x}_{-i}^{\boldsymbol{\theta}}$. Finally, the tuple $[i, a_i', r_i, p_i]$ is stored in the buffer $\mathcal{M}_i$ (line 11 of Algorithm 1), and $v_i$ is updated as $v_i \leftarrow v_i + r_i$ (line 12 of Algorithm 1).

**Updating.** The updating procedure is outlined from lines 14 to 25 in Algorithm 1. We first initialize the estimator for $\mathcal{L}_{NAL}^{\tau}(\boldsymbol{\theta})$ as $\tilde{\mathcal{L}}_{NAL}^{\tau}(\boldsymbol{\theta}) \leftarrow 0$ and normalize $v_i$ by setting $v_i \leftarrow \frac{v_i}{S}$ (lines 14 and 15 of Algorithm 1). The expectation $\mathbb{E}[v_i]$ corresponds to $\langle \boldsymbol{F}_i^{\tau,\boldsymbol{x}^{\boldsymbol{\theta}}}, \hat{\boldsymbol{x}}_i \rangle$. Formally,

$$
\begin{aligned}
\mathbb{E}[v_i] = \mathbb{E}\left[\frac{1}{S}\sum_{s=1}^{S} r_i^s\right] =& \mathbb{E}\left[\frac{1}{S}\sum_{s=1}^{S} \boldsymbol{F}_i^{\tau,\boldsymbol{x}^{\boldsymbol{\theta}}}(a_i^s)\right] \\
=& \mathbb{E}_{a_i^s \sim \hat{\boldsymbol{x}}_i}\left[\boldsymbol{F}_i^{\tau,\boldsymbol{x}^{\boldsymbol{\theta}}}(a_i^s)\right] \\
=& \langle \boldsymbol{F}_i^{\tau,\boldsymbol{x}^{\boldsymbol{\theta}}}, \hat{\boldsymbol{x}}_i \rangle,
\end{aligned} \tag{4}
$$

where $a_i^s$ and $r_i^s$ come from the $s$-th tuple $[i, a_i^s, r_i^s, p_i^s]$ stored in buffer $\mathcal{M}_i$, the second equality is from $\mathbb{E}[r_i^s] = \boldsymbol{F}_i^{\tau,\boldsymbol{x}^{\boldsymbol{\theta}}}(a_i^s)$ (Eq. (3)), and the third equality is from that $a_i^s$ is sampled via $\hat{\boldsymbol{x}}_i$. Additionally, we use the tuples in $\mathcal{M}_i$ (line 17 of Algorithm 1) to estimate $\boldsymbol{F}_i^{\tau,\boldsymbol{x}^{\boldsymbol{\theta}}} - \langle \boldsymbol{F}_i^{\tau,\boldsymbol{x}^{\boldsymbol{\theta}}}, \hat{\boldsymbol{x}}_i \rangle \boldsymbol{1}$ through the computation $\boldsymbol{g}_i^s \leftarrow \frac{r_i^s - v_i}{p_i^s} \boldsymbol{e}_{a_i^s}$ (line 18 of Algorithm 1), where $\boldsymbol{e}_{a_i^s}$ is a vector whose the coordinate $a_i^s$ is

1 and all other coordinates are 0. It is straightforward to verify that $\mathbb{E}[\boldsymbol{g}_i^s] = \boldsymbol{F}_i^{\tau,\boldsymbol{x}^{\boldsymbol{\theta}}} - \langle \boldsymbol{F}_i^{\tau,\boldsymbol{x}^{\boldsymbol{\theta}}}, \hat{\boldsymbol{x}}_i \rangle \boldsymbol{1}$. Formally,

$$
\begin{aligned}
\mathbb{E}[\boldsymbol{g}_i^s] =& \mathbb{E}_{s \sim \hat{\boldsymbol{x}}_i}\left[\frac{r_i^s - v_i}{p_i^s} \boldsymbol{e}_{a_i^s}\right] \\
=& \mathbb{E}_{s \sim \hat{\boldsymbol{x}}_i}\left[\frac{\boldsymbol{F}_i^{\tau,\boldsymbol{x}^{\boldsymbol{\theta}}}(a_i^s) - \langle \boldsymbol{F}_i^{\tau,\boldsymbol{x}^{\boldsymbol{\theta}}}, \hat{\boldsymbol{x}}_i \rangle}{p_i^s} \boldsymbol{e}_{a_i^s}\right],
\end{aligned} \tag{5}
$$

where the second equality is from $\mathbb{E}[r_i^s] = \boldsymbol{F}_i^{\tau,\boldsymbol{x}^{\boldsymbol{\theta}}}(a_i^s)$ (Eq. (3)), $\mathbb{E}[v_i] = \langle \boldsymbol{F}_i^{\tau,\boldsymbol{x}^{\boldsymbol{\theta}}}, \hat{\boldsymbol{x}}_i \rangle$ (Eq. (4)), and $(r_i^s - v_i) \perp p_i^s$ (as $p_i^s$ is given and not sampled, which can be seen as a constant). As the rightest side of Eq. (5) a standard importance sampling process, it follows from the properties of importance sampling that

$$
\begin{aligned}
\mathbb{E}[\boldsymbol{g}_i^s] =& \mathbb{E}_{s \sim \hat{\boldsymbol{x}}_i}\left[\frac{\boldsymbol{F}_i^{\tau,\boldsymbol{x}^{\boldsymbol{\theta}}}(a_i^s) - \langle \boldsymbol{F}_i^{\tau,\boldsymbol{x}^{\boldsymbol{\theta}}}, \hat{\boldsymbol{x}}_i \rangle}{p_i^s} \boldsymbol{e}_{a_i^s}\right] \\
=& \boldsymbol{F}_i^{\tau,\boldsymbol{x}^{\boldsymbol{\theta}}} - \langle \boldsymbol{F}_i^{\tau,\boldsymbol{x}^{\boldsymbol{\theta}}}, \hat{\boldsymbol{x}}_i \rangle \boldsymbol{1}.
\end{aligned}
$$

The estimator $\tilde{\mathcal{L}}_{NAL}^{\tau}(\boldsymbol{\theta})$ is updated via $\tilde{\mathcal{L}}_{NAL}^{\tau}(\boldsymbol{\theta}) \leftarrow \tilde{\mathcal{L}}_{NAL}^{\tau}(\boldsymbol{\theta}) + \langle \boldsymbol{g}_i^s, \boldsymbol{x}_i^{\boldsymbol{\theta}} \rangle$ (line 19 of Algorithm 1). Since $\mathbb{E}[\boldsymbol{g}_i^s] = \boldsymbol{F}_i^{\tau,\boldsymbol{x}^{\boldsymbol{\theta}}} - \langle \boldsymbol{F}_i^{\tau,\boldsymbol{x}^{\boldsymbol{\theta}}}, \hat{\boldsymbol{x}}_i \rangle \boldsymbol{1}$ and $\boldsymbol{x}_i^{\boldsymbol{\theta}}$ is known, it follows that $\frac{1}{S}\mathbb{E}[\tilde{\mathcal{L}}_{NAL}^{\tau}(\boldsymbol{\theta})] = \mathcal{L}_{NAL}^{\tau}(\boldsymbol{\theta})$. Therefore, $\tilde{\mathcal{L}}_{NAL}^{\tau}(\boldsymbol{\theta})$ provides an unbiased estimate of $\mathcal{L}_{NAL}^{\tau}(\boldsymbol{\theta})$. The estimator $\tilde{\mathcal{L}}_{NAL}^{\tau}(\boldsymbol{\theta})$ is then passed to the optimizer $\mathcal{OPT}$ for updating $\boldsymbol{\theta}$ (line 22 of Algorithm 1). If $t\%T_u = 0$ (line 23 of Algorithm 1), the parameters $\eta$ and $\tau$ are updated as $\eta \leftarrow \alpha\eta$ and $\tau \leftarrow \beta\tau$, where $0 < \alpha, \beta < 1$ (line 24 of Algorithm 1). These adjustments ensure that an NE of the regularization game approaches an NE of the original game. Specifically, as shown in Theorem 4.3, decreasing $\tau$ brings the NE of the regularization game, defined by the utility function $u_i^{\tau}(\boldsymbol{x}) = u_i(\boldsymbol{x}) - \tau \boldsymbol{x}_i^{\mathrm{T}} \log \boldsymbol{x}_i$, closer to that of the original game. Furthermore, reducing $\eta$ stabilizes the algorithm as we find that without a corresponding reduction in $\eta$, decreasing $\tau$ could destabilize the learning process.

We do not provide the convergence for our algorithm as this convergence depends on the stochastic optimization technique used, which is not the focus of this work. Theoretically learning the zero point of the first-order gradient via non-convex stochastic optimization techniques is an urgent problem to be solved. Solving this problem falls under the research direction of optimization rather than game theory.

## 5. Experiments

**Configurations.** We compare our algorithm with algorithms that minimize the loss function in Gemp et al. (2024), ADI (Gemp et al., 2022), or NashApr (Duan et al., 2023), respectively. Our loss function and the loss function in Gemp et al. (2024) are unbiased loss functions, while others are biased loss functions. The implementation details of the

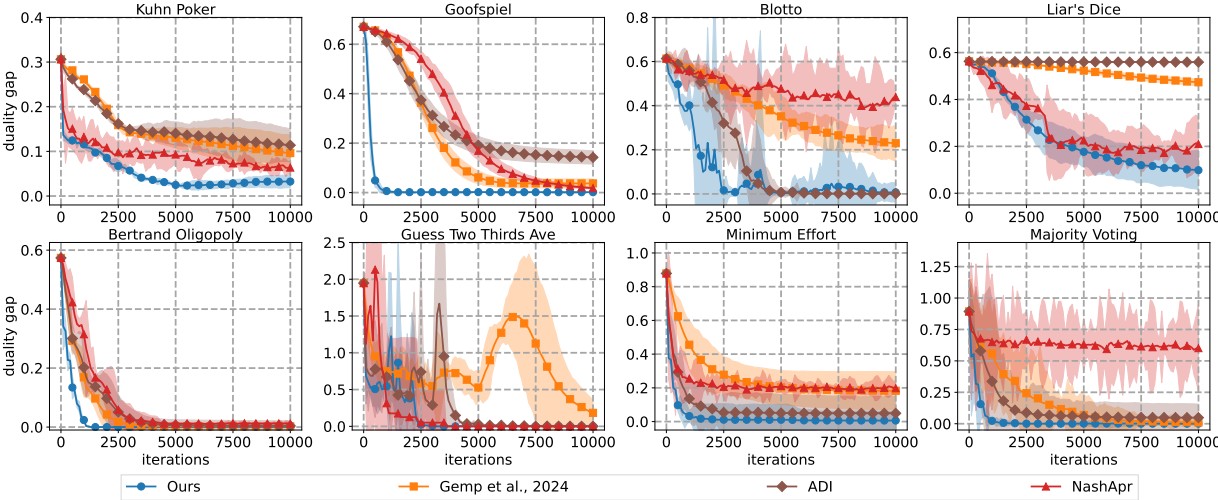

Figure 1. Empirical convergence rates of tested algorithms when the optimizer is Adam. The top row shows the following scenarios from left to right: 2 players with 64 actions, 2 players with 384 actions, 4 players with 66 actions, and 2 players with 2304 actions. The bottom row displays, from left to right: 4 players with 50 actions, 4 players with 50 actions, 5 players with 30 actions, and 11 players with 5 actions. The shaded regions represent one standard deviation of the results, calculated across four different random seeds.

compared loss functions are in Appendix G. Notably, the implementation of all tested loss functions includes the stop-gradient operator. We conduct experiments on eight NFGs from OpenSpiel (Lanctot et al., 2019) and GAMUT (Nudelman et al., 2004), specifically Kuhn Poker, Goofspiel, Blotto, Liars Dice, Bertrand Oligopoly, Guess Two Thirds Ave, Minimum Effort, and Majority Voting. The former four games are sourced from OpenSpiel, while other games are implemented by GAMUT. The payoff matrix components of each game are normalized to a range between 0 and 1. All experiments are performed on a machine equipped with four RTX 3060 GPUs and 376 GB of memory.

We use a DNN parameterized by $\boldsymbol{\theta}$ to represent strategy profiles due to the strong expressive power of DNNs, as did in previous works (Goktas et al., 2022; Marris et al., 2022; Liu et al., 2024) (see discussions in Appendix B). The network used in this paper is a three-layer MLP, with 1024 neurons in both the input and hidden layers, and $|\mathcal{N}|$ heads in the output layer, each corresponding to the action space of a player. For all games, the input is a 1024-dimensional vector with all coordinates set to 1. ReLU activation is used in the hidden layers (Krizhevsky et al., 2012), and Softmax activation is applied in the output layer (Dempster et al., 1977) (unless otherwise stated), ensuring the output remains within the simplex. Neural networks and optimizers (stochastic optimization techniques) are implemented using PyTorch (Paszke et al., 2019), with Adam (Kingma & Ba, 2014) as the optimizer (unless otherwise stated), due to its widespread use in training modern neural networks, including GANs (Goodfellow et al., 2014), BERT (Devlin, 2018), GPT (Brown, 2020), and ViT (Dosovitskiy, 2020). For all

tested algorithms, $\epsilon$ is fixed at 1, $T$ at 10,000, and $S$ at 10 across all games (unless otherwise stated). We perform an extensive hyperparameter search, varying the learning rate $\eta \in \{0.0001, 0.00001\}$, regularization scalar $\tau \in \{0.1, 1\}$, update frequency $T_u \in \{200, 500, 1000\}$, and momentum coefficients $\alpha, \beta \in \{0.9, 0.5\}$. The selected hyperparameters are listed in Appendix I.

**Results on convergence rates and variances.** We run each algorithm four times with different random seeds. The results, including convergence rates and variances, are presented in Figures 1 and 2, respectively. Our algorithm achieves the fastest empirical convergence rate and the lowest variance. Specifically, we find that, in Goofspiel and Minimum Effort, the algorithm minimizing the existing unbiased loss function in Gemp et al. (2024) fails to converge to an NE. In contrast, the algorithm minimizing NAL is able to converge to an NE. Additionally, algorithms based on biased loss functions occasionally fail to converge. For example, the algorithm minimizing ADI does not converge in Blotto, and the algorithm minimizing NashApr fails in Liars Dice. In addition, the variance in estimating NAL decreases by at least two orders of magnitude for all tested games compared to using the existing unbiased loss function, and in Liars Dice, this variance reduction reaches up to six orders of magnitude. We also find a strong correlation between variance and convergence performance. In Bertrand Oligopoly, where the algorithm minimizing the existing unbiased loss function in Gemp et al. (2024) performs closest to ours, it is the only case where this algorithm's variance in estimating the value of the loss function is lower than that of our algorithm. However, due to the extremely

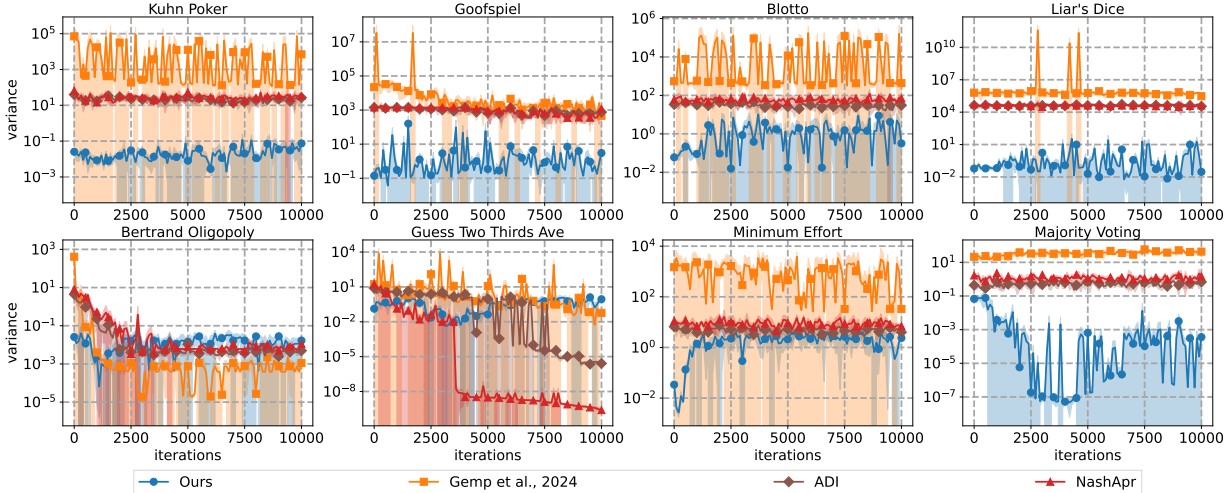

*Figure 2.* Variances observed in estimating the value of loss functions used by different algorithms when the optimizer is Adam.

high variance early on, this algorithm's convergence rate remains slower than ours. Although the algorithm minimizing NAL does not appear to converge to an exact NE in Kuhn Poker and Liars Dice, this is primarily due to the value of $S$ being insufficiently large. As shown in Appendix J, when $S$ is increased, our algorithms can also converge to a more and more accurate NE in both Kuhn Poker and Liars Dice.

**Results on differences between estimated and true loss values.** To determine whether NAL is an unbiased loss function, we compare the differences between the estimated and true loss values across the various algorithms, as shown in Figure 3. Empirical results confirm that NAL behaves as an unbiased loss function, exhibiting significantly smaller differences between true and estimated values compared to other loss functions. More precisely, the difference between the estimated and true values for NAL is usually two orders of magnitude smaller compared to that of other tested loss functions. As the difference between the true value and the estimated value of NAL is considerably smaller than that of other loss functions, we present a more detailed graph highlighting this difference for NAL in Appendix J. In addition, the estimated value of NAL is also in Appendix J.

**Results on convergence rates, variances, and differences with different optimizers.** We further assess the robustness of our loss function with different optimizers by evaluating performance using other famous optimizers, such as RMSprop (Bottou, 2010) and SGD (Robbins & Monro, 1951), with the parameter fine tuned in the scenario where Adam is used. Key metrics, such as convergence rate, the variance of estimating the value loss function, and the difference between the estimated value and true value of loss functions, are analyzed. The results on convergence rates, variances, and differences when using RMSprop or SGD as the opti-

mizer are in Appendix J. Consistent with the results using Adam, our algorithm exhibits the fastest convergence rates, lowest variance, and smallest difference.

**Results on sampling times and convergence rates with different sampling methods.** We also present experimental results for algorithms that employ the sampling method from (Gemp et al., 2022) and Gemp et al. (2024), as described in Appendix J. Specifically, we compare sampling times between the method in (Gemp et al., 2022) and Gemp et al. (2024) with the method in Algorithm 1, and evaluate the convergence rates of algorithms employing the sampling method used in (Gemp et al., 2022) and Gemp et al. (2024) as well as the sampling method in Algorithm 1, respectively. Experimental results show that both the sampling method in Algorithm 1 and our loss function, NAL, significantly enhance the convergence rate.

**Results on convergence rates with different neural network structures.** Additionally, we evaluate the performance of different algorithms under various network architectures. Specifically, we replace Softmax with Sparsemax (Martins & Astudillo, 2016). The experimental results are shown in Appendix J. We observe that our algorithm still exhibits the fastest convergence rate. In fact, the convergence rate of our algorithm remains largely unchanged. In contrast, the convergence rates of the other algorithms experience significant degradation.

**Results on convergence rates of NAL with or without the term $\langle F_i^{\tau, x}, \hat{x}_i \rangle \mathbf{1}$.** Moreover, we investigate the impact of the term $\langle F_i^{\tau, x^\theta}, \hat{x}_i \rangle \mathbf{1}$ in NAL. Our results show that this term not only reduces the variance but also ensures that the algorithm minimizing NAL converges to an NE. In the absence of this term, the algorithm may fail to learn an NE. Further details are provided in Appendix J.

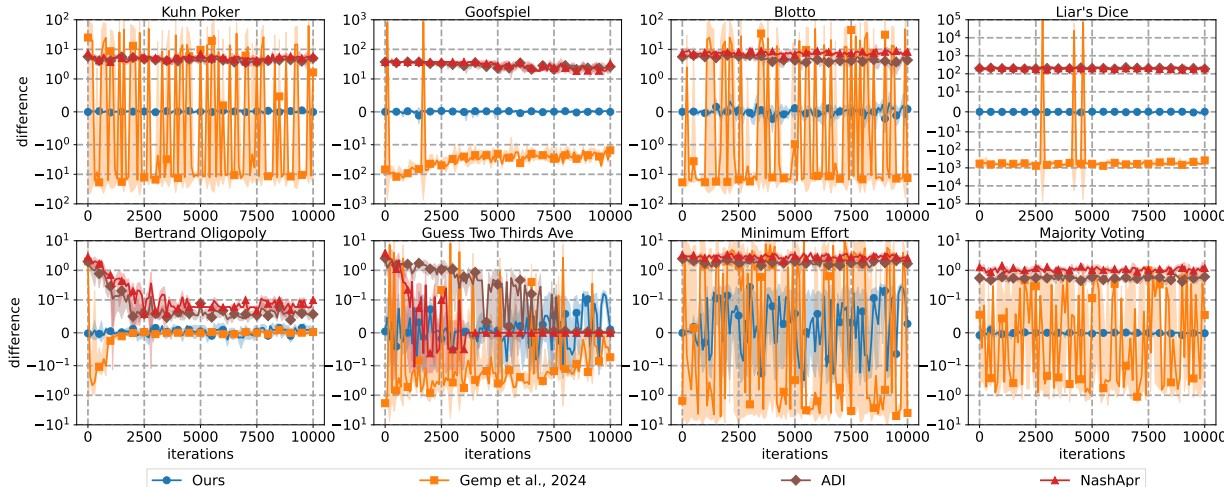

*Figure 3.* Difference between the true value and the estimated value of loss functions when the optimizer is Adam. Since the difference between the true value and the estimated value of our loss function NAL is considerably smaller than that of other loss functions, we present a more detailed graph highlighting this difference for NAL in Appendix J.

**Results on convergence rates of NAL with different values of $\epsilon$.** To strengthen the robustness of our results, we also include experiments with various $\epsilon$ values (0, 0.1, 0.5, and 0.9), as shown in Appendix J. Across all tested $\epsilon$ values, our algorithm consistently outperforms the baselines, further validating that the variance reduction achieved by our loss function leads to an accelerated convergence rate.

**Results on convergence rates when the strategy is represented using a real vector.** As mentioned in Appendix B, we employ a DNN due to its capability to approximate arbitrary non-linear functions, enabling the discovery of complex equilibrium strategies that simpler representations may overlook. In contrast, a real vector lacks this expressive power. The results, where the strategy is represented using a real vector, are shown in Appendix J. All algorithms exhibit varying degrees of performance degradation, yet our algorithm still outperforms the others.

## 6. Conclusions

We introduce a novel surrogate loss function for using non-convex stochastic optimization techniques of ML to compute an NE, named NAL. It can be estimated without bias and will incur an significantly lower variance than the existing unbiased loss function. Experimental results show that the algorithm minimizing NAL significantly outperforms other tested algorithms. Our approach offers a promising new direction for computing an NE, with the potential to address the challenges posed by large-scale games. One direction of our future works is to extend our approach to solve imperfect information extensive-form games.

## Acknowledgements

This work is supported in part by the National Natural Science Foundation of China under Grant 62192783, the Jiangsu Science and Technology Major Project BG2024031, the Fundamental Research Funds for the Central Universities (14380128), the Collaborative Innovation Center of Novel Software Technology and Industrialization, and the InnoHK funding.

## Impact Statement

This paper presents work whose goal is to advance the field of Machine Learning. There are many potential societal consequences of our work, none which we feel must be specifically highlighted here.

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

# A. Table of Notations and Definitions

Table 1: Table of Notations and Definitions

| Notation/Definition | Section | Description |
|---|---|---|
| $\mathcal{N}$ | 3 | the set of players |
| $i$ | 3 | the index of the player |
| $-i$ | 3 | all players except player $i$ |
| $\mathcal{A}_i$ | 3 | the action set of player $i$ |
| $a_i$ | 3 | an action of player $i$, $e.g.$, $a_i \in \mathcal{A}_i$ |
| $a_{-i}$ | 3 | the actions selected by the players except player $i$ |
| $u_i(a_i, a_{-i})$ | 3 | the utility received by player $i$ if player $i$ select $a_i$, while other players select $a_{-i}$ |
| $\boldsymbol{x}_i$ | 3 | the strategy of player $i$ |
| $\boldsymbol{\mathcal{X}}_i$ | 3 | the set of the strategies of player $i$, a $(|\mathcal{A}_i| - 1)$-dimensional simplex |
| $\boldsymbol{x}$ | 3 | $\{\boldsymbol{x}_i \in \boldsymbol{\mathcal{X}}_i \mid i \in \mathcal{N}\}$; the strategy profile |
| $\boldsymbol{\mathcal{X}}$ | 3 | $\times_{i \in \mathcal{N}} \boldsymbol{\mathcal{X}}_i$; the strategy space of all players |
| $\boldsymbol{\mathcal{X}}^{\circ}$ | 3 | the interior of $\boldsymbol{\mathcal{X}}$, $e.g.$, for each $\boldsymbol{x} \in \boldsymbol{\mathcal{X}}^{\circ}$, $\boldsymbol{x}_i(a_i) > 0, \forall i \in \mathcal{N}$ and $a_i \in \mathcal{A}_i$ |
| $u_i(\boldsymbol{x}_i, \boldsymbol{x}_{-i})$ | 3 | $\sum_{\boldsymbol{a} \in \times_{i \in \mathcal{N}} \mathcal{A}_i} u_i(\boldsymbol{a}) \prod_{j \in \mathcal{N}} \boldsymbol{x}_j(a_j)$, where $a_j \in \mathcal{A}_j$ denotes player $j$'s component of the joint action $\boldsymbol{a}$; the utility received by player $i$, given that all players follow the strategy profile $\boldsymbol{x} \in \boldsymbol{\mathcal{X}}$ |
| $\nabla_{\boldsymbol{x}_i} u_i(\boldsymbol{x})$ | 3 | the first-order gradient of $u_i(\boldsymbol{x})$ w.r.t. $\boldsymbol{x}_i$ |
| $\boldsymbol{\mathcal{X}}^*$ | 3 | the set of NEs in the game with the utility function $u_i(\boldsymbol{x})$ |
| $\boldsymbol{\mathcal{X}}^{*,\circ}$ | 3 | the set of interior NEs in the game with the utility function $u_i(\boldsymbol{x})$ such that $\forall \boldsymbol{x}^* \in \boldsymbol{\mathcal{X}}^{*,\circ}$, $\boldsymbol{x}_i^*(a_i) > 0, \forall i \in \mathcal{N},\ a_i \in \mathcal{A}_i$ |
| $\mathrm{dg}(\boldsymbol{x})$ | 3 | $\sum_{i \in \mathcal{N}} \max_{\boldsymbol{x}_i' \in \boldsymbol{\mathcal{X}}_i} \langle \nabla_{\boldsymbol{x}_i} u_i(\boldsymbol{x}), \boldsymbol{x}_i' - \boldsymbol{x}_i \rangle$; the duality gap in the game with the utility function $u_i(\boldsymbol{x})$, a well-known metric to measure the distance from the strategy profile $\boldsymbol{x}$ to NE; if and only if $\mathrm{dg}(\boldsymbol{x}) = 0$, $\boldsymbol{x} \in \boldsymbol{\mathcal{X}}^*$; if $\mathrm{dg}(\boldsymbol{x}) = \delta$, then $\boldsymbol{x}$ is a $\delta$-NE |
| $\exp(\boldsymbol{x})$ | 3 | $\frac{\sum_{i \in \mathcal{N}} (\max_{\boldsymbol{x}_i'} u_i(\boldsymbol{x}_i', \boldsymbol{x}_{-i}) - u_i(\boldsymbol{x}_i, \boldsymbol{x}_{-i}))}{|\mathcal{N}|}$; the exploitability in the game with the utility function $u_i(\boldsymbol{x})$, a well-known metric to measure the distance from the strategy profile $\boldsymbol{x}$ to NE |
| $u_i^{\tau}(\boldsymbol{x})$ | 3 | $u_i(\boldsymbol{x}) - \tau \boldsymbol{x}_i^{\mathrm{T}} \log \boldsymbol{x}_i$; the utility function of the regularization game defined in Gemp et al. (2024) |

| Notation/Definition | Section | Description |
|---|---|---|
| $\boldsymbol{F}_i^{\tau,\boldsymbol{x}}$ | 3 | $-\nabla_{\boldsymbol{x}_i} u_i^\tau(\boldsymbol{x})$, which can also be represented by $-\nabla_{\boldsymbol{x}_i} u_i(\boldsymbol{x}) + \tau \log \boldsymbol{x}_i$ |
| $\overline{\boldsymbol{F}_i^{\tau,\boldsymbol{x}}}$ | 3 | $\frac{\sum_{a_i \in \mathcal{A}_i} \boldsymbol{F}_i^{\tau,\boldsymbol{x}}(a_i)}{\lvert\mathcal{A}_i\rvert}\mathbf{1}$ |
| $\mathcal{L}_G^\tau(\boldsymbol{x})$ | 3 | $\sum_{i \in \mathcal{N}} \lVert \boldsymbol{F}_i^{\tau,\boldsymbol{x}} - \overline{\boldsymbol{F}_i^{\tau,\boldsymbol{x}}} \rVert_2^2$; 
 the loss function proposed by Gemp et al. (2024) |
| $sg[\cdot]$ | 4 | the stop-gradient operator such that 
 for any variable $\boldsymbol{b} \in \mathbb{R}^d$, $sg[\boldsymbol{b}] = \boldsymbol{b}$ while $\nabla_{\boldsymbol{b}} sg[\boldsymbol{b}] = \mathbf{0} \in \mathbb{R}^{d \times d}$ |
| $\mathcal{L}_{NAL}^\tau(\boldsymbol{x})$ | 4 | $\sum_{i \in \mathcal{N}} \langle sg[\boldsymbol{F}_i^{\tau,\boldsymbol{x}} - \langle \boldsymbol{F}_i^{\tau,\boldsymbol{x}}, \hat{\boldsymbol{x}}_i \rangle \mathbf{1}], \boldsymbol{x}_i \rangle$, 
 where $\hat{\boldsymbol{x}} = [\hat{\boldsymbol{x}}_0, \hat{\boldsymbol{x}}_1, \cdots, \hat{\boldsymbol{x}}_{\lvert\mathcal{N}\rvert-1}]$ can be any strategy profile in $\boldsymbol{\mathcal{X}}$; 
 NAL, the unbiased loss function proposed by us |
| $\nabla_{\boldsymbol{x}_i} \mathcal{L}_{NAL}^\tau(\boldsymbol{x})$ | 4 | $sg[\boldsymbol{F}_i^{\tau,\boldsymbol{x}} - \langle \boldsymbol{F}_i^{\tau,\boldsymbol{x}}, \hat{\boldsymbol{x}}_i \rangle \mathbf{1}]$; 
 the first-order gradient of $\mathcal{L}_{NAL}^\tau(\boldsymbol{x})$ w.r.t. $\boldsymbol{x}_i$ |
| $\nabla_{\boldsymbol{x}} \mathcal{L}_{NAL}^\tau(\boldsymbol{x})$ | 4 | $[sg[\boldsymbol{F}_i^{\tau,\boldsymbol{x}} - \langle \boldsymbol{F}_i^{\tau,\boldsymbol{x}}, \hat{\boldsymbol{x}}_i \rangle \mathbf{1}] \mid i \in \mathcal{N}]$; 
 the first-order gradient of $\mathcal{L}_{NAL}^\tau(\boldsymbol{x})$ w.r.t. $\boldsymbol{x}$ |
| $\mathrm{dg}^\tau(\boldsymbol{x})$ | 4 | $\sum_{i \in \mathcal{N}} \max_{\boldsymbol{x}_i' \in \boldsymbol{\mathcal{X}}_i} \langle \nabla_{\boldsymbol{x}_i} u_i^\tau(\boldsymbol{x}), \boldsymbol{x}_i' - \boldsymbol{x}_i \rangle$; 
 the duality gap in the game with the utility function $u_i^\tau(\boldsymbol{x})$ |
| $\exp^\tau(\boldsymbol{x})$ | 4 | $\frac{\sum_{i \in \mathcal{N}} (\max_{\boldsymbol{x}_i'} u_i^\tau(\boldsymbol{x}_i', \boldsymbol{x}_{-i}) - u_i^\tau(\boldsymbol{x}_i, \boldsymbol{x}_{-i}))}{\lvert\mathcal{N}\rvert}$; 
 the exploitability in the game with the utility function $u_i^\tau(\boldsymbol{x})$ |
| $\perp$ | 4 | this notation represents that two random variables are independent, 
 *e.g.*, for random variables $b$ and $c$, $b \perp c$ implies 
 that $b$ and $c$ are independent |
| $\mathrm{Var}[\cdot]$ | 4 | the variance |
| $\hat{\boldsymbol{g}}_i^{\tau,\boldsymbol{x}}(a_i)$ | 4 | the estimation of $\boldsymbol{F}_i^{\tau,\boldsymbol{x}} - \langle \boldsymbol{F}_i^{\tau,\boldsymbol{x}}, \hat{\boldsymbol{x}}_i \rangle \mathbf{1}$ at action $a_i \in \mathcal{A}_i$ in NAL |
| $\bar{\boldsymbol{g}}_i^{\tau,\boldsymbol{x},j}(a_i)$ | 4 | the $j$-th estimation of $\boldsymbol{F}_i^{\tau,\boldsymbol{x}} - \overline{\boldsymbol{F}_i^{\tau,\boldsymbol{x}}}$ at action $a_i \in \mathcal{A}_i$ 
 in the loss function proposed by Gemp et al. (2024), where $j \in \{1,2\}$ |
| $\sigma$ | 4 | in Section 4.2, we assume that $\mathrm{Var}[\hat{\boldsymbol{g}}_i^{\tau,\boldsymbol{x}}(a_i)] = \sigma$ 
 and $\mathrm{Var}[\bar{\boldsymbol{g}}_i^{\tau,\boldsymbol{x},j}(a_i)] = \sigma, \forall i \in \mathcal{N}, \ j \in \{1,2\}$, and $a_i \in \mathcal{A}_i$ |
| $\boldsymbol{x}^{\boldsymbol{\theta}}$ | 4 | the strategy profile represented by an approximation 
 function $\Pi(\cdot)$ parameterized by $\boldsymbol{\theta}$ |
| $\mathcal{L}_{NAL}^\tau(\boldsymbol{\theta})$ | 4 | $\sum_{i \in \mathcal{N}} \langle sg[\boldsymbol{F}_i^{\tau,\boldsymbol{x}^{\boldsymbol{\theta}}} - \langle \boldsymbol{F}_i^{\tau,\boldsymbol{x}^{\boldsymbol{\theta}}}, \hat{\boldsymbol{x}}_i \rangle \mathbf{1}], \boldsymbol{x}_i^{\boldsymbol{\theta}} \rangle$; 
 the NAL related to the parameter $\boldsymbol{\theta}$ |
| $\mathcal{OPT}$ | 4 | the optimizer in Algorithm 1 |
| $\epsilon$ | 4 | the exploration ratio in Algorithm 1 |
| $\boldsymbol{x}_i^u$ | 4 | $\frac{\mathbf{1}}{\lvert\mathcal{A}_i\rvert}$; the uniform strategy of player $i$ |
| $\boldsymbol{x}^u$ | 4 | $[\boldsymbol{x}_i^u \mid i \in \mathcal{N}]$; the uniform strategy profile |
| $\eta$ | 4 | the learning rate in Algorithm 1 |
| $T$ | 4 | the number of total iterations in Algorithm 1 |

*Continued on next page*

| Notation/Definition | Section | Description |
|---|---|---|
| $S$ | 4 | the number of instances sampled at per iteration in Algorithm 1 |
| $T_u$ | 4 | the frequency of updating $\eta$ and $\tau$ in Algorithm 1 |
| $\alpha$ | 4 | the weight on updating $\eta$ in Algorithm 1, *e.g.*, $\eta \leftarrow \alpha\eta$ |
| $\beta$ | 4 | the weight on updating $\tau$ in Algorithm 1, *e.g.*, $\tau \leftarrow \beta\tau$ |
| $\mathcal{G}$ | 4 | the simulator in Algorithm 1 that
returns player $i$'s payoff given a joint action |
| $\mathcal{M}_i$ | 4 | the buffer in Algorithm 1 |
| $t$ | 4 | the current iteration in Algorithm 1 |
| $\boldsymbol{a}$ | 4 | $[a_i : i \in \mathcal{N}]$; the joint action in Algorithm 1 |
| $a_i'$ | 4 | sampled from $(1-\epsilon)\boldsymbol{x}_i^{\boldsymbol{\theta}} + \epsilon\boldsymbol{x}_i^u$,
used in Algorithm 1 for estimating $\boldsymbol{F}_i^{\tau,\boldsymbol{x}^{\boldsymbol{\theta}}} - \langle \boldsymbol{F}_i^{\tau,\boldsymbol{x}^{\boldsymbol{\theta}}}, \hat{\boldsymbol{x}}_i\rangle\mathbf{1}$ |
| $p_i$ | 4 | $(1-\epsilon)\boldsymbol{x}_i^{\boldsymbol{\theta}}(a_i') + \epsilon\boldsymbol{x}_i^u(a_i')$;
used in Algorithm 1 for estimating $\boldsymbol{F}_i^{\tau,\boldsymbol{x}^{\boldsymbol{\theta}}} - \langle \boldsymbol{F}_i^{\tau,\boldsymbol{x}^{\boldsymbol{\theta}}}, \hat{\boldsymbol{x}}_i\rangle\mathbf{1}$ |
| $r_i$ | 4 | $-\mathcal{G}(i, a_i', a_{-i}) + \tau \log \boldsymbol{x}_i^{\boldsymbol{\theta}}(a_i')$;
used in Algorithm 1 for estimating $\boldsymbol{F}_i^{\tau,\boldsymbol{x}^{\boldsymbol{\theta}}} - \langle \boldsymbol{F}_i^{\tau,\boldsymbol{x}^{\boldsymbol{\theta}}}, \hat{\boldsymbol{x}}_i\rangle\mathbf{1}$ |
| $v_i$ | 4 | used in Algorithm 1 for estimating $\boldsymbol{F}_i^{\tau,\boldsymbol{x}^{\boldsymbol{\theta}}} - \langle \boldsymbol{F}_i^{\tau,\boldsymbol{x}^{\boldsymbol{\theta}}}, \hat{\boldsymbol{x}}_i\rangle\mathbf{1}$ |
| $\tilde{\mathcal{L}}_{NAL}^{\tau}(\boldsymbol{\theta})$ | 4 | the estimation of $\mathcal{L}_{NAL}^{\tau}(\boldsymbol{\theta})$ in Algorithm 1 |
| $[i, a_i^s, r_i^s, p_i^s]$ | 4 | the $s$-th tuple stored in buffer $\mathcal{M}_i$ in Algorithm 1 |
| $\boldsymbol{g}_i^s$ | 4 | $\frac{r_i^s - v_i}{p_i^s}\boldsymbol{e}_{a_i^s}$, where $\boldsymbol{e}_{a_i^s}$ is a vector whose the coordinate $a_i^s$ is 1
and all other coordinates are 0;
used in Algorithm 1 for estimating $\boldsymbol{F}_i^{\tau,\boldsymbol{x}^{\boldsymbol{\theta}}} - \langle \boldsymbol{F}_i^{\tau,\boldsymbol{x}^{\boldsymbol{\theta}}}, \hat{\boldsymbol{x}}_i\rangle\mathbf{1}$ |
| $r^{tan}(\boldsymbol{x})$ | E | $\min_{\boldsymbol{z}\in\mathcal{N}_{\boldsymbol{\mathcal{X}}}(\boldsymbol{x})} \| -\nabla_{\boldsymbol{x}}u(\boldsymbol{x}) + \boldsymbol{z}\|_2$,
where $\mathcal{N}_{\boldsymbol{\mathcal{X}}}(\boldsymbol{x}) = \{\boldsymbol{v}\in\mathbb{R}^{|\boldsymbol{\mathcal{X}}|} : \langle \boldsymbol{v}, \boldsymbol{x}' - \boldsymbol{x}\rangle \leq 0, \forall \boldsymbol{x}' \in \boldsymbol{\mathcal{X}}\}$
is the normal cone of $\boldsymbol{x}$,
and $\nabla_{\boldsymbol{x}}u(\boldsymbol{x}) = [\nabla_{\boldsymbol{x}_0}u_0(\boldsymbol{x}); \nabla_{\boldsymbol{x}_1}u_1(\boldsymbol{x}); \cdots ; \nabla_{\boldsymbol{x}_{|\mathcal{N}|-1}}u_{|\mathcal{N}|-1}(\boldsymbol{x})]$;
the tangent residual in the game with the utility function $u_i(\boldsymbol{x})$,
a new metric to measure the distance from the strategy profile $\boldsymbol{x}$ to NE;
if and only if $r^{tan}(\boldsymbol{x}) = 0$, $\boldsymbol{x} \in \boldsymbol{\mathcal{X}}^*$;
used for proving Theorem 4.2 |
| $r^{\tan,\tau}(\boldsymbol{x})$ | E | $\min_{\boldsymbol{z}\in\mathcal{N}_{\boldsymbol{\mathcal{X}}}(\boldsymbol{x})} \| -\nabla_{\boldsymbol{x}}u^{\tau}(\boldsymbol{x}) + \boldsymbol{z}\|_2$;
the tangent residual in the game with the utility function $u_i^{\tau}(\boldsymbol{x})$;
used for proving Theorem 4.2 |
| $Y^{(1)}$, $Y^{(2)}$ | F | two independent variables,
used for deriving the variance of the product of two independent variables |
| $Y$ | F | the expectations of $Y^{(1)}$ and $Y^{(2)}$ |
| $[i, a_i^{2s-1}, r_i^{2s-1}, p_i^{2s-1}]$ | G | the $(2s-1)$-th tuple stored in buffer $\mathcal{M}_i$ in Algorithm 1 |

*Continued on next page*

| Notation/Definition | Section | Description |
|---|---|---|
| $[i, a_i^{2s}, r_i^{2s}, p_i^{2s}]$ | G | the $2s$-th tuple stored in buffer $\mathcal{M}_i$ in Algorithm 1 |
| $\hat{\mathcal{L}}_G^\tau(\boldsymbol{\theta})$ | G | $\sum_{i \in \mathcal{N}} \|sg[\boldsymbol{F}_i^{\boldsymbol{x^\theta}}] + \tau \log \boldsymbol{x}_i^{\boldsymbol{\theta}} - sg[\overline{\boldsymbol{F}_i^{\boldsymbol{x^\theta}}}] - \tau \overline{\log \boldsymbol{x}_i^{\boldsymbol{\theta}}}\|_2^2;$ 
 the implementation of the loss of Gemp et al. (2024) used in our paper |
| $\tilde{\mathcal{L}}_G^\tau(\boldsymbol{\theta})$ | G | $\sum_{i \in \mathcal{N}} \sum_{s=1}^{s=\frac{S}{2}} \langle sg[\hat{\boldsymbol{F}}_{i,2s-1}^{\boldsymbol{x^\theta}}] + \tau \log \boldsymbol{x}_i^{\boldsymbol{\theta}} - sg[\overline{\hat{\boldsymbol{F}}_{i,2s-1}^{\boldsymbol{x^\theta}}}] - \tau \overline{\log \boldsymbol{x}_i^{\boldsymbol{\theta}}},$ 
 $sg[\hat{\boldsymbol{F}}_{i,2s}^{\boldsymbol{x^\theta}}] + \tau \log \boldsymbol{x}_i^{\boldsymbol{\theta}} - sg[\hat{\boldsymbol{F}}_{i,2s}^{\boldsymbol{x^\theta}}] - \tau \overline{\log \boldsymbol{x}_i^{\boldsymbol{\theta}}} \rangle,$ 
 where $\hat{\boldsymbol{F}}_{i,2s-1}^{\boldsymbol{x^\theta}} = \frac{r_i^{2s-1} - \tau \log p_i^{2s-1}}{p_i^{2s-1}} \boldsymbol{e}_{a_i^{2s-1}}$, $\hat{\boldsymbol{F}}_{i,2s}^{\boldsymbol{x^\theta}} = \frac{r_i^{2s} - \tau \log p_i^{2s}}{p_i^{2s}} \boldsymbol{e}_{a_i^{2s}}$, as well as 
 $\boldsymbol{e}_{a_i^{2s-1}}$ ($\boldsymbol{e}_{a_i^{2s}}$) is a vector in which the coordinate $a_i^{2s-1}$ ($a_i^{2s}$) is 1 
 and all other coordinates are 0; 
 the estimation of $\hat{\mathcal{L}}_G^\tau(\boldsymbol{\theta})$ |
| $\mathcal{L}_{ADI}^\tau(\boldsymbol{\theta})$ | G | $\sum_{i \in \mathcal{N}} \max_{\boldsymbol{x}_i' \mathcal{X}_i} \langle sg[\boldsymbol{F}_i^{\boldsymbol{x^\theta}}] + \tau \log \boldsymbol{x}_i^{\boldsymbol{\theta}}, \boldsymbol{x}' - \boldsymbol{x}_i^{\boldsymbol{\theta}} \rangle;$ 
 the implementation of the loss of Gemp et al. (2022) used in our paper |
| $\tilde{\mathcal{L}}_{ADI}^\tau(\boldsymbol{\theta})$ | G | $\sum_{i \in \mathcal{N}} \max_{\boldsymbol{x}_i' \mathcal{X}_i} \langle sg[\hat{\boldsymbol{F}}_i^{\boldsymbol{x^\theta}}] + \tau \log \boldsymbol{x}_i^{\boldsymbol{\theta}}, \boldsymbol{x}' - \boldsymbol{x}_i^{\boldsymbol{\theta}} \rangle,$ 
 where $\hat{\boldsymbol{F}}_{i,s}^{\boldsymbol{x^\theta}} = \frac{r_i^s - \tau \log p_i^s}{p_i^s} \boldsymbol{e}_{a_i^s}$, $\hat{\boldsymbol{F}}_i^{\boldsymbol{x^\theta}} = \sum_{s=1}^{s=S} \hat{\boldsymbol{F}}_{i,s}^{\boldsymbol{x^\theta}},$ 
 as well as $\boldsymbol{e}_{a_i^s}$ is a vector in which the coordinate $a_i^s$ is 1 
 and all other coordinates are 0; 
 the estimation of $\mathcal{L}_{ADI}^\tau(\boldsymbol{\theta})$ |
| $\mathcal{L}_{NashApr}(\boldsymbol{\theta})$ | G | $\max_{i \in \mathcal{N}} \max_{\boldsymbol{x}_i' \mathcal{X}_i} \langle sg[\boldsymbol{F}_i^{\boldsymbol{x^\theta}}], \boldsymbol{x}' - \boldsymbol{x}_i^{\boldsymbol{\theta}} \rangle;$ 
 the implementation of the loss of Duan et al. (2023) used in our paper |
| $\tilde{\mathcal{L}}_{NashApr}^\tau(\boldsymbol{\theta})$ | G | $\max_{i \in \mathcal{N}} \max_{\boldsymbol{x}_i' \mathcal{X}_i} \langle sg[\hat{\boldsymbol{F}}_i^{\boldsymbol{x^\theta}}], \boldsymbol{x}' - \boldsymbol{x}_i^{\boldsymbol{\theta}} \rangle,$ 
 where $\hat{\boldsymbol{F}}_{i,s}^{\boldsymbol{x^\theta}} = \frac{r_i^s - \tau \log p_i^s}{p_i^s} \boldsymbol{e}_{a_i^s}$, $\hat{\boldsymbol{F}}_i^{\boldsymbol{x^\theta}} = \sum_{s=1}^{s=S} \hat{\boldsymbol{F}}_{i,s}^{\boldsymbol{x^\theta}},$ 
 as well as $\boldsymbol{e}_{a_i^s}$ is a vector in which the coordinate $a_i^s$ is 1 
 and all other coordinates are 0; 
 the estimation of $\mathcal{L}_{NashApr}(\boldsymbol{\theta})$ |
| $\mathcal{L}_0^\tau(\boldsymbol{x})$ | H | $\sum_{i \in \mathcal{N}} \|sg[\boldsymbol{F}_i^{\tau, \boldsymbol{x}}] - sg[\overline{\boldsymbol{F}_i^{\tau, \boldsymbol{x}}}] + \tau \boldsymbol{x}_i - \overline{\tau \boldsymbol{x}_i}\|_2^2,$ 
 where $\overline{\tau \boldsymbol{x}_i} = \frac{\tau \sum_{i \in \mathcal{A}_i} \boldsymbol{x}_i}{|\mathcal{A}|} \mathbf{1};$ 
 a variant of the loss of Gemp et al. (2024) that is used for deriving 
 the relationship between NAL and the loss of Gemp et al. (2024) |
| $\mathcal{L}_1^\tau(\boldsymbol{x})$ | H | $\sum_{i \in \mathcal{N}} (2\tau \langle sg[\boldsymbol{F}_i^{\tau, \boldsymbol{x}} - \langle \boldsymbol{F}_i^{\tau, \boldsymbol{x}}, \boldsymbol{x}_i^u \rangle \mathbf{1}], \boldsymbol{x}_i \rangle + \langle \tau \boldsymbol{x}_i, \tau \boldsymbol{x}_i \rangle - \langle \tau \overline{\boldsymbol{x}_i}, \tau \overline{\boldsymbol{x}_i} \rangle);$ 
 used for deriving the relationship 
 between NAL and the loss of Gemp et al. (2024) |

## B. Discussion on Learning Nash equilibria via the Combination of Deep Neural Networks with Non-Convex (Stochastic) Optimization Techniques

Deep neural networks (DNNs) have been widely applied to solve complex optimization and decision-making problems across diverse domains, owing to their expressive power. This enables them to capture intricate patterns and relationships that are challenging to model using traditional methods. DNNs can approximate arbitrary non-linear functions (Hornik, 1991), making them particularly suitable for representing strategic decision-making processes in game theory. Furthermore, the ability of DNNs to learn directly from raw data—without requiring handcrafted features—allows them to uncover complex equilibrium strategies that might otherwise be overlooked (Brown et al., 2019; Goktas et al., 2022; Marris et al., 2022; Liu et al., 2024).

A variety of methods combining DNNs with non-convex (stochastic) optimization techniques has been developed to compute an NE, where strategy profiles are represented by DNNs and the equilibrium is directly learned by minimizing a loss

function through non-convex (stochastic) optimization techniques (Goktas et al., 2022; Marris et al., 2022; Liu et al., 2024). To the best of our knowledge, the first work on computing an NE via the combination of DNNs with non-convex (stochastic) optimization techniques was proposed by Duan et al. (2023), demonstrating the feasibility of combining DNNs with non-convex optimization techniques for computing an NE. Following this, Marris et al. (2022), Goktas et al. (2022), and Liu et al. (2024) conducted extensive research on network architectures tailored for computing an NE through this combination. However, these architectures are not suitable for solving real-world games, as they assume that the payoff matrix can be fully loaded into memory as input to the network. In contrast, real-world games often feature payoff matrices too large to fit into memory, necessitating solutions based on sampling a subset of the matrix, referred to as sampled play, which can be seen as the stochastic optimization setting in ML.

In the context of stochastic optimization, designing appropriate loss functions specifically tailored for equilibrium computation remains a significant challenge. Many existing loss functions rely on sampling to estimate gradients or payoffs, which can introduce considerable biases (Duan et al., 2023; Gemp et al., 2022) or lead to high variance (Gemp et al., 2024), thereby making training unstable. This limitation highlights the need for further research in developing unbiased, low-variance loss functions that better align with the requirements of equilibrium computation. Nonetheless, the intersection of DNNs, non-convex (stochastic) optimization techniques, and game theory presents a promising avenue for addressing complex, real-world problems, ranging from economic markets to strategic decision-making in games.

## C. Relationship between Our Algorithm and Simultaneous Gradient Descent Algorithms

Our algorithm is similar to existing simultaneous gradient descent algorithms, specifically to its improved version, magnetic mirror descent (Sokota et al., 2023), if we set $\hat{\boldsymbol{x}}_i = \boldsymbol{0}$. However, as mentioned in Section 4, the property of NAL does not hold when $\hat{\boldsymbol{x}}_i = \boldsymbol{0}$ (we also highlight around Eq. (2) that $\hat{\boldsymbol{x}}_i \neq \boldsymbol{0}$ in NAL).

Firstly, we cannot relate the zero-point of the first-order gradient of existing simultaneous gradient descent algorithms (NAL with $\hat{\boldsymbol{x}}_i = \boldsymbol{0}$) to NE (Theorem 4.2 and 4.3). Specifically, when $\hat{\boldsymbol{x}}_i = \boldsymbol{0}$ and applying commonly used non-convex stochastic optimization techniques in ML, we cannot guarantee that the algorithm will stop updating upon reaching NE. This is because, even if $\boldsymbol{x}$ is an NE of the regularization game, existing simultaneous gradient descent algorithms only ensure that its first-order gradient $sg[\boldsymbol{F}_i^{\tau,\boldsymbol{x}} - \langle \boldsymbol{F}_i^{\tau,\boldsymbol{x}}, \boldsymbol{0}\rangle \mathbf{1}] = sg[\boldsymbol{F}_i^{\tau,\boldsymbol{x}}]$ is the same across all actions (as shown in the analysis in Section 4), but does not guarantee that this gradient is $\boldsymbol{0}$. Unfortunately, the algorithm that employs commonly used non-convex stochastic optimization techniques in ML, will only stop updating when the first-order gradient of the loss function is $\boldsymbol{0}$. In other words, even if an NE is learned by existing simultaneous gradient descent algorithms, the updates may not stop, and existing simultaneous gradient descent algorithms will eventually deviate from this NE. In contrast, NAL ensures that its first-order gradient $sg[\boldsymbol{F}_i^{\tau,\boldsymbol{x}} - \langle \boldsymbol{F}_i^{\tau,\boldsymbol{x}}, \hat{\boldsymbol{x}}_i\rangle \mathbf{1}] = \boldsymbol{0}$ if and only if $\boldsymbol{x}$ is an NE of the regularization game (as shown in the analysis in Section 4), which enables the algorithm employing commonly used non-convex stochastic optimization techniques in ML to stop updating once an NE is learned.

Secondly, no connection can be established between existing simultaneous gradient descent algorithms and the loss function proposed by Gemp et al. (2024). However, as shown in Theorem H.1, the first-order gradient of NAL approximates a variant of the first-order gradient of the loss function introduced by Gemp et al. (2024). This suggests that, if we can establish theoretical convergence for the loss function proposed by Gemp et al. (2024) when using commonly applied non-convex stochastic optimization techniques to minimize this loss function, we may also be able to establish similar convergence results for NAL under these techniques. However, this does not hold for existing simultaneous gradient descent algorithms.

Regarding the theoretical convergence of existing simultaneous gradient descent algorithms, these algorithms are primarily used in training GANs (Goodfellow et al., 2014; Mescheder et al., 2018), which typically involve two-player zero-sum games, as opposed to multi-player general-sum games that we consider. Notably, while research (Milionis et al., 2023) suggests that there may be games where dynamics do not converge to an NE, their theory does not account for randomness. As the authors state: "our impossibility results do not apply to stochastic dynamics—e.g., discrete-time dynamics in which $\varphi(\boldsymbol{x})$ is a distribution of possible next points" (where $\varphi(\boldsymbol{x})$ is the notation in Milionis et al. (2023)). Yongacoglu et al. (2024) demonstrate that the presence of randomness can enable convergence to an NE, noting that "we note that our possibility result does not contradict the impossibility results of [22, 2] or [38]. In particular, the functions $\{f_\Gamma^i\}_{i=1}^n$ need not be (and usually will not be) continuous, violating the regularity conditions of [22] and [38]" (where $\{f_\Gamma^i\}_{i=1}^n$ is the notation in Yongacoglu et al. (2024), and [38] refers to (Milionis et al., 2023)). Our algorithm incorporates randomness. Studying the theoretical convergence of our algorithm will be the future work.

Thirdly, the experimental results in Figure 20 and Figure 21 further highlight the distinct nature of NAL and existing simultaneous gradient descent algorithms, as their performance diverges significantly. Specifically, a minor modification to the neural network architecture—namely, replacing the activation function in the final layer—causes existing simultaneous gradient descent algorithms to fail to converge, whereas NAL's convergence rate remains unaffected by this structural change.

## D. More Details of Stop-Gradient Operator

Let $\boldsymbol{b} \in \mathbb{R}^d$ be a variable. The stop-gradient operator is defined as $sg[\boldsymbol{b}] = \boldsymbol{b} \in \mathbb{R}^d$ with $\nabla_{\boldsymbol{b}} sg[\boldsymbol{b}] = 0 \in \mathbb{R}^{n \times n}$. This implies that $sg[\boldsymbol{b}]$ passes the value of $\boldsymbol{b}$ unchanged in the forward pass, but blocks its gradient during backpropagation. Intuitively, $sg[\cdot]$ can be regarded as a constant during differentiation. In summary,

- **Forward pass.** $sg[\boldsymbol{b}]$ returns the value of $\boldsymbol{b}$.

- **Backward pass.** The gradient is blocked—no gradients are propagated through $\boldsymbol{b}$.

In fact, this operator has already been widely adopted in previous works (Grill et al., 2020; Flennerhag et al., 2020; Chen & He, 2021). We are inspired by the use of this operator in previous works and adopt it in our work accordingly.

## E. Missing Proofs in Section 4

### E.1. Proof of Lemma 4.1

*Proof.* Let $\boldsymbol{b} = (b_1, b_2, \ldots, b_d) \in \mathbb{R}^d$ and $\boldsymbol{y} = (y_1, y_2, \ldots, y_d)$ be a vector in the standard simplex, i.e., $y_i \geq 0$ and $\sum_{k=1}^{n} y_k = 1$.

The inner product $\langle \boldsymbol{b}, \boldsymbol{y} \rangle$ is defined as

$$\langle \boldsymbol{b}, \boldsymbol{y} \rangle = \sum_{k=1}^{n} b_k y_k.$$

We will now prove the lemma in two parts: sufficiency and necessity.

**Sufficiency:**

Assume that all coordinates of $\boldsymbol{b}$ are equal, i.e., $b_1 = b_2 = \cdots = b_d = c$, where $c$ is some constant. In this case, we have

$$\langle \boldsymbol{b}, \boldsymbol{y} \rangle = \sum_{k=1}^{n} b_k y_k = \sum_{k=1}^{n} c y_k = c \sum_{k=1}^{n} y_k = c \cdot 1 = c.$$

Thus,

$$\boldsymbol{b} - \langle \boldsymbol{b}, \boldsymbol{y} \rangle \mathbf{1} = (c, c, \ldots, c) - c = (0, 0, \ldots, 0),$$

which implies that $\boldsymbol{b} - \langle \boldsymbol{b}, \boldsymbol{y} \rangle \mathbf{1} = \mathbf{0}$. Hence, the sufficiency holds.

**Necessity:**

Now, assume that $\boldsymbol{b} - \langle \boldsymbol{b}, \boldsymbol{y} \rangle \mathbf{1} = \mathbf{0}$. We need to show that this implies that all coordinates of $\boldsymbol{b}$ are equal. From the equation $\boldsymbol{b} - \langle \boldsymbol{b}, \boldsymbol{y} \rangle \mathbf{1} = \mathbf{0}$, we have

$$\boldsymbol{b} = \langle \boldsymbol{b}, \boldsymbol{y} \rangle \mathbf{1},$$

where $\mathbf{1} = (1, 1, \ldots, 1)$ is the vector of all ones. This implies that

$$b_k = \langle \boldsymbol{b}, \boldsymbol{y} \rangle \quad \text{for all } k = 1, 2, \ldots, d.$$

In other words, all $b_k$ are equal to $\langle \boldsymbol{b}, \boldsymbol{y} \rangle$, meaning $b_1 = b_2 = \cdots = b_d$. Thus, the necessity holds.

Since both sufficiency and necessity have been proven, the lemma is true. $\square$

### E.2. Proof of Theorem 4.2

*Proof.* We prove Theorem 4.2 via the tangent residual (Cai et al., 2022). Therefore, before we start the proof, we first introduce the tangent residual. Formally, for any game, whose utility function $u(\cdot)$ of each player $i$ is concave over $\mathcal{X}_i$, $\forall \boldsymbol{x} \in \mathcal{X}$, its tangent residual is

$$r^{tan}(\boldsymbol{x}) = \min_{\boldsymbol{z} \in \mathcal{N}_{\mathcal{X}}(\boldsymbol{x})} \| - \nabla_{\boldsymbol{x}} u(\boldsymbol{x}) + \boldsymbol{z} \|_2,$$

where $\mathcal{N}_{\mathcal{X}}(\boldsymbol{x}) = \{ \boldsymbol{v} \in \mathbb{R}^{|\mathcal{X}|} : \langle \boldsymbol{v}, \boldsymbol{x}' - \boldsymbol{x} \rangle \leq 0, \forall \boldsymbol{x}' \in \mathcal{X} \}$ is the normal cone of $\boldsymbol{x}$, and $\nabla_{\boldsymbol{x}} u(\boldsymbol{x}) = [\nabla_{\boldsymbol{x}_0} u_0(\boldsymbol{x}); \nabla_{\boldsymbol{x}_1} u_1(\boldsymbol{x}); \cdots; \nabla_{\boldsymbol{x}_{n-1}} u_{n-1}(\boldsymbol{x})]$. If $r^{tan}(\boldsymbol{x}) = 0$, then, by Lemma E.1, $\boldsymbol{x}$ is an NE. Additionally, if $\boldsymbol{x}$ is an NE, then $\nabla_{\boldsymbol{x}} u(\boldsymbol{x}) \in \mathcal{N}_{\mathcal{X}}(\boldsymbol{x})$, since $\langle \nabla_{\boldsymbol{x}} u(\boldsymbol{x}), \boldsymbol{x}' - \boldsymbol{x} \rangle \leq 0, \forall \boldsymbol{x}' \in \mathcal{X}$ when $\boldsymbol{x}$ is an NE. Therefore, $r^{tan}(\boldsymbol{x}) = \| - \nabla_{\boldsymbol{x}} u(\boldsymbol{x}) + \nabla_{\boldsymbol{x}} u(\boldsymbol{x}) \|_2 = 0$ if and only if $\boldsymbol{x}$ is an NE.

**Lemma E.1** (Proof is in Appendix E.3). *For any game, whose utility function of each player $i$ is concave over $\mathcal{X}_i$, it holds that*

$$dg(\boldsymbol{x}) \leq \sqrt{2|\mathcal{N}|} r^{tan}(\boldsymbol{x}).$$

From the definition of $-\langle \boldsymbol{F}_i^{\tau,\boldsymbol{x}}, \boldsymbol{x}_i' \rangle \mathbf{1}$, $\forall \boldsymbol{x}, \boldsymbol{x}' \in \mathcal{X}$, we have

$$\sum_{i \in \mathcal{N}} \langle -\langle \boldsymbol{F}_i^{\tau,\boldsymbol{x}}, \hat{\boldsymbol{x}}_i \rangle \mathbf{1}, \boldsymbol{x}_i' - \boldsymbol{x}_i \rangle = \sum_{i \in \mathcal{N}} -\langle \boldsymbol{F}_i^{\tau,\boldsymbol{x}}, \hat{\boldsymbol{x}}_i \rangle + \langle \boldsymbol{F}_i^{\tau,\boldsymbol{x}}, \hat{\boldsymbol{x}}_i \rangle = 0,$$

where the first equality comes from $\langle \mathbf{1}, \boldsymbol{x}_i' \rangle = \langle \mathbf{1}, \boldsymbol{x}_i \rangle = 1$ since $\boldsymbol{x}_i'$ and $\boldsymbol{x}_i$ are in the simplex. Therefore, $[-\langle \boldsymbol{F}_i^{\tau,\boldsymbol{x}}, \hat{\boldsymbol{x}}_0 \rangle \mathbf{1}, -\langle \boldsymbol{F}_i^{\tau,\boldsymbol{x}}; \hat{\boldsymbol{x}}_1 \rangle \mathbf{1}; \cdots; -\langle \boldsymbol{F}_i^{\tau,\boldsymbol{x}}, \hat{\boldsymbol{x}}_{|\mathcal{N}|-1} \rangle \mathbf{1}]$ is in the normal cone of $\boldsymbol{x}$. Then, from the definition of the tangent residual,

$$r^{\tan,\tau}(\boldsymbol{x}) = \min_{\boldsymbol{z} \in \mathcal{N}_{\mathcal{X}}(\boldsymbol{x})} \| - \nabla_{\boldsymbol{x}} u^{\tau}(\boldsymbol{x}) + \boldsymbol{z} \|_2,$$

with $\mathcal{N}_{\mathcal{X}}(\boldsymbol{x})$ is the normal cone of $\boldsymbol{x}$, and

$$-\nabla_{\boldsymbol{x}_i} u_i^{\tau}(\boldsymbol{x}) = \boldsymbol{F}_i^{\tau,\boldsymbol{x}},$$

we have that

$$r^{\tan,\tau}(\boldsymbol{x}) \leq \| \nabla_{\boldsymbol{x}} \mathcal{L}_{NAL}^{\tau}(\boldsymbol{x}) \|_2.$$

In addition, from Lemma E.1, we can obtain that

$$dg^{\tau}(\boldsymbol{x}) \leq \sqrt{2|\mathcal{N}|} r^{\tan,\tau}(\boldsymbol{x}) \leq \sqrt{2|\mathcal{N}|} \| \nabla_{\boldsymbol{x}} \mathcal{L}_{NAL}^{\tau}(\boldsymbol{x}) \|_2.$$

It completes the proof. $\square$

### E.3. Proof of Lemma E.1

*Proof.* Let $\boldsymbol{x}_i' = \arg\max_{\boldsymbol{x}_i' \in \mathcal{X}_i} \langle \nabla_{\boldsymbol{x}_i} u_i(\boldsymbol{x}), \boldsymbol{x}_i' - \boldsymbol{x}_i \rangle$, for the definition of the duality gap and normal cone, $\forall \boldsymbol{z} \in \mathcal{N}_{\mathcal{X}}(\boldsymbol{x})$, we have

$$
\begin{aligned}
dg(\boldsymbol{x}) &= \sum_{i \in \mathcal{N}} \langle \nabla_{\boldsymbol{x}_i} u_i(\boldsymbol{x}), \boldsymbol{x}_i' - \boldsymbol{x}' \rangle \\
&\leq \sum_{i \in \mathcal{N}} \langle \nabla_{\boldsymbol{x}_i} u_i(\boldsymbol{x}), \boldsymbol{x}_i' - \boldsymbol{x}_i \rangle + \langle \boldsymbol{z}, \boldsymbol{x} - \boldsymbol{x}' \rangle \\
&= \langle -\nabla_{\boldsymbol{x}} u(\boldsymbol{x}) + \boldsymbol{z}, \boldsymbol{x} - \boldsymbol{x}' \rangle \\
&\leq \| - \nabla_{\boldsymbol{x}} u(\boldsymbol{x}) + \boldsymbol{z} \|_2 \| \boldsymbol{x} - \boldsymbol{x}' \|_2,
\end{aligned}
\tag{6}
$$

where the second lines comes from the fact that, $\forall \boldsymbol{z} \in \mathcal{N}_{\mathcal{X}}(\boldsymbol{x})$ and $\boldsymbol{x}'' \in \mathcal{X}, \langle \boldsymbol{z}, \boldsymbol{x} - \boldsymbol{x}'' \rangle \geq 0$ holds. As Eq. (6) holds for all $\boldsymbol{z} \in \mathcal{N}_{\mathcal{X}}(\boldsymbol{x})$, we can get

$$dg(\boldsymbol{x}) \leq \| \boldsymbol{x} - \boldsymbol{x}' \|_2 \min_{\boldsymbol{z} \in \mathcal{N}_{\mathcal{X}}(\boldsymbol{x})} \| - \nabla_{\boldsymbol{x}} u(\boldsymbol{x}) + \boldsymbol{z} \|_2,$$

which implies

$$dg(\boldsymbol{x}) \leq \sqrt{2|\mathcal{N}|} r^{\tan}(\boldsymbol{x}),$$

where the right-hand side comes from $\max_{\boldsymbol{x}'', \boldsymbol{x}''' \in \mathcal{X}} \| \boldsymbol{x}'' - \boldsymbol{x}''' \|_2 \leq \sqrt{\sum_{i \in \mathcal{N}} \max_{\boldsymbol{x}_i'', \boldsymbol{x}_i''' \in \mathcal{X}_i} \| \boldsymbol{x}_i'' - \boldsymbol{x}_i''' \|_2^2} \leq \sqrt{\sum_{i \in \mathcal{N}} 2} = \sqrt{2|\mathcal{N}|}$. It completes the proof. $\square$

### E.4. Proof of Theorem 4.3

*Proof.* Beginning with the definition of the duality gap, we find

$$
\begin{aligned}
\mathrm{dg}(\boldsymbol{x}) &= \sum_{i\in\mathcal{N}} \max_{\boldsymbol{x}_i'\in\boldsymbol{\mathcal{X}}_i} \langle \nabla_{\boldsymbol{x}_i} u_i(\boldsymbol{x}), \boldsymbol{x}_i' - \boldsymbol{x}_i \rangle \\
&= \sum_{i\in\mathcal{N}} \max_{\boldsymbol{x}_i'\in\boldsymbol{\mathcal{X}}_i} \langle \nabla_{\boldsymbol{x}_i} u_i(\boldsymbol{x}) - \tau\log(\boldsymbol{x}_i) + \tau\log(\boldsymbol{x}_i), \boldsymbol{x}_i' - \boldsymbol{x}_i \rangle \\
&\leq \sum_{i\in\mathcal{N}} \max_{\boldsymbol{x}_i'\in\boldsymbol{\mathcal{X}}_i} \langle \nabla_{\boldsymbol{x}_i} u_i(\boldsymbol{x}) - \tau\log(\boldsymbol{x}_i), \boldsymbol{x}_i' - \boldsymbol{x}_i \rangle + \sum_{i\in\mathcal{N}} \max_{\boldsymbol{x}_i'\in\boldsymbol{\mathcal{X}}_i} \langle \tau\log(\boldsymbol{x}_i), \boldsymbol{x}_i' - \boldsymbol{x}_i \rangle \\
&\leq \mathrm{dg}^\tau(\boldsymbol{x}) + \sum_{i\in\mathcal{N}} \max_{\boldsymbol{x}_i'\in\boldsymbol{\mathcal{X}}_i} \langle \tau\log(\boldsymbol{x}_i), \boldsymbol{x}_i' \rangle + \sum_{i\in\mathcal{N}} \langle \tau\log(\boldsymbol{x}_i), -\boldsymbol{x}_i \rangle \\
&\leq \sqrt{2|\mathcal{N}|} \|\nabla_{\boldsymbol{x}} \mathcal{L}_{NAL}^\tau(\boldsymbol{x})\|_2 + \sum_{i\in\mathcal{N}} \langle \tau\log(\boldsymbol{x}_i), -\boldsymbol{x}_i \rangle \\
&\leq \sqrt{2|\mathcal{N}|} \|\nabla_{\boldsymbol{x}} \mathcal{L}_{NAL}^\tau(\boldsymbol{x})\|_2 + \tau \sum_{i\in\mathcal{N}} \log(|\mathcal{A}_i|).
\end{aligned}
$$

where the second inequality follows from the definition of the duality gap and NE, and the third inequality comes from $\log(\boldsymbol{x}_i) \leq \boldsymbol{0}, \forall 0 \leq \boldsymbol{x}_i \leq 1$. It completes the proof. $\square$

## F. Variance of Estimating via Two Independent and Identically Distributed Random Variables

Let two independent samples from two corresponding identically distributed random variables be $Y^{(1)}$ and $Y^{(2)}$. Due to the definition of $Y^{(1)}$ and $Y^{(2)}$, we have $\mathbb{E}[Y^{(1)}] = \mathbb{E}[Y^{(2)}] = Y$. Assume $\mathrm{Var}[Y^{(1)}] = \sigma$ and $\mathrm{Var}[Y^{(2)}] = \sigma$. Now, we aim to analyze the variance $\mathrm{Var}[Y^{(1)}Y^{(2)}]$. From the definition of $\mathrm{Var}[Y^{(1)}Y^{(2)}]$, we have

$$
\mathrm{Var}[Y^{(1)}Y^{(2)}] = \mathbb{E}[(Y^{(1)})^2]\mathbb{E}[(Y^{(2)})^2] - (\mathbb{E}[Y^{(1)}]\mathbb{E}[Y^{(2)}])^2.
$$

For the term $\mathbb{E}[Y^{(1)}]^2$, from the definition of $\mathrm{Var}[Y^{(1)}]$, we get

$$
\begin{aligned}
\mathrm{Var}[Y^{(1)}] &= \mathbb{E}[(Y^{(1)})^2] - (\mathbb{E}[Y^{(1)}])^2 = \mathbb{E}[(Y^{(1)})^2] - Y^2 = \sigma \\
\Rightarrow \mathbb{E}[(Y^{(1)})^2] &= \sigma + Y^2.
\end{aligned}
$$

Similarly, we have $\mathbb{E}[(Y^{(2)})^2] = \sigma + Y^2$. Combining the above equities, we have

$$
\mathrm{Var}[Y^{(1)}Y^{(2)}] = (\sigma + Y^2)(\sigma + Y^2) - (Y^2)^2 = \sigma^2 + 2\sigma Y^2.
$$

## G. Implementation Details of Compared Loss Functions

**The loss function proposed by Gemp et al. (2024).** As shown in Algorithm 1, we do not employ the sampling method used in Gemp et al. (2022) and Gemp et al. (2024) due to the high per-sample sampling complexity of their sampling method. Formally, the per-sample sampling complexity of their estimating method is $O(|\mathcal{N}||\mathcal{A}_i|^2)$ while that of our estimating method as shown in Algorithm 1 is $O(|\mathcal{N}|)$. In Appendix J, we provide a comparison of the sampling time between our sampling method and the sampling method used in Gemp et al. (2022) and Gemp et al. (2024), under the same number of sampled instances $S$. In our sampling method, the estimated variable of $\boldsymbol{F}_i^{\boldsymbol{x}}$ (*i.e.*, $-\nabla_{\boldsymbol{x}_i} u_i(\boldsymbol{x})$) cannot participate in gradient backpropagation, and only the variable $\boldsymbol{x}_i^{\boldsymbol{\theta}}$ participates in gradient backpropagation. In other words, we minimize the following loss function

$$
\hat{\mathcal{L}}_G^\tau(\boldsymbol{\theta}) = \sum_{i\in\mathcal{N}} \|sg[\boldsymbol{F}_i^{\boldsymbol{x}^{\boldsymbol{\theta}}}] + \tau\log\boldsymbol{x}_i^{\boldsymbol{\theta}} - sg[\overline{\boldsymbol{F}_i^{\boldsymbol{x}^{\boldsymbol{\theta}}}} + \tau\overline{\log\boldsymbol{x}_i^{\boldsymbol{\theta}}}]\|_2^2.
$$

The only distinction between $\hat{\mathcal{L}}_G^\tau(\boldsymbol{\theta})$ and $\mathcal{L}_G^\tau(\boldsymbol{\theta})$ in Eq. (1) (or $\mathcal{L}_G^\tau(\boldsymbol{x})$) is the omission of the gradient backpropagation path for $sg[\boldsymbol{F}_i^{\boldsymbol{x}^{\boldsymbol{\theta}}}]$ and $sg[\overline{\boldsymbol{F}_i^{\boldsymbol{x}^{\boldsymbol{\theta}}}}]$. For the definition of the stop-gradient operator in Appendix D, the stop-gradient operator

only affects the gradient computation. Specifically, the stop-gradient operator treats a term as a constant, preventing it from participating in the gradient backpropagation, but does not alter the value of the term. Based on the analysis in Section 3, for all $\boldsymbol{x^\theta} \in \mathcal{X}$, $\boldsymbol{x^\theta}$ is an NE of the regularization game with utility function $u_i^\tau(\boldsymbol{x^\theta}) = u_i(\boldsymbol{x^\theta}) - \tau \boldsymbol{x_i^\theta}^{\mathrm{T}} \log \boldsymbol{x_i^\theta}$ if and only if $\mathcal{L}_G^\tau(\boldsymbol{x^\theta}) = 0$. Since the stop-gradient operator does not alter the value, it follows that for all $\boldsymbol{x^\theta} \in \mathcal{X}$, $\boldsymbol{x^\theta}$ is an NE of the regularization game with the utility function $u_i^\tau(\boldsymbol{x^\theta}) = u_i(\boldsymbol{x^\theta}) - \tau \boldsymbol{x_i^\theta}^{\mathrm{T}} \log \boldsymbol{x_i^\theta}$ if and only if $\hat{\mathcal{L}}_G^\tau(\boldsymbol{\theta}) = 0$.

As done in Gemp et al. (2024), we use the following loss function to estimate the value of $\hat{\mathcal{L}}_G^\tau(\boldsymbol{\theta})$ via the $(2s-1)$-th and $(2s)$-th tuples ($[i, a_i^{2s-1}, r_i^{2s-1}, p_i^{2s-1}]$ and $[i, a_i^{2s}, r_i^{2s}, p_i^{2s}]$) stored in $\mathcal{M}_i$:

$$\tilde{\mathcal{L}}_G^\tau(\boldsymbol{\theta}) = \sum_{i \in \mathcal{N}} \sum_{s=1}^{s=\frac{S}{2}} \langle sg[\hat{\boldsymbol{F}}_{i,2s-1}^{\boldsymbol{x^\theta}}] + \tau \log \boldsymbol{x_i^\theta} - sg[\overline{\hat{\boldsymbol{F}}_{i,2s-1}^{\boldsymbol{x^\theta}}}] - \tau \overline{\log \boldsymbol{x_i^\theta}}, sg[\hat{\boldsymbol{F}}_{i,2s}^{\boldsymbol{x^\theta}}] + \tau \log \boldsymbol{x_i^\theta} - sg[\overline{\hat{\boldsymbol{F}}_{i,2s}^{\boldsymbol{x^\theta}}}] - \tau \overline{\log \boldsymbol{x_i^\theta}} \rangle,$$

$$\hat{\boldsymbol{F}}_{i,2s-1}^{\boldsymbol{x^\theta}} = \frac{r_i^{2s-1} - \tau \log p_i^{2s-1}}{p_i^{2s-1}} \boldsymbol{e}_{a_i^{2s-1}}, \quad \hat{\boldsymbol{F}}_{i,2s}^{\boldsymbol{x^\theta}} = \frac{r_i^{2s} - \tau \log p_i^{2s}}{p_i^{2s}} \boldsymbol{e}_{a_i^{2s}},$$

where $\boldsymbol{e}_{a_i^{2s-1}}$ ($\boldsymbol{e}_{a_i^{2s}}$) is a vector in which the coordinate $a_i^{2s-1}$ ($a_i^{2s}$) is 1 and all other coordinates are 0. From the analysis in Gemp et al. (2024), $\mathbb{E}[\hat{\boldsymbol{F}}_{i,2s-1}^{\boldsymbol{x^\theta}}] = \boldsymbol{F}_i^{\boldsymbol{x^\theta}}$, $\mathbb{E}[\hat{\boldsymbol{F}}_{i,2s}^{\boldsymbol{x^\theta}}] = \boldsymbol{F}_i^{\boldsymbol{x^\theta}}$, and $\frac{2}{S} \mathbb{E}[\tilde{\mathcal{L}}_G^\tau(\boldsymbol{\theta})] = \hat{\mathcal{L}}_G^\tau(\boldsymbol{\theta})$.

**ADI (Gemp et al., 2022).** Since the variable $\boldsymbol{F}_i^{\boldsymbol{x}}$ cannot participate in gradient backpropagation via our sampling method, we defined this loss function as

$$\mathcal{L}_{ADI}^\tau(\boldsymbol{\theta}) = \sum_{i \in \mathcal{N}} \max_{\boldsymbol{x}_i' \mathcal{X}_i} \langle sg[\boldsymbol{F}_i^{\boldsymbol{x^\theta}}] + \tau \log \boldsymbol{x_i^\theta}, \boldsymbol{x}' - \boldsymbol{x_i^\theta} \rangle.$$

We use the following loss function to estimate the value of $\mathcal{L}_{ADI}^\tau(\boldsymbol{\theta})$ via the tuples $[i, a_i^s, r_i^s, p_i^s]$ ($s \in [1, 2, \cdots, S]$) stored in $\mathcal{M}_i$:

$$\hat{\boldsymbol{F}}_{i,s}^{\boldsymbol{x^\theta}} = \frac{r_i^s - \tau \log p_i^s}{p_i^s} \boldsymbol{e}_{a_i^s}, \quad \hat{\boldsymbol{F}}_i^{\boldsymbol{x^\theta}} = \sum_{s=1}^{s=S} \hat{\boldsymbol{F}}_{i,s}^{\boldsymbol{x^\theta}},$$

$$\tilde{\mathcal{L}}_{ADI}^\tau(\boldsymbol{\theta}) = \sum_{i \in \mathcal{N}} \max_{\boldsymbol{x}_i' \mathcal{X}_i} \langle sg[\hat{\boldsymbol{F}}_i^{\boldsymbol{x^\theta}}] + \tau \log \boldsymbol{x_i^\theta}, \boldsymbol{x}' - \boldsymbol{x_i^\theta} \rangle.$$

**NashApr (Duan et al., 2023).** Since the variable $\boldsymbol{F}_i^{\boldsymbol{x}}$ cannot participate in gradient backpropagation via our sampling method, we defined this loss function as

$$\mathcal{L}_{NashApr}(\boldsymbol{\theta}) = \max_{i \in \mathcal{N}} \max_{\boldsymbol{x}_i' \mathcal{X}_i} \langle sg[\boldsymbol{F}_i^{\boldsymbol{x^\theta}}], \boldsymbol{x}' - \boldsymbol{x_i^\theta} \rangle.$$

We use the following loss function to estimate the value of $\mathcal{L}_{NashApr}(\boldsymbol{\theta})$ via the tuples $[i, a_i^s, r_i^s, p_i^s]$ ($s \in [1, 2, \cdots, S]$) stored in $\mathcal{M}_i$:

$$\hat{\boldsymbol{F}}_{i,s}^{\boldsymbol{x^\theta}} = \frac{r_i^s - \tau \log p_i^s}{p_i^s} \boldsymbol{e}_{a_i^s}, \quad \hat{\boldsymbol{F}}_i^{\boldsymbol{x^\theta}} = \sum_{s=1}^{s=S} \hat{\boldsymbol{F}}_{i,s}^{\boldsymbol{x^\theta}}, \quad \tilde{\mathcal{L}}_{NashApr}^\tau(\boldsymbol{\theta}) = \max_{i \in \mathcal{N}} \max_{\boldsymbol{x}_i' \mathcal{X}_i} \langle sg[\hat{\boldsymbol{F}}_i^{\boldsymbol{x^\theta}}], \boldsymbol{x}' - \boldsymbol{x_i^\theta} \rangle.$$

# H. Minimizing NAL Approximates the Process of Minimizing the Loss Function in Gemp et al. (2024)

We now show that when using commonly used non-convex stochastic optimization techniques in ML, minimizing NAL approximates the process of minimizing the loss function in Gemp et al. (2024), while mitigating the high variance caused by the inner product of two estimated variables.

As mentioned in Appendix G, for the loss function in Gemp et al. (2024), it is difficult to backpropagate the gradient of the term $\nabla_{\boldsymbol{x}_i} u_i(\boldsymbol{x})$ in $\boldsymbol{F}_i^{\tau,\boldsymbol{x}}$ when solving real-world games. Therefore, in practice, we have to use the following loss function as a surrogate function for the loss in Gemp et al. (2024):

$$\hat{\mathcal{L}}_G^\tau(\boldsymbol{x}) = \sum_{i \in \mathcal{N}} \left\| sg[\boldsymbol{F}_i^{\boldsymbol{x}}] - sg[\overline{\boldsymbol{F}_i^{\boldsymbol{x}}}] + \tau \log \boldsymbol{x}_i - \tau \overline{\log \boldsymbol{x}_i} \right\|_2^2, \tag{7}$$

where $\boldsymbol{F}_i^{\boldsymbol{x}} = -\nabla_{\boldsymbol{x}_i} u_i(\boldsymbol{x})$. Then, consider the following variant of $\hat{\mathcal{L}}_G^\tau(\boldsymbol{x})$:

$$\mathcal{L}_0^\tau(\boldsymbol{x}) = \sum_{i \in \mathcal{N}} \|\text{sg}[\boldsymbol{F}_i^{\tau,\boldsymbol{x}}] - \text{sg}[\overline{\boldsymbol{F}_i^{\tau,\boldsymbol{x}}}] + \tau\boldsymbol{x}_i - \overline{\tau\boldsymbol{x}_i}\|_2^2.$$

Compared to $\hat{\mathcal{L}}_G^\tau(\boldsymbol{x})$, the key difference is that $\mathcal{L}_0^\tau(\boldsymbol{x})$ replaces the gradient backpropagation paths involving $\log \boldsymbol{x}_i$ with those involving $\boldsymbol{x}_i$. The reason for this modification is that the gradient of $\log \boldsymbol{x}_i$ is $1/\boldsymbol{x}_i$, which may become very large in some cases, leading to instability (as shown in Figure 18). This modification ensures that a global minimum of $\hat{\mathcal{L}}_G^\tau(\boldsymbol{x})$ is an NE of a regularization game with the utility function $\hat{u}_i^\tau(\boldsymbol{x})$. Formally, since $\nabla_{\boldsymbol{x}_i} \hat{u}_i^\tau(\boldsymbol{x}) = \boldsymbol{F}_i^{\tau,\boldsymbol{x}} + \tau\boldsymbol{x}_i = \nabla_{\boldsymbol{x}_i} u_i(\boldsymbol{x}) - \tau \log \boldsymbol{x}_i - \tau\boldsymbol{x}_i$ is concave over $\mathcal{X}_i$, from the analysis in Gemp et al. (2024), we have that $\boldsymbol{F}_i^{\tau,\boldsymbol{x}} + \tau\boldsymbol{x}_i = \overline{\boldsymbol{F}_i^{\tau,\boldsymbol{x}}} + \tau\overline{\boldsymbol{x}_i}, \forall i \in \mathcal{N}$ if and only if $\boldsymbol{x}$ is an NE of the regularization game with the utility function $\hat{u}_i^\tau(\boldsymbol{x})$. Then, consider the following loss function

$$\mathcal{L}_1^\tau(\boldsymbol{x}) = \sum_{i \in \mathcal{N}} \big(2\tau\langle sg[\boldsymbol{F}_i^{\tau,\boldsymbol{x}} - \langle \boldsymbol{F}_i^{\tau,\boldsymbol{x}}, \boldsymbol{x}_i^u\rangle \mathbf{1}], \boldsymbol{x}_i\rangle + \langle \tau\boldsymbol{x}_i, \tau\boldsymbol{x}_i\rangle - \langle \tau\overline{\boldsymbol{x}_i}, \tau\overline{\boldsymbol{x}_i}\rangle\big).$$

**Theorem H.1** (Proof is in Appendix H.1). *The first-order gradients of $\mathcal{L}_0^\tau(\boldsymbol{x})$ and $\mathcal{L}_1^\tau(\boldsymbol{x})$ are identical.*

As shown in Theorem H.1, we are surprised to find that the first-order gradient of $\mathcal{L}_0^\tau(\boldsymbol{x})$ is the same as the first-order gradient of $\mathcal{L}_1^\tau(\boldsymbol{x})$. It is evident that non-convex stochastic optimization techniques commonly used in ML require only first-order gradients as input, minimizing $\mathcal{L}_0^\tau(\boldsymbol{x})$ is equivalent to minimizing $\mathcal{L}_1^\tau(\boldsymbol{x})$. Moreover, $\mathcal{L}_1^\tau(\boldsymbol{x})$ does not involve the inner product of two estimated variables that leads to high variance compared to $\mathcal{L}_0^\tau(\boldsymbol{x})$.

It is evident that $2\tau\mathcal{L}_{NAL}^\tau(\boldsymbol{x})$ approximates $\mathcal{L}_1^\tau(\boldsymbol{x})$, as the difference between them is merely the term $\langle \tau\boldsymbol{x}_i, \tau\boldsymbol{x}_i\rangle - \langle \tau\overline{\boldsymbol{x}_i}, \tau\overline{\boldsymbol{x}_i}\rangle$. Therefore, we have that when employing commonly used non-convex stochastic optimization methods in ML, minimizing NAL approximates the process of minimizing $\mathcal{L}_0^\tau(\boldsymbol{x})$ while reducing high variance. In our experiments (Figure 18), the algorithm minimizing $\mathcal{L}_1^\tau(\boldsymbol{x})$ significantly outperforms the algorithm minimizing $\mathcal{L}_0^\tau(\boldsymbol{x})$ (in fact, we compare the algorithm minimizing $\mathcal{L}_0^\tau(\boldsymbol{x})/2\tau$ with the one minimizing $\mathcal{L}_1^\tau(\boldsymbol{x})/2\tau$, as the first-order gradients of $\mathcal{L}_0^\tau(\boldsymbol{x})/2\tau$ and $\mathcal{L}_1^\tau(\boldsymbol{x})/2\tau$ are on the same scale as the first-order gradient of NAL. This ensures a consistent comparison framework). This highlights the substantial impact of variance on convergence, as the performance of both algorithms would be consistent when using commonly used non-convex stochastic techniques without sampling. However, under the sampled play, the algorithm minimizing $\mathcal{L}_1^\tau(\boldsymbol{x})$ surpasses the one minimizing $\mathcal{L}_0^\tau$ by a significant margin. Furthermore, the algorithm minimizing NAL also achieves considerably better performance compared to the one minimizing $\mathcal{L}_0^\tau(\boldsymbol{x})$, further emphasizing the critical role of variance in convergence, given that NAL is merely an approximation of $\mathcal{L}_1^\tau(\boldsymbol{x})$. For detailed results, refer to Appendix J.

We employ NAL instead of $\mathcal{L}_1^\tau(\boldsymbol{x})$ in our paper because the computation of $\mathcal{L}_1^\tau(\boldsymbol{x})$ involves $\langle \tau\boldsymbol{x}_i, \tau\boldsymbol{x}_i\rangle - \langle \tau\overline{\boldsymbol{x}_i}, \tau\overline{\boldsymbol{x}_i}\rangle$. When the action size of the game is large, the entire $\boldsymbol{x}_i$ is unknown, and only the value $\boldsymbol{x}_i(a_i)$ on the sampled action $a_i \in \mathcal{A}_i$ is available. For instance, in games used for preference alignment in large language models (Munos et al., 2023; Wu et al., 2024; Ye et al., 2024), where actions consist of all possible natural language sentences, it is evident that $\boldsymbol{x}_i$ cannot be fully known. Note that for the computation of $\langle sg[\boldsymbol{F}_i^{\tau,\boldsymbol{x}} - \langle \boldsymbol{F}_i^{\tau,\boldsymbol{x}}, \boldsymbol{x}_i^u\rangle \mathbf{1}], \boldsymbol{x}_i\rangle$, due to the importance sampling, $sg[\boldsymbol{F}_i^{\tau,\boldsymbol{x}} - \langle \boldsymbol{F}_i^{\tau,\boldsymbol{x}}, \boldsymbol{x}_i^u\rangle \mathbf{1}]$ is zero except for the sampled actions, thus the entire $\boldsymbol{x}_i$ is not required. Our experimental results, as depicted in Figure 18, demonstrate that the algorithm minimizing NAL exhibits nearly identical performance to the one minimizing $\mathcal{L}_1^\tau(\boldsymbol{x})$. We hypothesize that this is primarily because the gradient $\nabla_{\boldsymbol{x}_i}(\langle \tau\boldsymbol{x}_i, \tau\boldsymbol{x}_i\rangle - \langle \tau\overline{\boldsymbol{x}_i}, \tau\overline{\boldsymbol{x}_i}\rangle) = 2\tau^2\boldsymbol{x}_i - 2\tau^2\overline{\boldsymbol{x}_i}$ (see derivation in Eq. (11), (12), and (13)) is relatively small compared to the value of $sg[\boldsymbol{F}_i^{\tau,\boldsymbol{x}} - \langle \boldsymbol{F}_i^{\tau,\boldsymbol{x}}, \boldsymbol{x}_i^u\rangle \mathbf{1}]$, due to the entropy term in $sg[\boldsymbol{F}_i^{\tau,\boldsymbol{x}} - \langle \boldsymbol{F}_i^{\tau,\boldsymbol{x}}, \boldsymbol{x}_i^u\rangle \mathbf{1}]$. See details in Appendix J.

### H.1. Proof of Theorem H.1

*Proof.* Now, we prove that the first-order gradients of the following two loss functions are identical:

$$\begin{aligned}
\mathcal{L}_0^\tau(\boldsymbol{x}) &= \sum_{i \in \mathcal{N}} \|sg[\boldsymbol{F}_i^{\tau,\boldsymbol{x}}] - sg[\overline{\boldsymbol{F}_i^{\tau,\boldsymbol{x}}}] + \tau\boldsymbol{x}_i - \overline{\tau\boldsymbol{x}_i}\|_2^2, \\
\mathcal{L}_1^\tau(\boldsymbol{x}) &= \sum_{i \in \mathcal{N}} (2\tau\langle sg[\boldsymbol{F}_i^{\tau,\boldsymbol{x}} - \langle \boldsymbol{F}_i^{\tau,\boldsymbol{x}}, \boldsymbol{x}_i^u\rangle \mathbf{1}], \boldsymbol{x}_i\rangle + \langle \tau\boldsymbol{x}_i, \tau\boldsymbol{x}_i\rangle - \langle \tau\overline{\boldsymbol{x}_i}, \tau\overline{\boldsymbol{x}_i}\rangle).
\end{aligned} \tag{8}$$

Expanding the first line in Eq. (8) ($\mathcal{L}_0^\tau(\boldsymbol{x})$), we have

$$
\begin{aligned}
&\|sg[\boldsymbol{F}_i^{\tau,\boldsymbol{x}}] + \tau\boldsymbol{x}_i - sg[\overline{\boldsymbol{F}_i^{\tau,\boldsymbol{x}}}] - \tau\overline{\boldsymbol{x}_i}\|_2^2 \\
&= \langle sg[\boldsymbol{F}_i^{\tau,\boldsymbol{x}}] + \tau\boldsymbol{x}_i, sg[\boldsymbol{F}_i^{\tau,\boldsymbol{x}}] + \tau\boldsymbol{x}_i \rangle - 2\left\langle sg[\boldsymbol{F}_i^{\tau,\boldsymbol{x}}] + \tau\boldsymbol{x}_i, sg[\overline{\boldsymbol{F}_i^{\tau,\boldsymbol{x}}}] + \tau\overline{\boldsymbol{x}_i} \right\rangle + \left\langle sg[\overline{\boldsymbol{F}_i^{\tau,\boldsymbol{x}}}] + \tau\overline{\boldsymbol{x}_i}, sg[\overline{\boldsymbol{F}_i^{\tau,\boldsymbol{x}}}] + \tau\overline{\boldsymbol{x}_i} \right\rangle.
\end{aligned}
\tag{9}
$$

**Lemma H.2.** *(Proof is in Appendix H.2) $\forall \boldsymbol{b}, \boldsymbol{c} \in \mathbb{R}^d$, we have*

$$
\langle \overline{\boldsymbol{b}}, \overline{\boldsymbol{c}} \rangle = \langle \overline{\boldsymbol{b}}, \boldsymbol{c} \rangle = \langle \boldsymbol{b}, \overline{\boldsymbol{c}} \rangle,
$$

*where $\overline{\boldsymbol{b}} = \frac{\sum_{j=1}^d \boldsymbol{b}(j)}{d}\mathbf{1}$ and $\overline{\boldsymbol{c}} = \frac{\sum_{j=1}^d \boldsymbol{c}(j)}{d}\mathbf{1}$.*

By substituting Lemma H.2 into Eq. (9) with $\boldsymbol{b} = \boldsymbol{c} = sg[\boldsymbol{F}_i^{\tau,\boldsymbol{x}}] + \tau\boldsymbol{x}_i$, we have

$$
\begin{aligned}
&\langle sg[\boldsymbol{F}_i^{\tau,\boldsymbol{x}}] + \tau\boldsymbol{x}_i, sg[\boldsymbol{F}_i^{\tau,\boldsymbol{x}}] + \tau\boldsymbol{x}_i \rangle - 2\left\langle sg[\boldsymbol{F}_i^{\tau,\boldsymbol{x}}] + \tau\boldsymbol{x}_i, sg[\overline{\boldsymbol{F}_i^{\tau,\boldsymbol{x}}}] + \tau\overline{\boldsymbol{x}_i} \right\rangle + \left\langle sg[\overline{\boldsymbol{F}_i^{\tau,\boldsymbol{x}}}] + \tau\overline{\boldsymbol{x}_i}, sg[\overline{\boldsymbol{F}_i^{\tau,\boldsymbol{x}}}] + \tau\overline{\boldsymbol{x}_i} \right\rangle \\
&= \langle sg[\boldsymbol{F}_i^{\tau,\boldsymbol{x}}] + \tau\boldsymbol{x}_i, sg[\boldsymbol{F}_i^{\tau,\boldsymbol{x}}] + \tau\boldsymbol{x}_i \rangle - 2\left\langle sg[\overline{\boldsymbol{F}_i^{\tau,\boldsymbol{x}}}] + \tau\overline{\boldsymbol{x}_i}, sg[\overline{\boldsymbol{F}_i^{\tau,\boldsymbol{x}}}] + \tau\overline{\boldsymbol{x}_i} \right\rangle + \left\langle sg[\overline{\boldsymbol{F}_i^{\tau,\boldsymbol{x}}}] + \tau\overline{\boldsymbol{x}_i}, sg[\overline{\boldsymbol{F}_i^{\tau,\boldsymbol{x}}}] + \tau\overline{\boldsymbol{x}_i} \right\rangle \\
&= \langle sg[\boldsymbol{F}_i^{\tau,\boldsymbol{x}}] + \tau\boldsymbol{x}_i, sg[\boldsymbol{F}_i^{\tau,\boldsymbol{x}}] + \tau\boldsymbol{x}_i \rangle - \left\langle sg[\overline{\boldsymbol{F}_i^{\tau,\boldsymbol{x}}}] + \tau\overline{\boldsymbol{x}_i}, sg[\overline{\boldsymbol{F}_i^{\tau,\boldsymbol{x}}}] + \tau\overline{\boldsymbol{x}_i} \right\rangle \\
&= \langle sg[\boldsymbol{F}_i^{\tau,\boldsymbol{x}}], sg[\boldsymbol{F}_i^{\tau,\boldsymbol{x}}] \rangle + 2\langle sg[\boldsymbol{F}_i^{\tau,\boldsymbol{x}}], \tau\boldsymbol{x}_i \rangle + \langle \tau\boldsymbol{x}_i, \tau\boldsymbol{x}_i \rangle - \langle sg[\overline{\boldsymbol{F}_i^{\tau,\boldsymbol{x}}}], sg[\overline{\boldsymbol{F}_i^{\tau,\boldsymbol{x}}}] \rangle - 2\langle sg[\overline{\boldsymbol{F}_i^{\tau,\boldsymbol{x}}}], \tau\overline{\boldsymbol{x}_i} \rangle - \langle \tau\overline{\boldsymbol{x}_i}, \tau\overline{\boldsymbol{x}_i} \rangle.
\end{aligned}
\tag{10}
$$

Combining Eq. (8), (9) , and (10), we get

$$
\mathcal{L}_0^\tau(\boldsymbol{x}) = \sum_{i \in \mathcal{N}} \left( \langle sg[\boldsymbol{F}_i^{\tau,\boldsymbol{x}}], sg[\boldsymbol{F}_i^{\tau,\boldsymbol{x}}] \rangle + 2\langle sg[\boldsymbol{F}_i^{\tau,\boldsymbol{x}}], \tau\boldsymbol{x}_i \rangle + \langle \tau\boldsymbol{x}_i, \tau\boldsymbol{x}_i \rangle - \langle sg[\overline{\boldsymbol{F}_i^{\tau,\boldsymbol{x}}}], sg[\overline{\boldsymbol{F}_i^{\tau,\boldsymbol{x}}}] \rangle - 2\langle sg[\overline{\boldsymbol{F}_i^{\tau,\boldsymbol{x}}}], \tau\overline{\boldsymbol{x}_i} \rangle - \langle \tau\overline{\boldsymbol{x}_i}, \tau\overline{\boldsymbol{x}_i} \rangle \right).
$$

Now, we present the first-order gradient of $\mathcal{L}_0^\tau(\boldsymbol{x})$ with respect to $\boldsymbol{x}_i(a_i)$. Formally, we have

$$
\begin{aligned}
&\nabla_{\boldsymbol{x}_i(a_i)} \mathcal{L}_0^\tau(\boldsymbol{x}) \\
&= \nabla_{\boldsymbol{x}_i(a_i)} \left( \sum_{j \in \mathcal{N}} \langle sg[\boldsymbol{F}_j^{\tau,\boldsymbol{x}}], sg[\boldsymbol{F}_j^{\tau,\boldsymbol{x}}] \rangle + 2\langle sg[\boldsymbol{F}_j^{\tau,\boldsymbol{x}}], \tau\boldsymbol{x}_j \rangle + \langle \tau\boldsymbol{x}_j, \tau\boldsymbol{x}_j \rangle - \langle sg[\overline{\boldsymbol{F}_j^{\tau,\boldsymbol{x}}}], sg[\overline{\boldsymbol{F}_j^{\tau,\boldsymbol{x}}}] \rangle - 2\langle sg[\overline{\boldsymbol{F}_j^{\tau,\boldsymbol{x}}}], \tau\overline{\boldsymbol{x}_j} \rangle - \langle \tau\overline{\boldsymbol{x}_j}, \tau\overline{\boldsymbol{x}_j} \rangle \right) \\
&= \nabla_{\boldsymbol{x}_i(a_i)} \left( 2\langle sg[\boldsymbol{F}_i^{\tau,\boldsymbol{x}}], \tau\boldsymbol{x}_i \rangle + \langle \tau\boldsymbol{x}_i, \tau\boldsymbol{x}_i \rangle - 2\langle sg[\overline{\boldsymbol{F}_i^{\tau,\boldsymbol{x}}}], \tau\overline{\boldsymbol{x}_i} \rangle - \langle \tau\overline{\boldsymbol{x}_i}, \tau\overline{\boldsymbol{x}_i} \rangle \right) \\
&= \nabla_{\boldsymbol{x}_i(a_i)} 2\langle sg[\boldsymbol{F}_i^{\tau,\boldsymbol{x}}], \tau\boldsymbol{x}_i \rangle + \nabla_{\boldsymbol{x}_i(a_i)} \langle \tau\boldsymbol{x}_i, \tau\boldsymbol{x}_i \rangle - \nabla_{\boldsymbol{x}_i(a_i)} 2\langle sg[\overline{\boldsymbol{F}_i^{\tau,\boldsymbol{x}}}], \tau\overline{\boldsymbol{x}_i} \rangle - \nabla_{\boldsymbol{x}_i(a_i)} \langle \tau\overline{\boldsymbol{x}_i}, \tau\overline{\boldsymbol{x}_i} \rangle \\
&= 2\tau sg[\boldsymbol{F}_i^{\tau,\boldsymbol{x}}(a_i)] + 2\tau^2 \boldsymbol{x}_i(a_i) - \nabla_{\boldsymbol{x}_i(a_i)} 2|\mathcal{A}_i| \tau \frac{\sum_{a_i' \in \mathcal{A}_i} sg[\boldsymbol{F}_i^{\tau,\boldsymbol{x}}(a_i')]}{|\mathcal{A}_i|} \frac{\sum_{a_i' \in \mathcal{A}_i} \boldsymbol{x}_i(a_i')}{|\mathcal{A}_i|} \\
&\quad - \nabla_{\boldsymbol{x}_i(a_i)} |\mathcal{A}_i| \tau^2 \frac{\sum_{a_i' \in \mathcal{A}_i} \boldsymbol{x}_i(a_i')}{|\mathcal{A}_i|} \frac{\sum_{a_i' \in \mathcal{A}_i} \boldsymbol{x}_i(a_i')}{|\mathcal{A}_i|} \\
&= 2\tau sg[\boldsymbol{F}_i^{\tau,\boldsymbol{x}}(a_i)] + 2\tau^2 \boldsymbol{x}_i(a_i) - 2\tau \frac{\sum_{a_i' \in \mathcal{A}_i} sg[\boldsymbol{F}_i^{\tau,\boldsymbol{x}}(a_i')]}{|\mathcal{A}_i|} - 2\tau^2 \frac{\sum_{a_i' \in \mathcal{A}_i} \boldsymbol{x}_i(a_i')}{|\mathcal{A}_i|},
\end{aligned}
\tag{11}
$$

where the last line comes from

$$
\begin{aligned}
\nabla_{\boldsymbol{x}_i(a_i)} |\mathcal{A}_i| \tau^2 \frac{\sum_{a_i' \in \mathcal{A}_i} \boldsymbol{x}_i(a_i')}{|\mathcal{A}_i|} \frac{\sum_{a_i' \in \mathcal{A}_i} \boldsymbol{x}_i(a_i')}{|\mathcal{A}_i|} &= \nabla_{\boldsymbol{x}_i(a_i)} \frac{\tau^2}{|\mathcal{A}_i|} \sum_{a_i' \in \mathcal{A}_i} \sum_{a'' \in \mathcal{A}_i} \boldsymbol{x}_i(a_i')\boldsymbol{x}_i(a'') \\
&= \frac{\tau^2}{|\mathcal{A}_i|} \left[ \nabla_{\boldsymbol{x}_i(a_i)} \boldsymbol{x}_i(a_i)\boldsymbol{x}_i(a_i) + \nabla_{\boldsymbol{x}_i(a_i)} \sum_{a_i' \in \mathcal{A}_i, a' \neq a} 2\boldsymbol{x}_i(a_i)\boldsymbol{x}_i(a_i') \right] \\
&= \frac{\tau^2}{|\mathcal{A}_i|} \left[ 2\boldsymbol{x}_i(a_i) + \sum_{a_i' \in \mathcal{A}_i, a_i' \neq a_i} 2\boldsymbol{x}_i(a_i') \right] = 2\tau^2 \frac{\sum_{a_i' \in \mathcal{A}_i} \boldsymbol{x}_i(a_i')}{|\mathcal{A}_i|}.
\end{aligned}
\tag{12}
$$

*Table 2.* The hyperparameters of the algorithm that learns an NE via minimizing NAL.

|  | $\eta$ | $\tau$ | $T_u$ | $\alpha$ | $\beta$ |
|---|---|---|---|---|---|
| Kuhn Poker | 0.0001 | 0.1 | 200 | 0.9 | 0.9 |
| Goofspiel | 0.0001 | 0.1 | 200 | 0.9 | 0.5 |
| Blotto | 0.0001 | 0.1 | 500 | 0.9 | 0.5 |
| Liar's Dice | 0.0001 | 0.1 | 500 | 0.9 | 0.5 |
| Bertrand Oligopoly | 0.0001 | 0.1 | 200 | 0.9 | 0.5 |
| Guess Two Thirds Ave | 0.0001 | 0.1 | 1000 | 0.9 | 0.5 |
| Minimum Effort | 0.0001 | 0.1 | 200 | 0.9 | 0.5 |
| Majority Voting | 0.0001 | 0.1 | 200 | 0.9 | 0.5 |

Therefore, from Eq. (11), we get

$$
\begin{aligned}
\nabla_{\boldsymbol{x}_i}\mathcal{L}_0^\tau(\boldsymbol{x}) &= 2\tau sg[\boldsymbol{F}_i^{\tau,\boldsymbol{x}}] + 2\tau^2\boldsymbol{x}_i - 2\tau sg[\overline{\boldsymbol{F}_i^{\tau,\boldsymbol{x}}}] - 2\tau^2\overline{\boldsymbol{x}_i} \\
&= 2\tau sg[\boldsymbol{F}_i^{\tau,\boldsymbol{x}} - \overline{\boldsymbol{F}_i^{\tau,\boldsymbol{x}}}] + 2\tau^2\boldsymbol{x}_i - 2\tau^2\overline{\boldsymbol{x}_i}.
\end{aligned}
\tag{13}
$$

For $\mathcal{L}_1^\tau(\boldsymbol{x}) = \sum_{i\in\mathcal{N}}(2\tau\langle sg[\boldsymbol{F}_i^{\tau,\boldsymbol{x}} - \langle\boldsymbol{F}_i^{\tau,\boldsymbol{x}}, \boldsymbol{x}_i^u\rangle\mathbf{1}], \boldsymbol{x}_i\rangle + \langle\tau\boldsymbol{x}_i, \tau\boldsymbol{x}_i\rangle - \langle\tau\overline{\boldsymbol{x}_i}, \tau\overline{\boldsymbol{x}_i}\rangle)$, obviously, we have

$$
\begin{aligned}
\nabla_{\boldsymbol{x}_i}\mathcal{L}_1^\tau(\boldsymbol{x}) &= 2\tau sg[\boldsymbol{F}_i^{\tau,\boldsymbol{x}} - \langle\boldsymbol{F}_i^{\tau,\boldsymbol{x}}, \boldsymbol{x}_i^u\rangle\mathbf{1}] + \nabla_{\boldsymbol{x}_i}(\langle\tau\boldsymbol{x}_i, \tau\boldsymbol{x}_i\rangle - \langle\tau\overline{\boldsymbol{x}_i}, \tau\overline{\boldsymbol{x}_i}\rangle) \\
&= 2\tau sg[\boldsymbol{F}_i^{\tau,\boldsymbol{x}} - \overline{\boldsymbol{F}_i^{\tau,\boldsymbol{x}}}] + 2\tau^2\boldsymbol{x}_i - 2\tau^2\overline{\boldsymbol{x}_i},
\end{aligned}
\tag{14}
$$

where the last line comes from the derivation of the first-order gradients of $\langle\tau\boldsymbol{x}_i, \tau\boldsymbol{x}_i\rangle$ and $\langle\tau\overline{\boldsymbol{x}_i}, \tau\overline{\boldsymbol{x}_i}\rangle$ in Eq. (11). By combining Eq. (13) and (14), we complete the proof. $\square$

### H.2. Proof of Lemma H.2

*Proof.* From the definition of $\langle\overline{b}, \overline{c}\rangle$, $\langle\overline{b}, c\rangle$ and $\langle b, \overline{c}\rangle$, we have

$$
\langle\overline{\boldsymbol{b}}, \overline{\boldsymbol{c}}\rangle = \langle\frac{\sum_{j=1}^d \boldsymbol{b}(j)}{d}\mathbf{1}, \frac{\sum_{j=1}^d \boldsymbol{c}(j)}{d}\mathbf{1}\rangle = d\frac{1}{d^2}\sum_{j=1}^d \boldsymbol{b}(j)\sum_{j'=1}^d \boldsymbol{c}(j') = \frac{1}{d}\sum_{j=1}^d \boldsymbol{b}(j)\sum_{j'=1}^d \boldsymbol{c}(j'),
$$

$$
\langle\overline{\boldsymbol{b}}, \boldsymbol{c}\rangle = \langle\frac{\sum_{j=1}^d \boldsymbol{b}(j)}{d}\mathbf{1}, \boldsymbol{c}\rangle = \frac{1}{d}\sum_{j=1}^d \boldsymbol{b}(j)\sum_{j'=1}^d \boldsymbol{c}(j'),
$$

$$
\langle\boldsymbol{b}, \overline{\boldsymbol{c}}\rangle = \langle\boldsymbol{b}, \frac{\sum_{j=1}^d \boldsymbol{c}(j)}{d}\mathbf{1}\rangle = \frac{1}{d}\sum_{j=1}^d \boldsymbol{b}(j)\sum_{j'=1}^d \boldsymbol{c}(j').
$$

Therefore, we have $\langle\overline{\boldsymbol{b}}, \overline{\boldsymbol{c}}\rangle = \langle\overline{\boldsymbol{b}}, \boldsymbol{c}\rangle = \langle\boldsymbol{b}, \overline{\boldsymbol{c}}\rangle$, which finishes the proof. $\square$

## I. The Hyperparameters Used in Experiments

The selected hyperparameters for algorithms that minimize NAL, the loss function in Gemp et al. (2024), ADI, and NashApr, are shown in Table 2, Table 3, Table 4, and Table 5, respectively (unless otherwise stated).

## J. Additional Experimental Results

**Results on differences between true and estimated values for NAL.** Firstly, as previously mentioned, the difference between true and estimated values for NAL is significantly smaller than other loss functions. To illustrate this difference more clearly, we present a detailed graph specifically for NAL, as shown in Figure 4.

*Table 3.* The hyperparameters of the algorithm that learns an NE via minimizing the loss function proposed by Gemp et al. (2024).

|  | $\eta$ | $\tau$ | $T_u$ | $\alpha$ | $\beta$ |
|---|---|---|---|---|---|
| Kuhn Poker | 0.00001 | 1 | 500 | 0.9 | 0.5 |
| Goofspiel | 0.00001 | 0.1 | 200 | 0.9 | 0.5 |
| Blotto | 0.00001 | 0.1 | 500 | 0.9 | 0.5 |
| Liar's Dice | 0.00001 | 0.1 | 500 | 0.9 | 0.5 |
| Bertrand Oligopoly | 0.00001 | 0.1 | 200 | 0.9 | 0.5 |
| Guess Two Thirds Ave | 0.00001 | 0.1 | 500 | 0.9 | 0.9 |
| Minimum Effort | 0.00001 | 0.1 | 200 | 0.9 | 0.9 |
| Majority Voting | 0.00001 | 0.1 | 1000 | 0.9 | 0.5 |

*Table 4.* The hyperparameters of the algorithm that learns an NE via minimizing ADI.

|  | $\eta$ | $\tau$ | $T_u$ | $\alpha$ | $\beta$ |
|---|---|---|---|---|---|
| Kuhn Poker | 0.00001 | 1 | 500 | 0.9 | 0.5 |
| Goofspiel | 0.0001 | 0.1 | 200 | 0.9 | 0.5 |
| Blotto | 0.00001 | 0.1 | 500 | 0.9 | 0.5 |
| Liar's Dice | 0.0001 | 0.1 | 500 | 0.9 | 0.5 |
| Bertrand Oligopoly | 0.00001 | 0.1 | 200 | 0.9 | 0.5 |
| Guess Two Thirds Ave | 0.0001 | 0.1 | 1000 | 0.9 | 0.5 |
| Minimum Effort | 0.00001 | 0.1 | 200 | 0.9 | 0.5 |
| Majority Voting | 0.00001 | 0.1 | 200 | 0.9 | 0.5 |

*Table 5.* The hyperparameters of the algorithm that learns an NE via minimizing NashApr.

|  | $\eta$ |
|---|---|
| Kuhn Poker | 0.0001 |
| Goofspiel | 0.00001 |
| Blotto | 0.0001 |
| Liar's Dice | 0.0001 |
| Bertrand Oligopoly | 0.00001 |
| Guess Two Thirds Ave | 0.0001 |
| Minimum Effort | 0.0001 |
| Majority Voting | 0.0001 |

**Results on the value of NAL.** We also present the value curves of NAL during the training process. Adam is employed as the optimizer. The parameter $\tau$ remains constant throughout the training because, as observed, continuously shrinking $\tau$ (as suggested in Algorithm 1) renders it excessively small, diminishing its impact. The experimental results are shown in Figure 5. Note that the absence of biased estimates in Goofspiel is an artifact of the logarithmic scaling of the y-axis, leading to visual distortion. In most cases, we observe that the values of NAL converge to zero, which aligns with an NE of the regularization game. However, it is important to emphasize that since an NE in NAL is only the zero point of the first-order gradient, the values of NAL may either exceed or fall below zero.

**Results on convergence rates with different optimizers.** The convergence results for RMSprop and SGD are shown in Figures 6 and 7, respectively. RMSprop shows minimal variation in empirical convergence compared to Adam, likely due to their similarities. In contrast, all other algorithms except ours, experience significant performance degradation with SGD. This is likely due to the considerable difference between SGD and momentum-based optimizers like Adam and RMSprop.

**Results on convergence rates of algorithm minimizing NashApr when the optimizer is SGD with larger learning rates in terms of epochs.** We observe that the convergence performance of NashApr is poor when the optimizer is SGD. Therefore, we also present the results when the loss function is NashApr, the optimizer is SGD, and the learning rate is increased by a factor of 10 and 100, respectively, compared to the learning rate of NashApr shown in Appendix I. The experimental results are illustrated in Figure 8 and Figure 9, corresponding to learning rates 10 and 100 times higher, respectively, while keeping the learning rates for algorithms using other loss functions unchanged. We observe that algorithms using NashApr as the loss function still perform poorly.

**Results on variances and differences between true and estimated values with different optimizers.** Now, we present the results of the variance in estimating the value of loss functions and the difference between the true and estimated values when RMSprop and SGD are used as optimizers. The variances in estimating the value of loss functions under RMSprop and SGD are shown in Figures 10 and 11, respectively. Similar to the case when Adam is used as the optimizer, our algorithm demonstrates the lowest variance. Additionally, the differences between the true and estimated loss values under RMSprop and SGD are presented in Figures 12 and 13, respectively. Again, consistent with the results using Adam, our algorithm exhibits the smallest difference.

**Results on convergence rates with different numbers of sampled instances.** Then, we investigate the empirical convergence rates of our algorithm and the algorithm minimizing the loss function proposed by Gemp et al. (2024) with varying numbers of sampled instances $S$ per iteration, using Adam as the optimizer. We focus on these two algorithms since they both minimize unbiased loss functions. The results are presented in Figure 14. We observe that as $S$ increases, our algorithm converges to a more and more accurate NE. In contrast, the algorithm minimizing the loss function proposed by Gemp et al. (2024) learns a more and more accurate NE in Liar's Dice, but fails to learn a more accurate NE in Kuhn Poker. These results demonstrate that finding an NE is easier by using our algorithm due to the lower variance.

**Results on sampling times with different sampling methods.** We also provide a comparison of the sampling time between our sampling method in Algorithm 1 with the sampling method used in Gemp et al. (2022) and Gemp et al. (2024), under the same number of sampled instances $S$. We conduct our tests on Liar's Dice because it has the largest number of actions for each player among the eight games evaluated in the experiments. The results are shown in Figure 15. We observe that the sampling time of the methods employed by Gemp et al. (2022) and Gemp et al. (2024) are at least 10,000 times greater than that of our sampling method. Specifically, when $S = 10$, which corresponds to the configuration used in our experiments (Section 5), our sampling method achieves a sampling time of approximately 0.004 seconds, while the methods from Gemp et al. (2022) and Gemp et al. (2024) require about 1590 seconds.

**Results on convergence rates with different sampling methods in terms of time.** Now, we compare the convergence rates of algorithms employing different sampling methods in terms of time. We focus on algorithms that minimize NAL or the loss function proposed by Gemp et al. (2024), the only known unbiased loss functions. We conduct experiments on Liar's Dice, which has the largest number of actions (2306) among all tested games. For the algorithms employing the sampling method outlined in Algorithm 1, we set $S = 100$ to learn a sufficiently accurate approximation of NE, while maintaining all other parameters as described in Section 5. For the algorithms that adopt the sampling method utilized in Gemp et al. (2022) and Gemp et al. (2024), We reduce $T$ and $T_u$ in Section 5 by a factor of 100 since the sampling time associated with the method in Gemp et al. (2022) and Gemp et al. (2024) is excessively large. In addition, we employ two different settings of the value of $S$, *e.g.*, $S = 2$ and $S = 100$. All other settings remain unchanged from Section 5. Notably, when employing the loss function from Gemp et al. (2024), $\boldsymbol{F}_i^{\boldsymbol{x}^{\boldsymbol{\theta}}}$ also contributes to the gradient backpropagation, consistent with the settings in the original paper by Gemp et al. (2024). The experimental results are presented in Figure 16. We observe that both algorithms minimizing NAL exhibit a faster convergence rate than minimizing the loss function proposed by Gemp et al. (2024). More importantly, we find that the wall times of the algorithms utilizing the sampling methods from Gemp et al. (2022) and Gemp et al. (2024) are significantly greater than those of the algorithms employing the sampling method in Algorithm 1. This suggests that, when addressing real-world games, the sampling method in Algorithm 1 is more advantageous, as the action space in real-world scenarios vastly exceeds the 2306 actions present in Liar's Dice.

**Results on convergence rates of algorithms employing the sampling method used in Gemp et al. (2022) and Gemp et al. (2024) in terms of epochs.** We compare the convergence rates of the algorithms when they employ the sampling method presented in Gemp et al. (2022) and Gemp et al. (2024) across all games, measured in terms of epochs. It is important to note that when using the sampling method from Gemp et al. (2022) and Gemp et al. (2024), the runtime of the algorithms is primarily determined by the sampling time. Therefore, the runtime difference between algorithms with the same number of epochs is negligible, which implies that analyzing the convergence rates in terms of epochs is sufficient to

reflect the convergence rates in terms of runtime. As shown in Figure 16, we focus on algorithms that minimize NAL and the loss function proposed by Gemp et al. (2024). We reduce $T$ and $T_u$ in Section 5 by a factor of 100, and set $S$ to 2 instead of 10, as the sampling time associated with the methods in Gemp et al. (2022) and Gemp et al. (2024) is excessively large. All other settings remain unchanged from Section 5. The experimental results are presented in Figure 17. In alignment with the findings in Section 5, we observe that the algorithm minimizing NAL exhibits a significantly superior convergence rate compared to the algorithm minimizing the loss function proposed by Gemp et al. (2024). More critically, in numerous games, the latter algorithm fails to learn a sufficiently accurate NE. Conversely, the algorithm minimizing NAL successfully learns an accurate NE in nearly all games, characterized by exploitability approaching zero.

**Results on convergence rates of $\mathcal{L}_0^\tau(\boldsymbol{x})$ and $\mathcal{L}_1^\tau(\boldsymbol{x})$.** As shown in Appendix H, it is clear that (i) the first-order gradient of $\mathcal{L}_0^\tau(\boldsymbol{x})$ (a variant of the loss function in Gemp et al. (2024)) is identical to that of $\mathcal{L}_1^\tau(\boldsymbol{x})$, and (ii) NAL is an approximation of $\mathcal{L}_0^\tau(\boldsymbol{x})/2\tau$. We evaluate the convergence rates of algorithms minimizing $\mathcal{L}_0^\tau(\boldsymbol{x})/2\tau$ or $\mathcal{L}_1^\tau(\boldsymbol{x})/2\tau$. Note that the term $2\tau$ from that NAL is an approximation of $\mathcal{L}_1^\tau(\boldsymbol{x})/2\tau$. For convenience, throughout the rest of this paper, we refer to algorithms minimizing $\mathcal{L}_0^\tau(\boldsymbol{x})$ ($\mathcal{L}_1^\tau(\boldsymbol{x})$) as those minimizing $\mathcal{L}_0^\tau(\boldsymbol{x})/2\tau$ ($\mathcal{L}_1^\tau(\boldsymbol{x})/2\tau$), respectively. For both algorithms, we use the parameters that minimize NAL. The experimental results are shown in Figure 18. The algorithm minimizing $\mathcal{L}_1^\tau(\boldsymbol{x})$ significantly outperforms the algorithm minimizing $\mathcal{L}_0^\tau(\boldsymbol{x})$. Since the first-order gradients of $\mathcal{L}_0^\tau(\boldsymbol{x})$ and $\mathcal{L}_1^\tau(\boldsymbol{x})$ are identical, in scenarios where the entire payoff matrix is known (i.e., no sampling required), the performance of both algorithms should be consistent when using standard non-convex stochastic methods, such as SGD. However, as shown in Figure 18, under the sampled play, the algorithm minimizing $\mathcal{L}_1^\tau(\boldsymbol{x})$ significantly outperforms the one minimizing $\mathcal{L}_0^\tau(\boldsymbol{x})$, highlighting the substantial impact of variance on convergence. The results also show that the algorithm minimizing NAL outperforms the one minimizing $\mathcal{L}_0^\tau(\boldsymbol{x})$ by a large margin, further demonstrating the strong influence of variance on convergence, as NAL is merely an approximation of $\mathcal{L}_1^\tau(\boldsymbol{x})$. Furthermore, the performance of the algorithms minimizing NAL or $\mathcal{L}_1^\tau(\boldsymbol{x})$ is nearly identical across all games. We hypothesize that this phenomenon arises primarily because the term $\nabla_{\boldsymbol{x}_i}(\langle\tau\boldsymbol{x}_i, \tau\boldsymbol{x}_i\rangle - \langle\tau\overline{\boldsymbol{x}_i}, \tau\overline{\boldsymbol{x}_i}\rangle) = 2\tau^2\boldsymbol{x}_i - 2\tau^2\overline{\boldsymbol{x}_i}$ is relatively small in magnitude compared to the term $sg[\boldsymbol{F}_i^{\tau,\boldsymbol{x}} - \langle\boldsymbol{F}_i^{\tau,\boldsymbol{x}}, \boldsymbol{x}_i^u\rangle\mathbf{1}]$ due to the influence of the entropy term embedded within $sg[\boldsymbol{F}_i^{\tau,\boldsymbol{x}} - \langle\boldsymbol{F}_i^{\tau,\boldsymbol{x}}, \boldsymbol{x}_i^u\rangle\mathbf{1}]$.

**Results on convergence rates with different neural network structures.** As mentioned in the main text, we evaluate the performance of different algorithms across various network architectures. Specifically, we replace the Softmax function with Sparsemax (Martins & Astudillo, 2016). The experimental results are presented in Figure 19. Our algorithm demonstrates the fastest convergence rate. Its convergence behavior remains stable, while the convergence rates of the other algorithms experience significant degradation. In particular, the algorithm minimizing the loss function in Gemp et al. (2024) exhibits output instability, often producing NaN values in numerous games, which also explains why the exploitability curve of the algorithm is incomplete in many games. This instability is likely due to excessively high variance. Moreover, the algorithms minimizing ADI or NashApr fail to converge in all games.

**Results on convergence rates of NAL with or without $\langle\boldsymbol{F}_i^{\tau,\boldsymbol{x}}, \hat{\boldsymbol{x}}_i\rangle\mathbf{1}$.** As mentioned in the main text, we analyze the convergence behavior of NAL when $\langle\boldsymbol{F}_i^{\tau,\boldsymbol{x}}, \hat{\boldsymbol{x}}_i\rangle\mathbf{1}$ is omitted. The case where NAL does not include $\langle\boldsymbol{F}_i^{\tau,\boldsymbol{x}}, \hat{\boldsymbol{x}}_i\rangle\mathbf{1}$ can also be interpreted as $\hat{\boldsymbol{x}}_i = 0$. In this scenario, NE is no longer a zero point of the first-order gradient of NAL, as the gradient $\boldsymbol{F}_i^{\tau,\boldsymbol{x}} - \langle\boldsymbol{F}_i^{\tau,\boldsymbol{x}}, \hat{\boldsymbol{x}}_i\rangle\mathbf{1}$ does not vanish at NE. Consequently, parameter updates induce a shift in strategy. We evaluate the performance across various values of $S$ ($S = 2, 10, 100, \infty$), where $S = \infty$ corresponds to the case without sampling. That is, for $S = \infty$, the variance introduced by sampling is zero. The experimental results are presented in Figure 20 and Figure 21, corresponding to Softmax and Sparsemax activation functions, respectively, in the final network layer. Note that the minimum value of $S$ is 2. For $S = 1$, it is impossible to estimate $\boldsymbol{F}_i^{\tau,\boldsymbol{x}} - \langle\boldsymbol{F}_i^{\tau,\boldsymbol{x}}, \hat{\boldsymbol{x}}_i\rangle\mathbf{1}$ using Algorithm 1, as the estimated value of $\boldsymbol{F}_i^{\tau,\boldsymbol{x}} - \langle\boldsymbol{F}_i^{\tau,\boldsymbol{x}}, \hat{\boldsymbol{x}}_i\rangle\mathbf{1}$ in this case will always be $\mathbf{0}$ (since $r_i^s - v_i = 0$).

We observe that when the activation function is Softmax and $S = \infty$, using NAL with $\hat{\boldsymbol{x}}_i = 0$ as the loss function performs equivalently to the standard NAL. This equivalence arises because the Softmax function normalizes outputs (i.e., for any $\boldsymbol{z} > 0$, it produces $\boldsymbol{z}/\text{sum}(\boldsymbol{z})$, constrained within the simplex), leading to identical parameter updates under $\boldsymbol{F}_i^{\tau,\boldsymbol{x}} - \langle\boldsymbol{F}_i^{\tau,\boldsymbol{x}}, \hat{\boldsymbol{x}}_i\rangle\mathbf{1}$ and $\boldsymbol{F}_i^{\tau,\boldsymbol{x}}$. However, when $S < \infty$, using NAL with $\hat{\boldsymbol{x}}_i = 0$ introduces higher variance compared to the standard NAL, resulting in degraded performance. This is further evidenced by the increasing performance gap between using $\boldsymbol{F}_i^{\tau,\boldsymbol{x}} - \langle\boldsymbol{F}_i^{\tau,\boldsymbol{x}}, \hat{\boldsymbol{x}}_i\rangle\mathbf{1}$ and $\boldsymbol{F}_i^{\tau,\boldsymbol{x}}$ as $S$ decreases. For Sparsemax activation, the standard NAL significantly outperforms NAL with $\hat{\boldsymbol{x}}_i = 0$. Notably, we do not observe convergence for NAL with $\hat{\boldsymbol{x}}_i = 0$ under Sparsemax. This observation suggests that the convergence of NAL with $\hat{\boldsymbol{x}}_i = 0$ when using the Softmax activation is likely a consequence of the identical parameter updates induced by $\boldsymbol{F}_i^{\tau,\boldsymbol{x}} - \langle\boldsymbol{F}_i^{\tau,\boldsymbol{x}}, \hat{\boldsymbol{x}}_i\rangle\mathbf{1}$ and $\boldsymbol{F}_i^{\tau,\boldsymbol{x}}$.

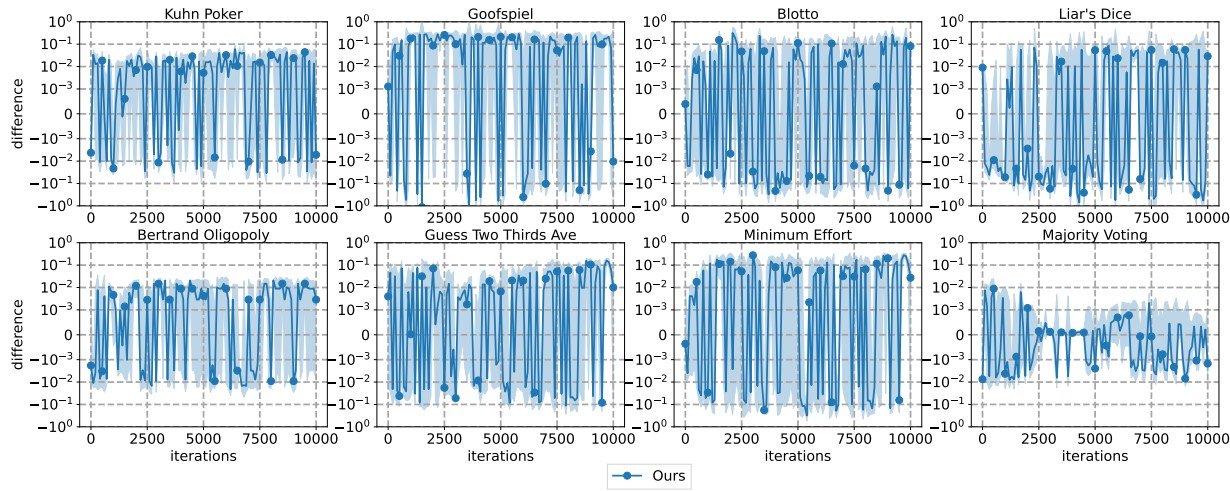

*Figure 4.* Difference between the true value and the estimated value of NAL when the optimizer is Adam.

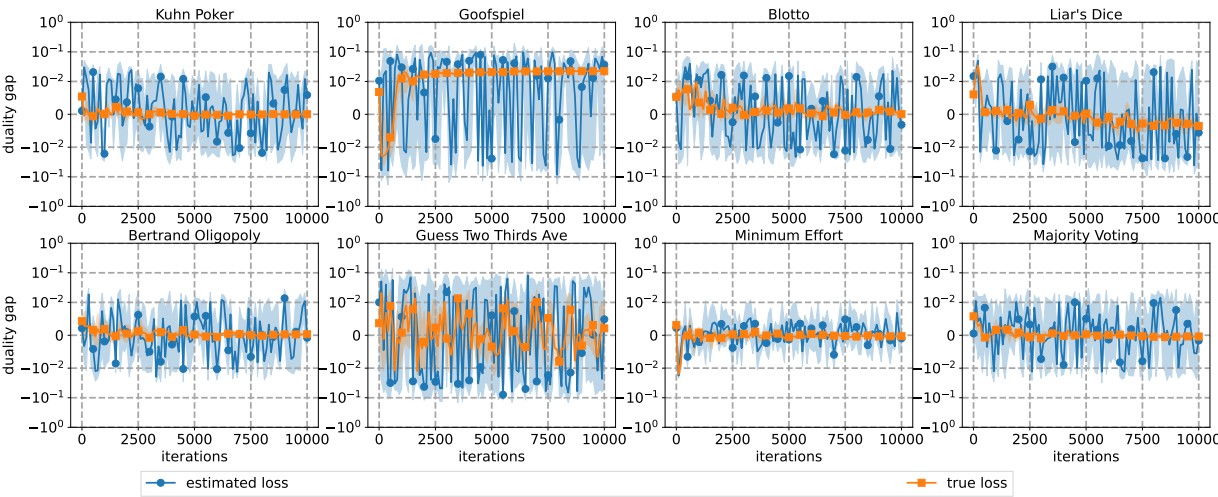

*Figure 5.* Value curves of NAL during the training. The optimizer is Adam. The parameter $\tau = 0.1$ (from the hyperparameters in Table 2) remains constant. Note that the absence of biased estimates in Goofspiel is an artifact of the logarithmic scaling of the y-axis, leading to a visual distortion.

**Results on convergence rates of NAL with different values of $\epsilon$.** To enhance the robustness of our results, we conduct experiments with various $\epsilon$ values (0, 0.1, 0.5, and 0.9), as illustrated in Figure 22. Our algorithm consistently surpasses the performance of baselines at all tested $\epsilon$ values. This consistency further confirms that the variance reduction facilitated by our loss function contributes to an accelerated convergence rate.

**Results on convergence rates when the strategy is represented using a real vector.** As highlighted in Appendix B, we utilize a DNN due to its ability to approximate arbitrary non-linear functions. This capability allows for the identification of complex equilibrium strategies that simpler representations, such as real vectors, may fail to capture. In Figure 23, the results are displayed with the strategy represented by a real vector. Although all algorithms show varying levels of performance degradation, our algorithm consistently outperforms the others.

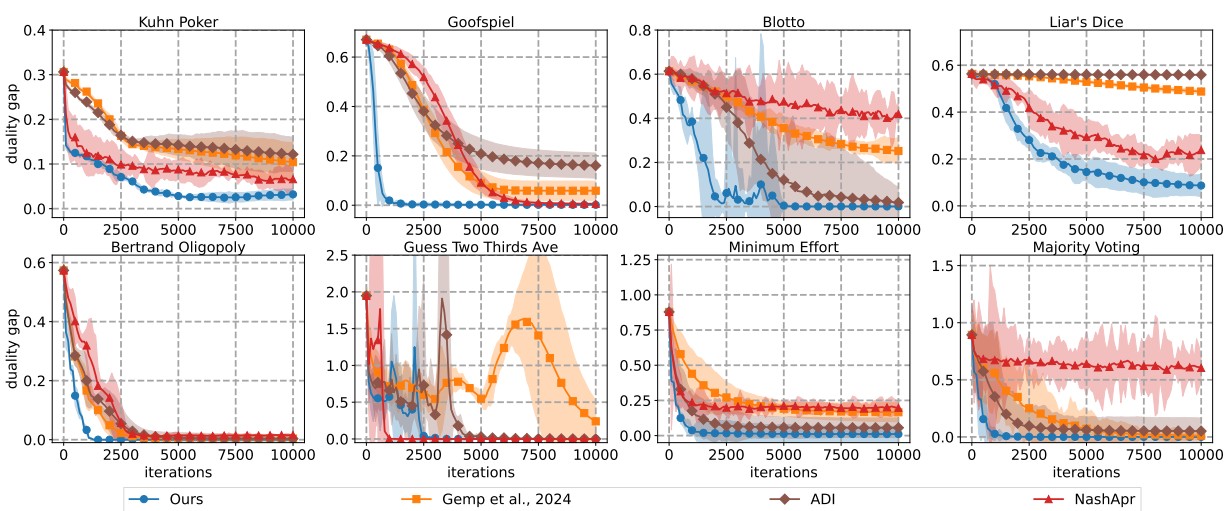

*Figure 6.* Empirical convergence rates of tested algorithms when the optimizer is RMSprop.

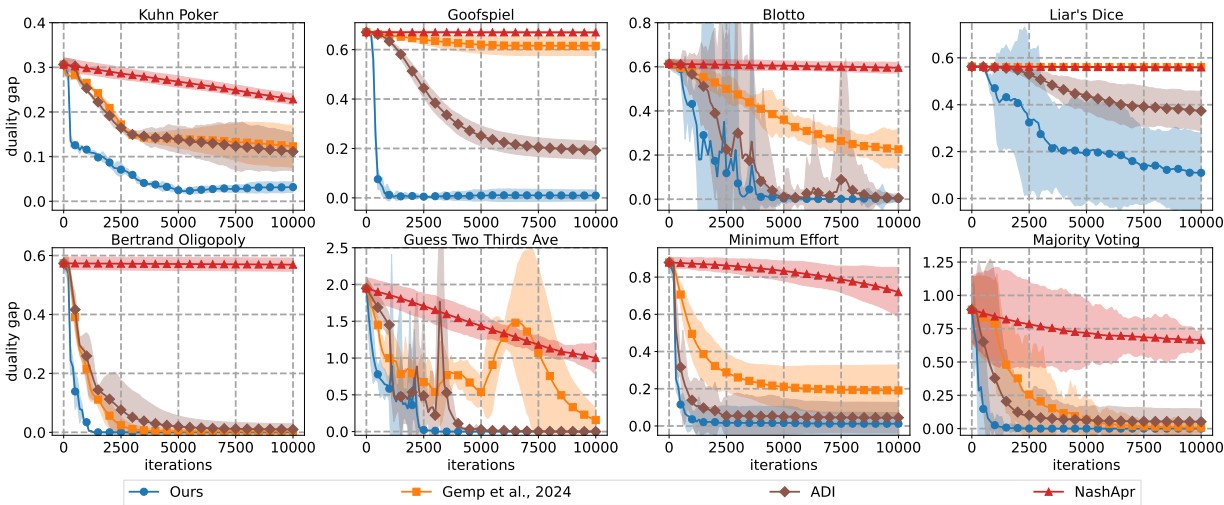

*Figure 7.* Empirical convergence rates of tested algorithms when the optimizer is SGD.

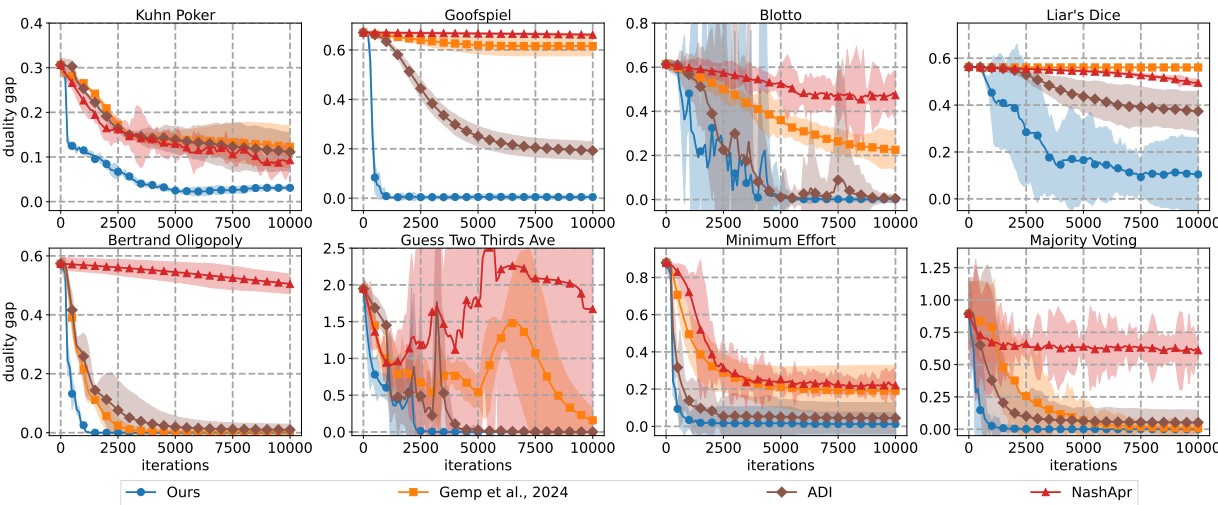

*Figure 8.* Empirical convergence rates of tested algorithms when the optimizer is SGD with 10 times larger learning rate for NashApr.

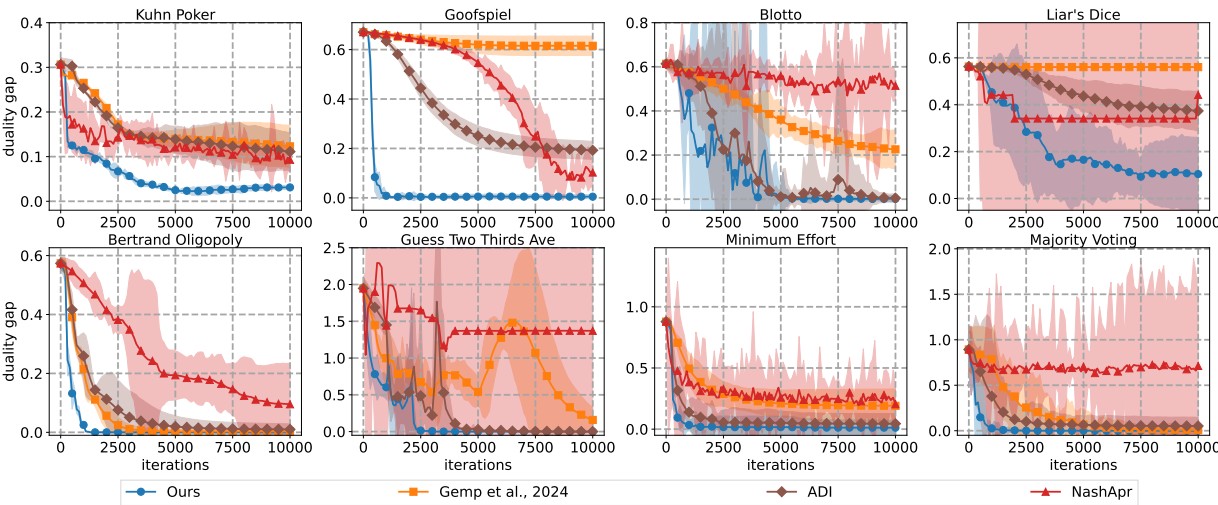

*Figure 9.* Empirical convergence rates of tested algorithms when the optimizer is SGD with 100 times larger learning rate for NashApr.

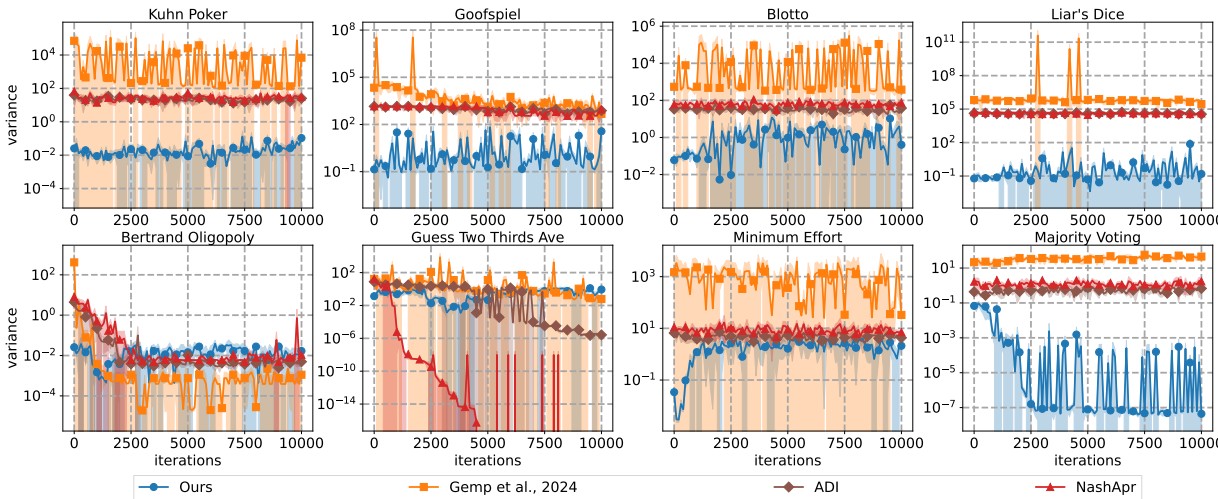

Figure 10. Variances observed in estimating the value of loss functions used by different algorithms when the optimizer is RMSprop.

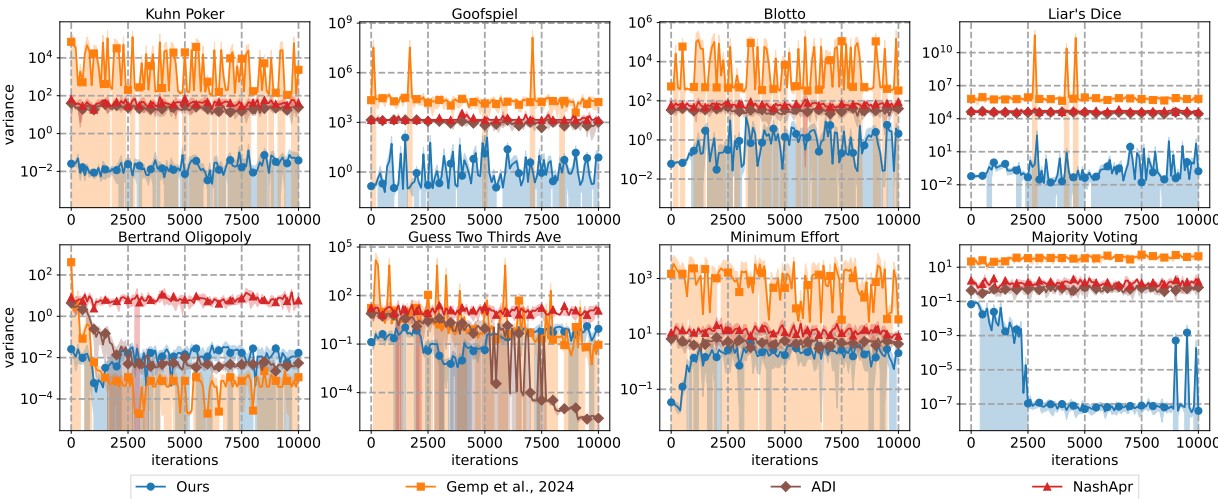

Figure 11. Variances observed in estimating the value of loss functions used by different algorithms when the optimizer is SGD.

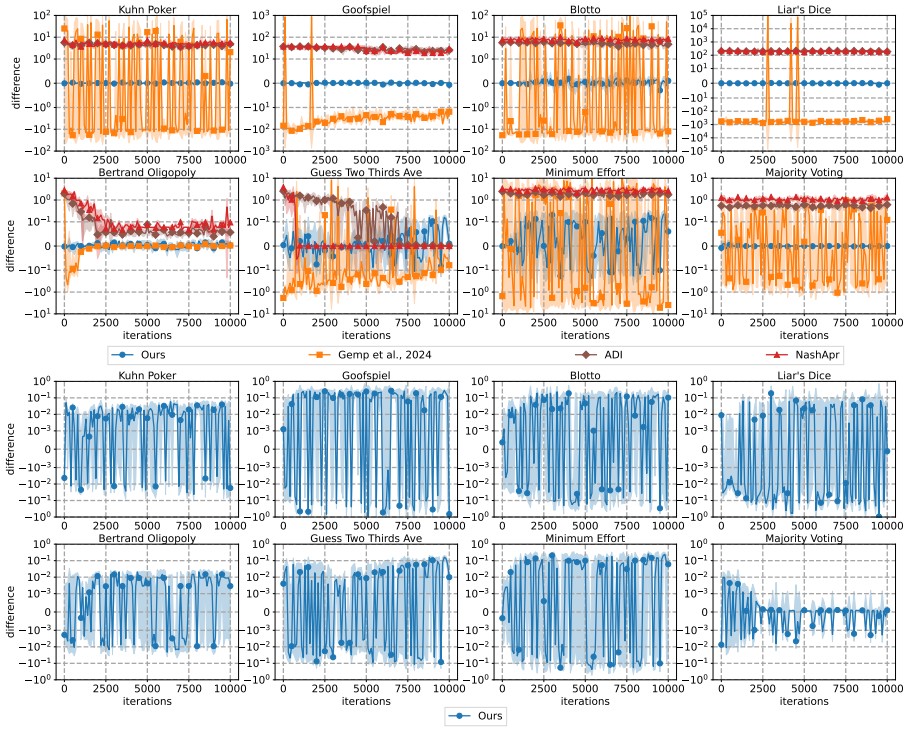

*Figure 12.* Difference between the true value and the estimated value of loss functions when the optimizer is RMSprop.

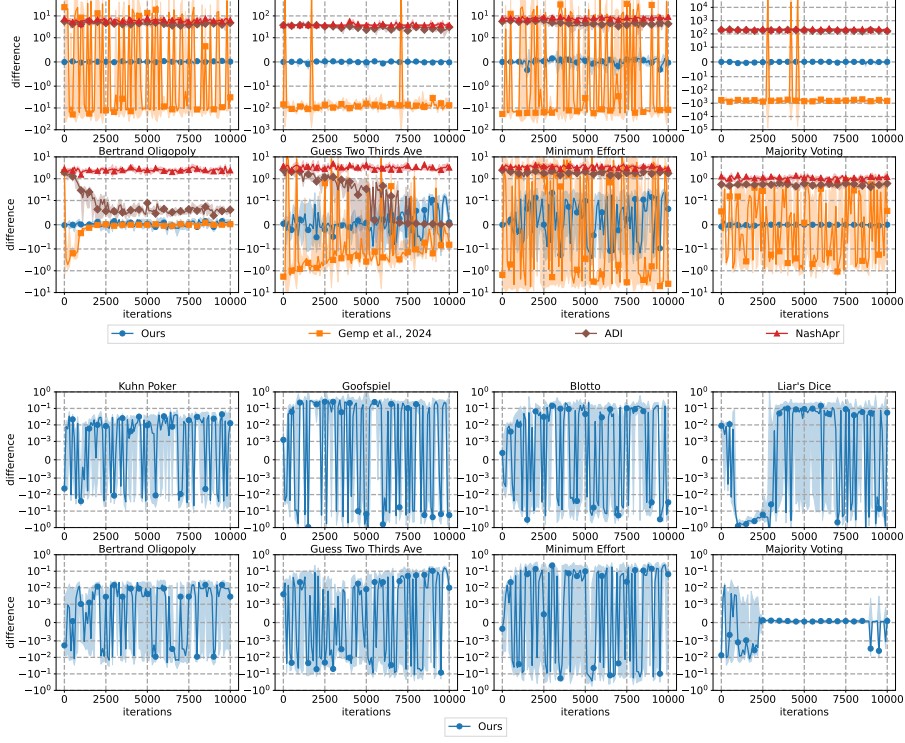

*Figure 13.* Difference between the true value and the estimated value of loss functions when the optimizer is SGD.

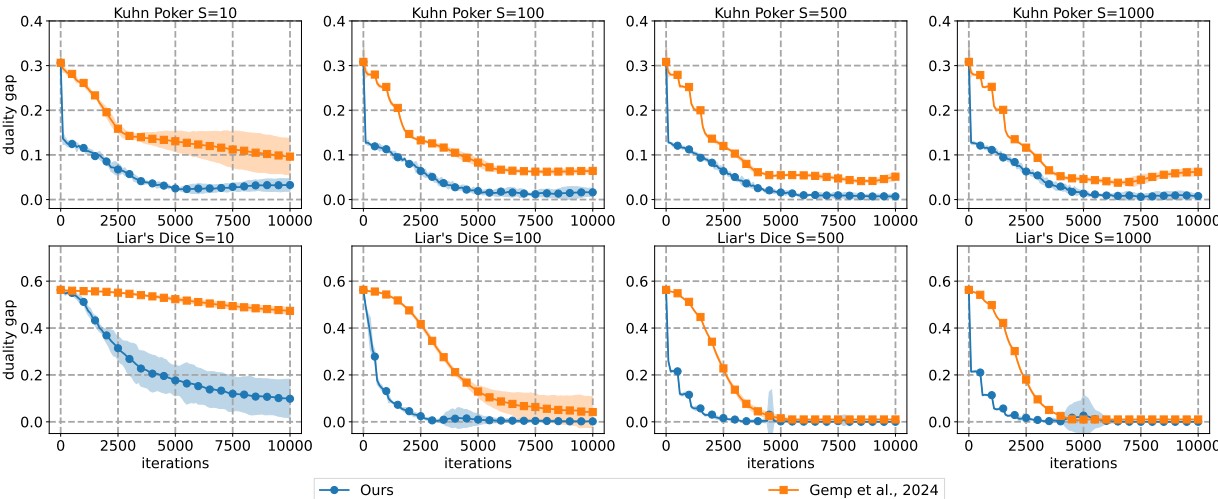

*Figure 14.* Empirical convergence rates of our algorithm, as well as the algorithm proposed by Gemp et al. (2024), with varying numbers of sampled instances $S$ at per iteration, are evaluated when the optimizer is Adam.

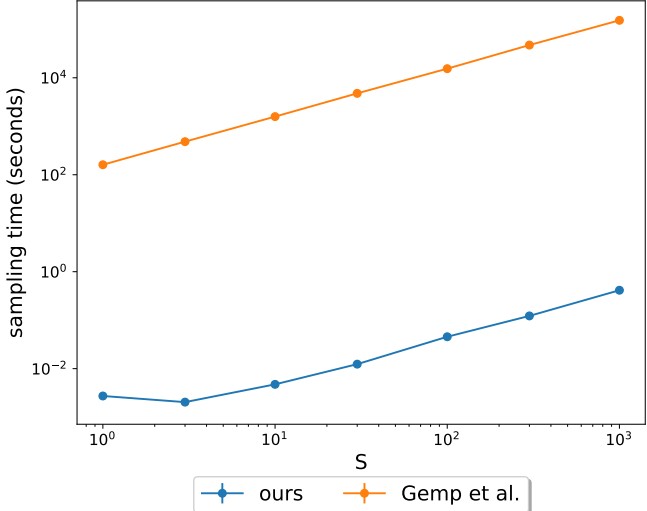

*Figure 15.* Comparison of the sampling times of our sampling method, shown in Algorithm 1, with the sampling method used in Gemp et al. (2022) and Gemp et al. (2024) for various values of the number $S$ of the sampled instance in Liar's Dice. We conduct our tests on Liar's Dice because it has the largest number of actions for each player among the eight games evaluated in the experiments. For each $S$, we run four seeds and report the average sampling times.

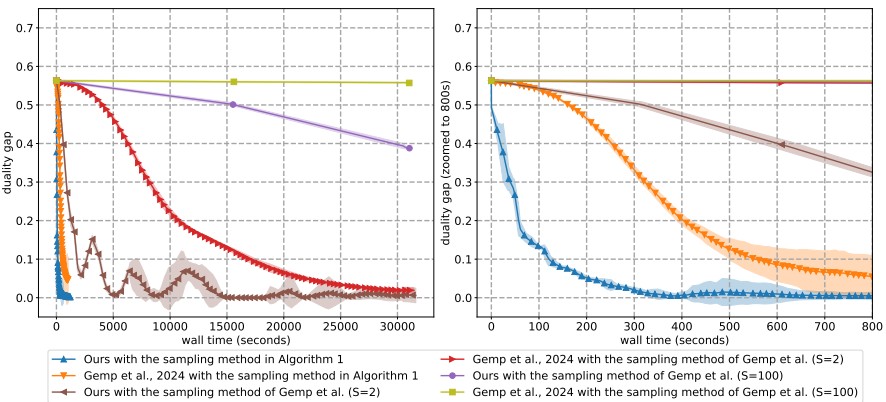

*Figure 16.* Empirical convergence rates of the algorithms utilizing various sampling methods in Liar's Dice. The x-axis represents the wall time. For algorithms that employ the sampling method outlined in Algorithm 1, the parameter $S$ is set to 100 to ensure the learning of a sufficiently accurate NE. For algorithms that employ the sampling method used in Gemp et al. (2022) and Gemp et al. (2024), we reduce the $T$ and $T_u$ in Section 5 by a factor of 100, as the sampling method used in Gemp et al. (2022) and Gemp et al. (2024) results in excessively higher sampling times for each instance than that of the sampling method in Algorithm 1, which is used in Section 5. The remaining hyperparameters for each algorithms remain consistent with those used in Section 5. The graph on the right is a scaled version of the one on the left.

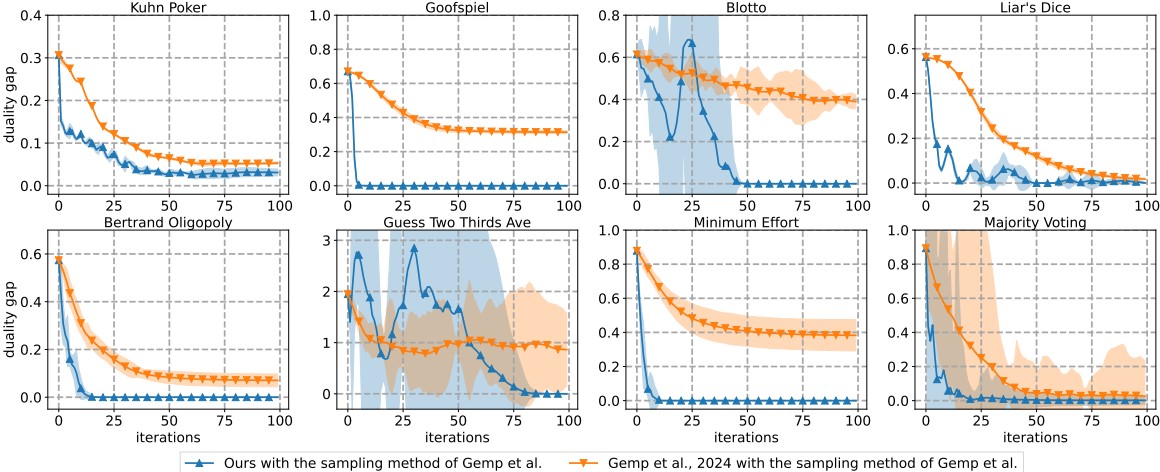

*Figure 17.* Empirical convergence rates of the algorithms that employ the sampling method used in Gemp et al. (2022) and Gemp et al. (2024). We reduce the $T$ and $T_u$ in Section 5 by a factor of 100, and set $S$ as 2 rather than 10 in Section 5. The remaining hyperparameters remain consistent with those used in Section 5.

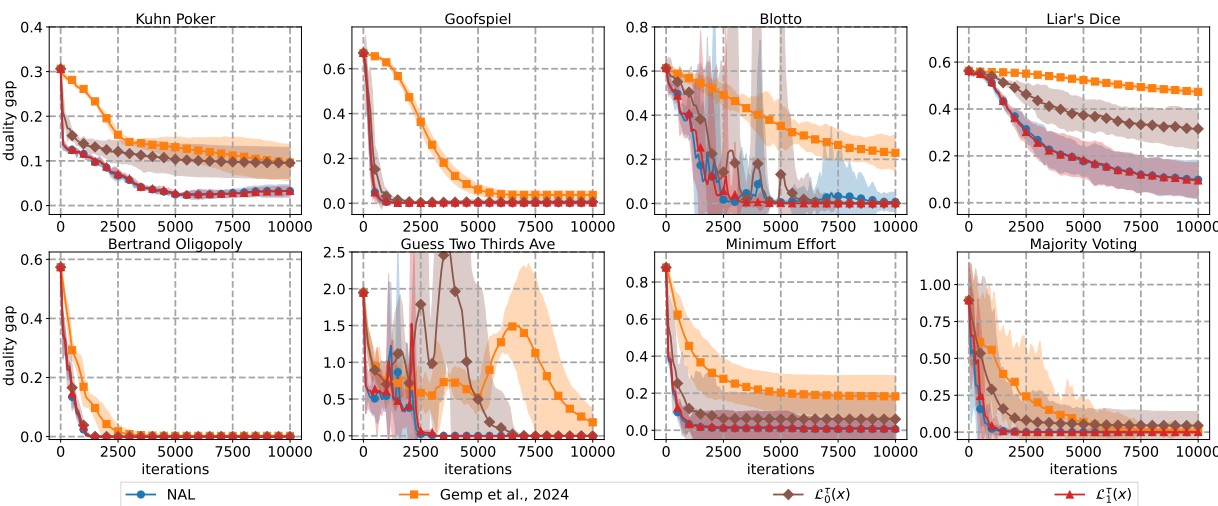

*Figure 18.* Empirical convergence rates of the algorithms that minimize $\mathcal{L}_0^\tau(\boldsymbol{x})$ or $\mathcal{L}_1^\tau(\boldsymbol{x})$ when the optimizer is Adam. All hyperparameters remain consistent with those used in Section 5.

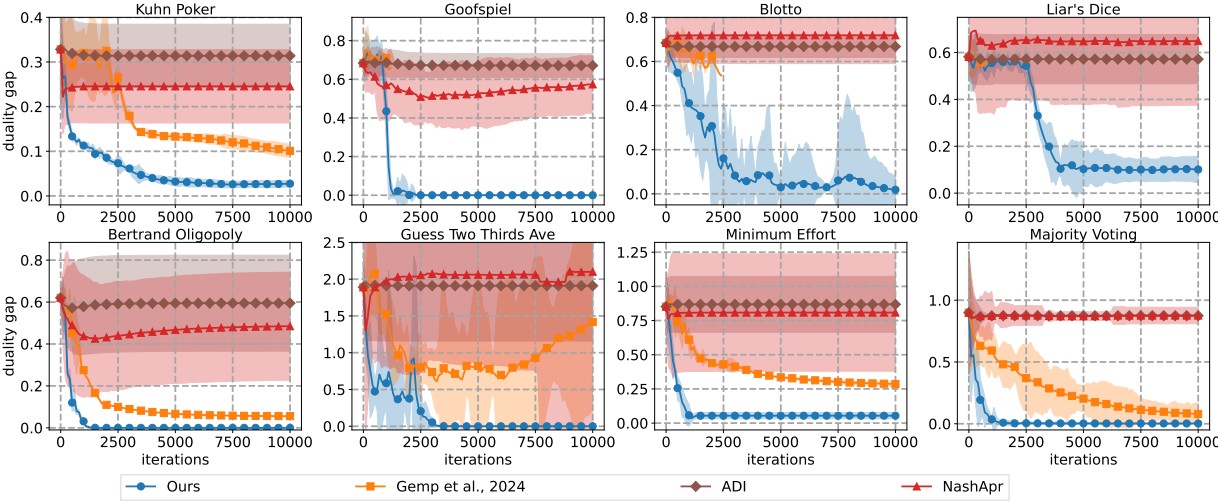

*Figure 19.* Empirical convergence rates of the algorithms when the Softmax function is replaced by Sparsemax. All hyperparameters remain consistent with those used in Section 5. The incompleteness of the exploitability curve of the algorithm is incomplete is due to that the network will output NaN value.

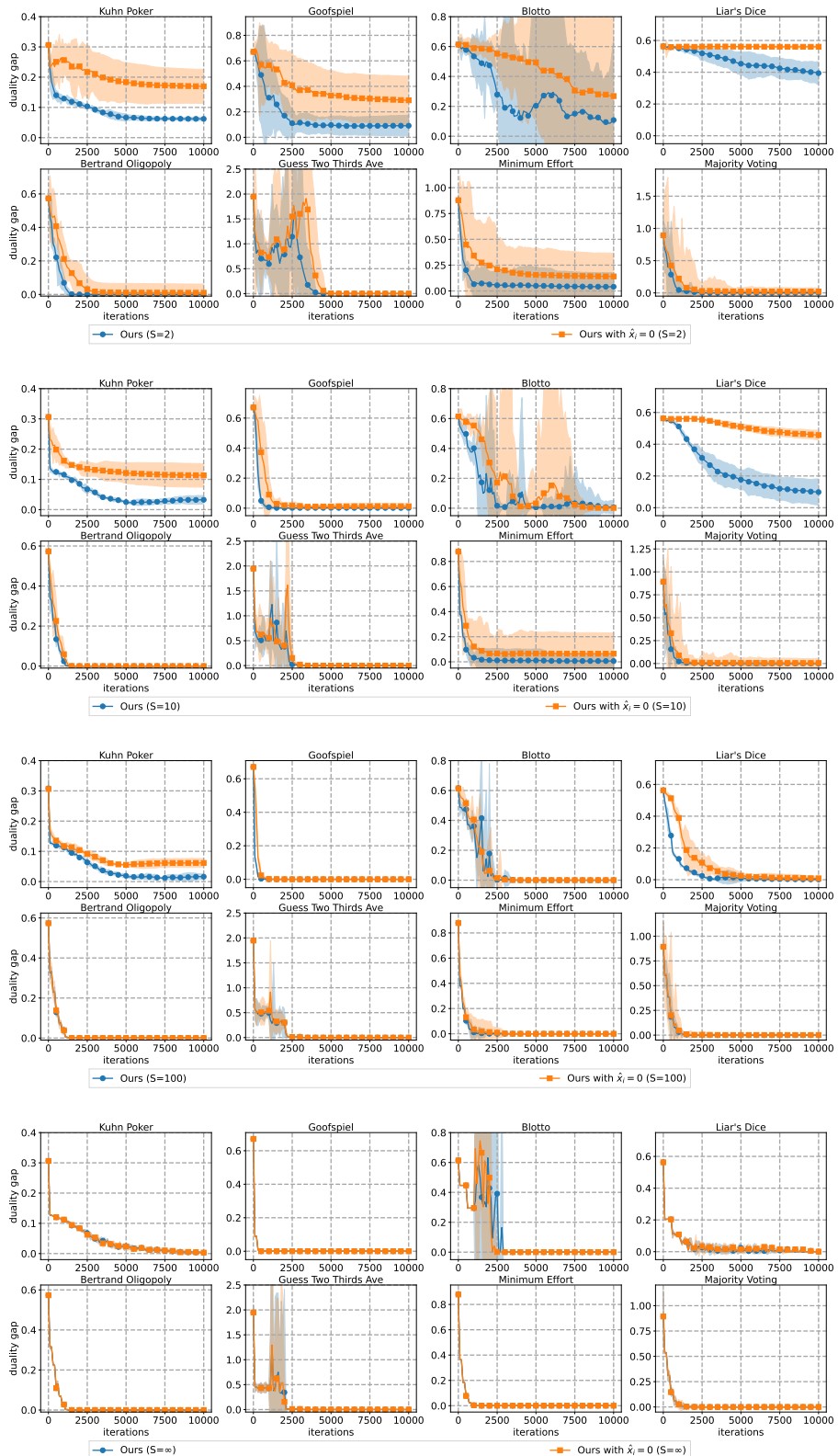

*Figure 20.* Empirical convergence rates of the algorithms that minimize NAL and NAL with $\hat{x}_i = 0$, respectively, when the optimizer is Adam. The notation $S = \infty$ denotes the scenario where sampling is not used.

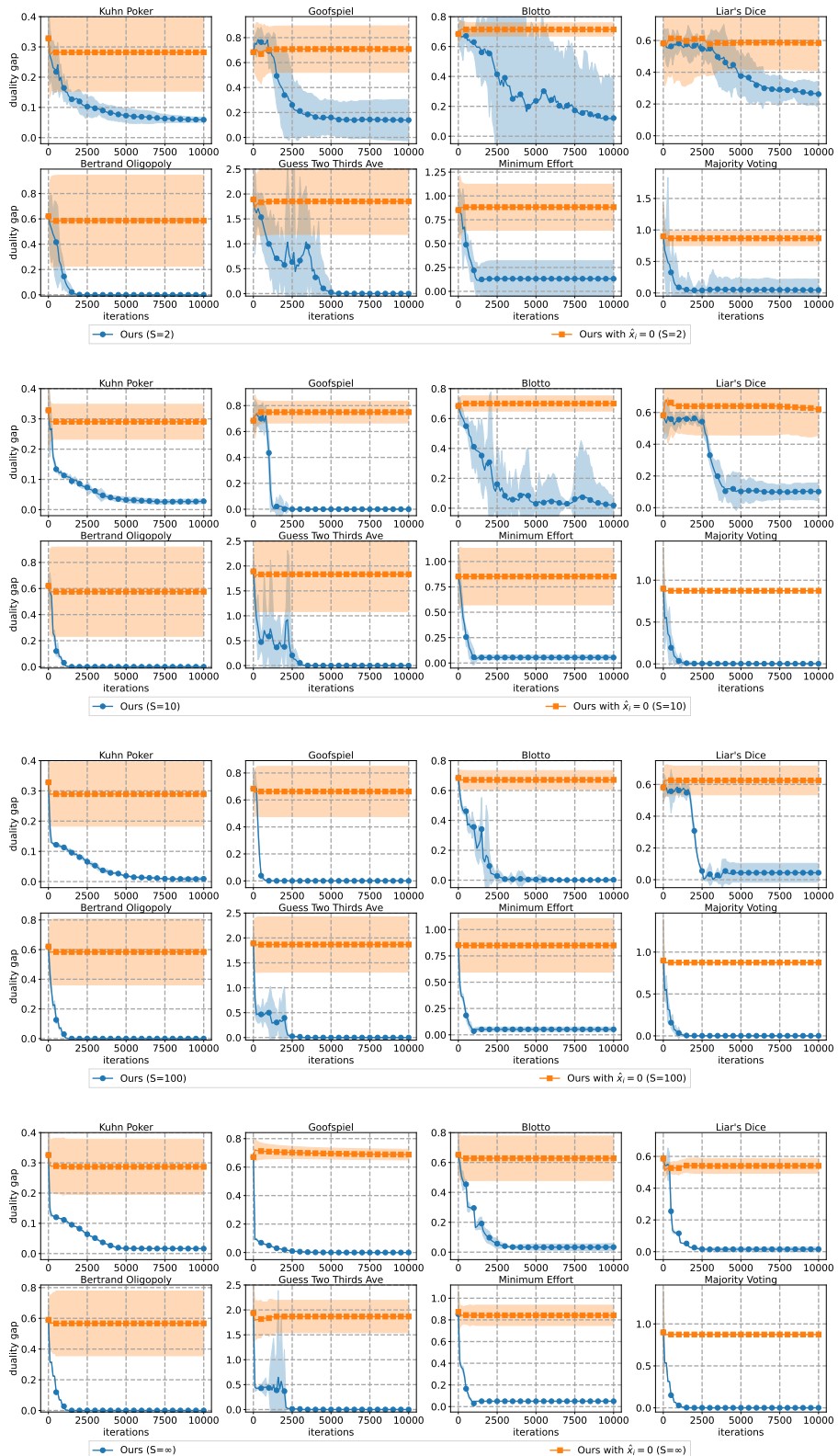

*Figure 21.* Empirical convergence rates of the algorithms that minimize NAL and NAL with $\hat{x}_i = 0$, respectively, when the optimizer is Adam and the activation function of the final layer is Sparsemax. The notation $S = \infty$ denotes the scenario where sampling is not used.

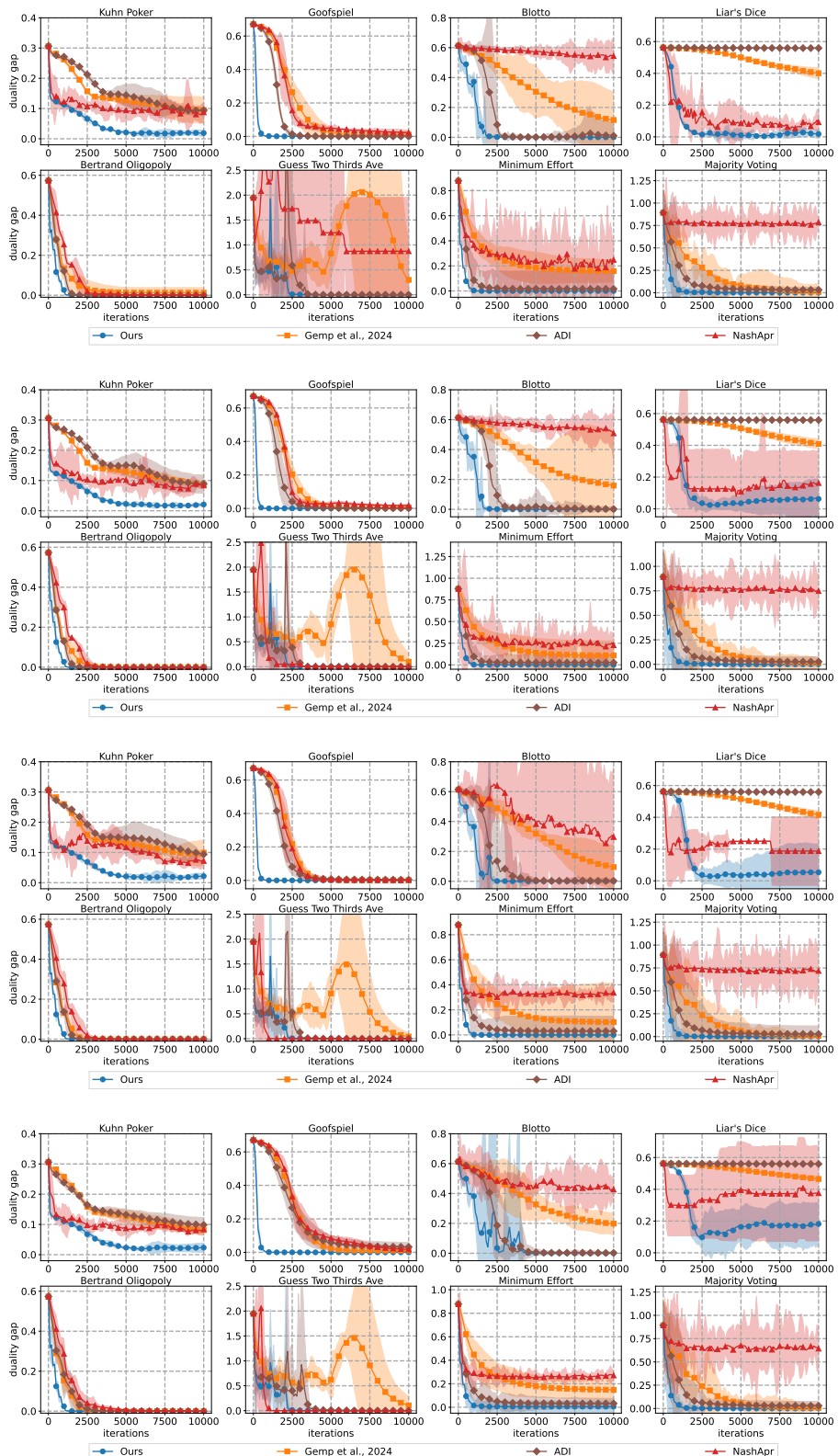

*Figure 22.* Empirical convergence rates of the algorithms that minimize NAL with various $\epsilon$ values. From top to bottom, $\epsilon$ takes values of 0, 0.1, 0.5, and 0.9.

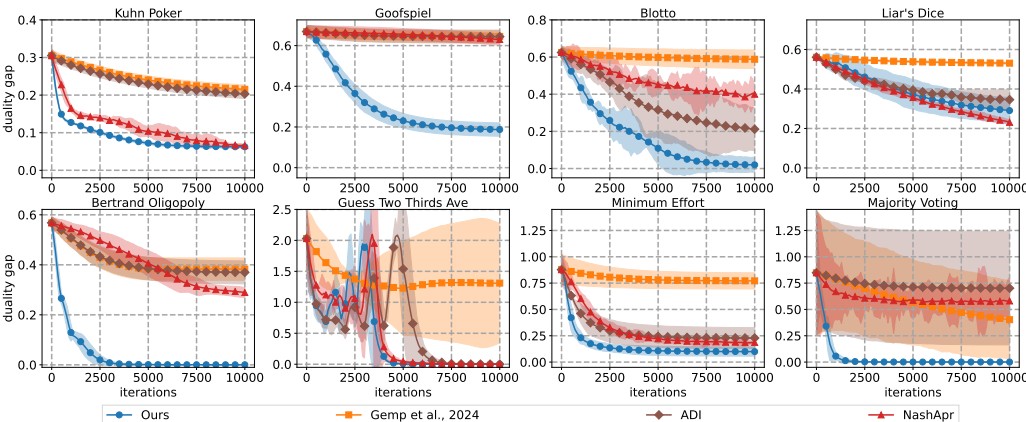

*Figure 23.* Empirical convergence rates of tested algorithms when the strategy is represented by a real vector rather than a DNN.

