# OpenReview forum: "Reducing Variance of Stochastic Optimization for Approximating Nash Equilibria in Normal-Form Games"
_ICML.cc/2025/Conference — ICML 2025 spotlightposter_

### Official Review · Reviewer_9MEJ · 2025-03-03

**Overall Recommendation:** 4

**Summary:**

This paper proposes NAL, a loss function that is unbiased and has lower variance compared with the only unbiased loss function proposed in Gemp et al. (2024). The paper conducts theoretical and empirical justifications to show that NAL theoretically and empirically exhibits lower variance and thus accelerates convergence.

**Claims And Evidence:**

Yes.

**Essential References Not Discussed:**

N/A

**Experimental Designs Or Analyses:**

Yes. The experiments in Section 5 well demonstrates the advantages of NAL compared with existing loss functions.

**Methods And Evaluation Criteria:**

Yes.

**Other Comments Or Suggestions:**

Comments:
C1: It seems that between RHS of line 380-383, the statement should be reverse (alternate Blotto and Liars Dice).
C2: I found the game size configurations in the caption of Figure 1. I think it should better appear in the main text, since the large game description is one of the motivation to estimate the loss.

**Other Strengths And Weaknesses:**

Strengths:
S1: The presentation is easy-to-follow and well-written.
S2: The observation of the unnecessary large variance in the loss function proposed by Gemp et al. (2024) is insightful.
S3: Both theoretical and empirical results about NAL are strong.

Weaknesses:
W1: The contribution of lower variance builds upon the idea of Gemp et al. (2024), which may slightly decrease the paper's originality.
W2: The theoretical justification of variance difference between NAL and loss in Gemp et al. (2024) is not rigorous, mainly for following reasons: 1) The expression derived between LHS of line 243-248 is an upper bound of the variance. Actually, to show the variance of Gemp et al. (2024) suffers from a $\sigma \max |\mathcal{A}_i|$ scaling, there should be an example demonstrating the $\Omega(|\mathcal{N}|\sigma^2 \max |\mathcal{A}_i|)$ lower bound, or show that the inequality between LHS of line 255-line 256 is tight in some sense. 2) The lower variance of the gradient is not the only evidence for faster convergence. The norm of the gradient should also be taken into consideration. For example, if you scale the NAL loss with a factor $\sigma \max |\mathcal{A}_i|$, then there seems to be no variance advantages for NAL.

**Questions For Authors:**

Questions:
Q1: In RHS of line 327, the authors mentioned that $\epsilon = 1$ in experiments. It means that $\hat{x}_i$ in NAL is always chosen to be the uniform strategy. Can authors provide more intuitions behind this practice?
Q2: In RHS of line 308-316, the authors use a DNN with constant input to represent a strategy. Why do not the authors choose to represent the strategy with real vectors directly? (with post-processed softmax activation)
Q3: How were the duality-gap and exact loss evaluated in experiments? Through brute-force computations or other cleverer approaches? It seems that the game size is not small and directly computing the duality gap and loss functions require costly computations.

**Relation To Broader Scientific Literature:**

The paper builds on the literature of equilibrium computation in normal-form games. Gemp et al., (2022) and Duan et al., (2023) propose biased loss functions, and Gemp et al. (2024) proposes an unbiased loss function with large variance. This paper contributes to the literature by proposing an unbiased loss function with lower variance. Besides, this paper also contributes to the general machine learning literature where unbiased estimators are key components for convergence of many first-order algorithms.

**Theoretical Claims:**

All proofs seem to be correct.

---

> ### Author Rebuttal · Authors · 2025-03-31
>
> Thank you for your positive assessment and helpful suggestions.
>
> **W1: The contribution of lower variance builds upon the idea of Gemp et al. (2024), which may slightly decrease the paper's originality.**
>
> **A:** Both our work and Gemp et al. (2024) explore leveraging the stochastic optimization techniques in ML for NE computation. However, our work is motivated by a distinct problem compared to Gemp et al. (2024). Gemp et al. (2024) resolve whether NE can be learned via these techniques in ML, while we address how efficiently NE can be learned through these techniques.
>
> Specifically, Gemp et al. (2024) mainly focus on removing bias introduced by previous loss functions, which could hinder convergence to NE when employing the stochastic optimization techniques in ML. In contrast, our work tackles the high variance issue within the loss function proposed by Gemp et al. (2024), as excessive variance can significantly slow down convergence when learning NE via the stochastic optimization techniques in ML. As you kindly pointed out, "the observation of the unnecessary large variance in the loss function proposed by Gemp et al. (2024) is insightful," we are the first to identify this high-variance issue and to propose a solution to it.
>
> ---
>
> **W2.1: The absence of a lower bound of the variance.**
>
> **A**: Now, we present that the variance in estimating ${L}^{\tau}\_{G}(x)$ is at least $\sigma \min\_{i \in N}|A\_i|$ times greater than that of NAL.
>
> Assume $Var[\bar{g}^{\tau,x,j}\_i(a\_i)] = \sigma$ (defined in Section 4.2, and $j \in \{ 1, 2\}$). Applying the derivations in Appendix D (where every "$\leq$" can be replaced with "$=$") to Section 4.2, for the loss in Gemp et al. (2024), we have
> $$
> Var[L^{\tau}\_{G}(x)] = \sum\_{i \in N} \sum\_{a\_i \in A\_i} Var[\bar{g}^{\tau,x,1}\_i(a\_i)\bar{g}^{\tau,x,2}\_i(a\_i)] \geq \sum\_{i \in N} \sum\_{a\_i \in A\_i} \sigma^2 \geq \sigma^2 |N|\min\_{i \in N}|A\_i|.
> $$
> Similarly, assuming $Var[\hat{g}^{\tau,x}\_i(a\_i)] = \sigma$ (defined in Section 4.2), we have
> $$
> Var[L^{\tau}\_{NAL}(x)] =
>    \sum\_{i \in N} \sum\_{a\_i \in A\_i} (x\_i(a\_i))^2 Var[\hat{g}^{\tau,x}\_i(a\_i)] = \sum\_{i \in N} \sum\_{a\_i \in A\_i} (x\_i(a\_i))^2 \sigma \leq \sigma|N|,
> $$
> where the last inequality comes from that $\sum\_{a\_i \in A\_i} (x\_i(a\_i))^2 \leq 1$. Clearly, the variance in estimating ${L}^{\tau}\_{G}(x)$ is at least $\sigma \min\_{i \in N}|A\_i|$ times greater than that of NAL, since intuitively, $\sigma$ increases with the size of the game and can grow significantly larger than 1.
>
> We will include this result in the revised version.
>
> ---
>
> **W2.2: The norm of the gradient should also be considered.**
>
> **A**: We sincerely appreciate your insightful suggestion. It is an important aspect that was not considered in our paper. However, since studying the norm of the gradient involves substantial theoretical and empirical investigation, and this paper primarily focuses on variance reduction, we leave a thorough exploration of the gradient norm to future work. Thank you for your thoughtful feedback and for helping to inform the direction of our ongoing research.
>
> ---
>
> **Q1: Why use $\epsilon=1$.**
>
> **A:** The reason is that $F^{\tau, x}_i - \overline{F^{\tau, x}_i}$ (used in Gemp et al. (2024)) and $F^{\tau, x}_i - \langle F^{\tau, x}_i , \hat{x}_i \rangle \mathbf{1}$ (used in NAL) is only equivalent when $\epsilon=1$ in Algorithm 1. This choice ensures a fair comparison between NAL and the loss in Gemp et al. (2024), as it mitigates the influence of the selection of $\hat{x}\_i$. To strengthen the robustness of our results, we include experiments with various $\epsilon$ values ($0$, $0.1$, $0.5$, and $0.9$), as shown in Figures 1–4 of https://anonymous.4open.science/api/repo/ICML-2025-ID-10862-Rebuttal/file/additional-experimental-results.pdf. Across all tested $\epsilon$ values, our algorithm consistently outperforms the others.
>
> ---
>
> **Q2: Why use a DNN rather than a real vector for strategy representation?**
>
> **A:**  We employ a DNN due to its capability to approximate arbitrary non-linear functions, enabling the discovery of complex equilibrium strategies that simpler representations may overlook (see Appendix A for a detailed discussion on the advantages of DNNs). In contrast, a real vector lacks this expressive power. The results, where the strategy is represented using a real vector, are shown in Figure 5 of https://anonymous.4open.science/api/repo/ICML-2025-ID-10862-Rebuttal/file/additional-experimental-results.pdf. All algorithms exhibit varying degrees of performance degradation, yet our algorithm still outperforms the others.
>
> ---
>
> **Q3: How were the duality-gap and exact loss evaluated in experiments? Through brute-force computations or other cleverer approaches?**
>
> **A:** Unfortunately, we rely on brute-force computation. This computation is used solely to assess the performance of the tested algorithms and are not involved in these algorithms' convergence process.

---

> > ### Comment · Reviewer_9MEJ · 2025-04-04
> >
> > Thank you for your response. All my concerns are resolved, and I will maintain my positive evaluation to this submission.

---

### Official Review · Reviewer_wd5S · 2025-03-12

**Overall Recommendation:** 4

**Summary:**

This paper studies computing Nash equilibria (NE) in normal-form games using non-convex stochastic optimization techniques from machine learning. The existing unbiased loss function for approximating NE has high variance, which degrades the convergence rate. To address this, the authors propose a novel surrogate loss function named Nash Advantage Loss (NAL). NAL is theoretically proven to be unbiased and has a significantly lower variance than the existing unbiased loss function. Experimental results on eight normal-form games show that the algorithm minimizing NAL converges much faster and has a lower variance in estimated loss values compared to other algorithms.

**Claims And Evidence:**

The claims made in the submission are supported by clear and convincing evidence

**Essential References Not Discussed:**

None.

**Experimental Designs Or Analyses:**

The experimental design and analysis sound.

**Methods And Evaluation Criteria:**

The proposed methods and evaluation criteria make sense for the problem.

**Other Comments Or Suggestions:**

NA

**Other Strengths And Weaknesses:**

NA

**Questions For Authors:**

NA

**Relation To Broader Scientific Literature:**

Approximating NE of normal-form games is a well-studied area. The main contribution of this paper is a novel surrogate loss function, which is theoretically proven to be unbiased and has a significantly lower variance than the existing unbiased loss function. As the author pointed out, this result show that the variance of the loss function may be one of the key issues influence the convergence rate for approximating NE.

**Theoretical Claims:**

As far as I checked, the proofs are correct.

---

> ### Author Rebuttal · Authors · 2025-03-31
>
> We sincerely appreciate your time and thoughtful review of our manuscript. Your positive feedback are truly encouraging our work. This paper focuses on leveraging the stochastic optimization techniques in ML to approximate NE of normal-form games. We propose a novel surrogate loss function, which has a significantly lower variance than the existing unbiased loss function. This reduction in variance accelerates the convergence rate for approximating NE. We hope our work can inspire more researchers in the community to engage with this emerging line of leveraging the stochastic optimization techniques in ML for NE computation.

---

### Official Review · Reviewer_NE5j · 2025-03-14

**Overall Recommendation:** 3

**Summary:**

The authors tackle the problem of NE computation in general-sum multiplayer NFGs, which is known to be a computational hard problem. They build on the work of Gemp et al. (2024) to come up with a novel approach involving a surrogate loss function they call Nash Advantage Loss. They show that NAL is unbiased and that its variance is lower compared to other surrogate loss functions. They provide empirical evidence to support these claims.

Post-rebuttal
While the authors' response was helpful, I do think that it would be valuable to see the exposition regarding the stop-gradient incorporated into the paper, in particular, a precise mathematical definition. While the references are helpful, given that it is a key notion in the paper, it should be defined at the beginning of the paper before the approach is explained in more detail. I will maintain my score.

**Claims And Evidence:**

Yes.

**Essential References Not Discussed:**

N/A

**Experimental Designs Or Analyses:**

Yes, the experimental design is sound.

**Methods And Evaluation Criteria:**

Yes.

**Other Comments Or Suggestions:**

1. In the Related Work section, it is a bit odd to refer to $\hat{\mathbf{x}}_i$ when you have not introduced any notation yet; it just leads to more confusion for the reader (especially, when you say your "algorithm does not support $\hat{\mathbf{x}}_i = 0$"; these statements would be more appropriately placed after notation has been introduced by your paper, and in the Related Work section, it would make sense to keep things more high-level without introducing notation.

**Other Strengths And Weaknesses:**

I have already commented on my confusion regarding the stop-gradient operator. The empirical results seem to suggest that the method is successful at reducing variance.

**Questions For Authors:**

1. Can the authors explain how the stop-gradient enables variance reduction in their stochastic optimization framework?

**Relation To Broader Scientific Literature:**

The authors have done a good job of contextualizing their work with respect to the broader literature, particular in appendices A and B.

**Theoretical Claims:**

I checked the proofs, although, it was unclear to me what the formal definition of the stop-gradient operator is; since the paper is about the novel loss function, and the analysis of the loss function hinges on the stop-gradient operator, it seems as though more exposition should be allocated to the stop-gradient operator.

---

> ### Author Rebuttal · Authors · 2025-03-31
>
> Thanks for your valuable and insightful comments.
>
> **Q1: The use of undefined symbols in the Related Work section.**
>
> **A:** We fully agree with your observation. The use of undefined symbols compromises the internal consistency of the paper. We will revise our paper to ensure that all symbols are properly introduced and clearly defined before they are used.
>
> ---
>
> **Q2: The definition of the stop-gradient operator.**
>
> **A:** We apologize for the confusion, the description of the stop-gradient operator in our paper (lines 175–178 and 767–770) is too brief. Below, we now provide a more detailed introduction to the stop-gradient operator.
>
> Let $b \in \mathbb{R}^n$ be a variable. The stop-gradient operator is defined as $sg\[b\] = b \in \mathbb{R}^n$ with $\nabla\_b sg\[b\] = 0 \in \mathbb{R}^{n \times n}$. This implies that $sg\[b\]$ passes the value of $b$ unchanged in the forward pass, but blocks its gradient during backpropagation. Intuitively, $sg[\cdot]$ can be regarded as a constant during differentiation. In summary,
>
> - **Forward pass:** $sg\[b\]$ returns the value of $b$.
> - **Backward pass:** The gradient is blocked—no gradients are propagated through $b$.
>
> We will provide a detailed definition of the stop-gradient operator in a future revision. In fact, this operator has already been widely adopted in previous works [1,2,3]. We are inspired by the use of this operator in previous works and adopt it in our work accordingly.
>
> References:
>
> 1. Grill, Jean-Bastien, et al. "Bootstrap your own latent: a new approach to self-supervised learning." *Advances in neural information processing systems* (NeurIPS). 2020.
> 2. Flennerhag, Sebastian, et al. "Meta-Learning with Warped Gradient Descent." *International Conference on Learning Representations* (ICLR). 2020.
> 3. Chen, Xinlei, and Kaiming He. "Exploring simple siamese representation learning." *Proceedings of the IEEE/CVF conference on computer vision and pattern recognition* (CVPR). 2021.
>
> ---
>
> **Q3: How the stop-gradient enables variance reduction？**
>
> **A:** Your observation is highly perceptive. The stop-gradient operator is crucial for variance reduction in our stochastic optimization framework. Specifically, it ensures that the variance of our NAL is determined solely by a single estimated variable. In contrast, the loss proposed by Gemp et al. (2024) involves the inner product of two independently estimated variables, leading to significantly higher variance. A more detailed explanation is provided below.
>
> The core idea behind our NAL is that the stochastic optimization techniques in ML rely solely on unbiased estimates of the first-order gradient, rather than the loss function itself. Leveraging this, we define NAL using the stop-gradient operator as $\langle sg\[b\], x\_i \rangle$, where the backpropagated gradient is simply $sg\[b\]$. Here, $b$ is an estimate of the first-order gradient, defined as ${F}^{\tau, x}\_i - \langle {F}^{\tau, x}\_i , \hat{x}\_i \rangle \mathbf{1}$, with $\hat{x}\_i$ being any strategy. Formally, for each $a\_i \in A\_i$,
>
> $ \quad   \nabla\_{x\_i(a\_i)} \langle sg\[b\], x\_i \rangle $
>
> $= \nabla\_{x\_i(a\_i)} \sum\_{a'\_i \in A\_i} sg\[b\](a'\_i) x\_i(a'\_i) $
>
> $= \sum\_{a'\_i \in A\_i} \nabla\_{x\_i(a\_i)} (sg\[b\](a'\_i) x\_i(a'\_i)) $
>
> $= \sum\_{a'\_i \in A\_i} \left( \nabla\_{x\_i(a\_i)} sg\[b\](a'\_i) \right) x\_i(a'\_i) + sg\[b\](a'\_i) \nabla\_{x\_i(a\_i)} x\_i(a'\_i) ) $
>
> $= \sum\_{a'\_i \in A\_i} ( \nabla\_b sg\[b\](a'\_i) \nabla\_{x\_i(a\_i)} b ) x\_i(a'\_i) + sg\[b\](a\_i) $
>
> $= sg\[b\](a\_i),$
>
>
> where the last equality follows from $\nabla\_b sg\[b\] = 0 \in \mathbb{R}^{n \times n}$ (considering $sg\[b\]$ as a constant during the differentiation process makes this process clearer and more understandable).
>
> In contrast, Gemp et al. (2024) define a loss based on the inner product $\langle b', b'' \rangle$, where $b'$ and $b''$ are independent estimates of ${F}^{\tau, x}\_i - \overline{{F}^{\tau, x}\_i}$. Notably, ${F}^{\tau, x}\_i - \overline{{F}^{\tau, x}\_i}$ and ${F}^{\tau, x}\_i - \langle {F}^{\tau, x}\_i , \hat{x}\_i \rangle \mathbf{1}$ is equivalent when $\hat{x}\_i$ is set to the uniform strategy, which is the setup used in our experiments. As analyzed in Section 4.2, the variance of our NAL estimate scales **linearly** with the variance of $b$, while the variance of Gemp et al.’s loss scales **quadratically** with the variance of $b'$ (and $b''$).

---

### Official Review · Reviewer_f3FW · 2025-03-15

**Overall Recommendation:** 4

**Summary:**

This paper addresses the challenge of efficiently computing Nash equilibria (NE) in normal-form games (NFGs) via non-convex optimization. Prior work, notably by Gemp et al. (2024), proposed an unbiased loss function for this purpose but encountered high variance, which hindered convergence. To overcome this, the authors introduce a novel surrogate loss function - the Nash Advantage Loss (NAL) - which remains unbiased and exhibits significantly lower variance. Empirical results indicate that minimizing NAL yields much faster convergence rates than existing methods. Extensive experiments conducted on eight different NFGs validate the efficacy of the proposed approach, including comparisons across various optimizers and network structures.

**Claims And Evidence:**

Yes (see the strengths discussion below).

**Essential References Not Discussed:**

--

**Experimental Designs Or Analyses:**

Yes (see the strengths discussion below).

**Methods And Evaluation Criteria:**

Yes (see the strengths discussion below).

**Other Comments Or Suggestions:**

--

**Other Strengths And Weaknesses:**

Strengths:
- The introduction of the Nash Advantage Loss (NAL) is a notable contribution. Its key properties - being unbiased and having reduced variance - directly address the limitations of previous approaches, thereby enhancing the efficiency of NE computation.
- The theoretical sections are presented rigorously. Although I did not check the details of the proofs (given my limited familiarity with this literature), the technical statements appear well-founded.
- The paper provides a systematic empirical evaluation of the proposed method against multiple baselines. The experiments not only compare convergence rates and variance of the loss estimates but also explore different optimizers and network structures, consistently demonstrating better performance of the proposed approach.

Weaknesses:
- The technical sections are dense, and navigating through the various notations and definitions can be challenging. Including a comprehensive table of notations and definitions in the appendix would greatly aid readers.

**Questions For Authors:**

--

**Relation To Broader Scientific Literature:**

The paper proposes a novel loss function to efficiently compute NE in NFGs.

**Theoretical Claims:**

Although I did not check the details of the proofs (given my limited familiarity with this literature), the technical statements appear well-founded.

---

> ### Author Rebuttal · Authors · 2025-03-31
>
> Thank you for your recognition of our work. In response to your suggestion regarding readability, we have added a table of notations and definitions, as presented in Table 1 of https://anonymous.4open.science/api/repo/ICML-2025-ID-10862-Rebuttal/file/notation-table.pdf.

---

### Decision · Program_Chairs · 2025-05-01

**Decision:**

Accept (spotlight poster)

**Comment:**

All reviewers unanimously agree that this paper makes nice contribution to the fundamental problem of approximating NE in normal-form games by proposing a new surrogate loss that leads to lower variance. The theoretical findings are backed up with extensive experiments on real data.  This is a clear accept.